# Structure of the far-red light utilizing photosystem I of *Acaryochloris marina*

Tasuku Hamaguchi [1,9], Keisuke Kawakami[2,8,9 ✉], Kyoko Shinzawa-Itoh[3,9], Natsuko Inoue-Kashino [3], Shigeru Itoh [4], Kentaro Ifuku [5], Eiki Yamashita [6], Kou Maeda[3], Koji Yonekura [1,7 ✉] & Yasuhiro Kashino [3 ✉]

*Acaryochloris marina* is one of the cyanobacterial species that can use far-red light to drive photochemical reactions for oxygenic photosynthesis. Here, we report the structure of *A. marina* photosystem I (PSI) reaction center, determined by cryo-electron microscopy at 2.58 Å resolution. The structure reveals an arrangement of electron carriers and light-harvesting pigments distinct from other type I reaction centers. The paired chlorophyll, or special pair (also referred to as P740 in this case), is a dimer of chlorophyll *d* and its epimer chlorophyll *d'*. The primary electron acceptor is pheophytin *a*, a metal-less chlorin. We show the architecture of this PSI reaction center is composed of 11 subunits and we identify key components that help explain how the low energy yield from far-red light is efficiently utilized for driving oxygenic photosynthesis.

[1] Biostructural Mechanism Laboratory, RIKEN SPring-8 Center, Sayo, Hyogo, Japan. [2] Research Center for Artificial Photosynthesis (ReCAP), Osaka City University, Sumiyoshi-ku, Osaka, Japan. [3] Graduate School of Life Science, University of Hyogo, Ako-gun, Hyogo, Japan. [4] Department of Physics, Graduate School of Science, Nagoya University, Nagoya, Japan. [5] Division of Integrated Life Science, Graduate School of Biostudies, Kyoto University, Sakyo-ku, Kyoto, Japan. [6] Laboratory of Supramolecular Crystallography, Institute for Protein Research, Osaka University, Suita, Osaka, Japan. [7] Institute of Multidisciplinary Research for Advanced Materials, Tohoku University, Aoba-ku, Sendai, Japan. [8] Present address: Biostructural Mechanism Laboratory, RIKEN SPring-8 Center, Sayo, Hyogo, Japan. [9] These authors contributed equally: Tasuku Hamaguchi, Keisuke Kawakami, Kyoko Shinzawa-Itoh. ✉email: kawakami.k@spring8.or.jp; yone@spring8.or.jp; kashino@sci.u-hyogo.ac.jp

Photosynthesis is driven by two types of photoreaction systems, namely type I and type II reaction centers. Photosystems I (ref. [1]) and II (ref. [2]; PSI and PSII, belonging to type I and type II reaction centers, respectively) work sequentially in oxygenic photosynthesis in plants and cyanobacteria[3]. They typically use chlorophyll (Chl) *a* (Supplementary Fig. 1) as the major pigment that absorbs visible light (670–700 nm in the red-light region; Supplementary Fig. 2). Anoxygenic photosynthetic bacteria use bacteriochlorophylls (BChl) that absorb 800–900-nm light in addition to blue/ultraviolet light and have either a type I or type II reaction center. *Acaryochloris marina* (*A. marina*) was found as a cyanobacterial species that uses Chl *d* (Supplementary Fig. 1), which absorbs 700–750-nm light in vivo[4] (Supplementary Fig. 2). Later, several cyanobacteria were found that induce a limited amount of Chl *f* for capturing far-red light under the far-red light condition[5–7] (Supplementary Fig. 2, and see below). *A. marina* was isolated from colonial ascidians, which harbor mainly Chl *a*-type cyanobacteria, resulting in an environment with low visible light and high far-red light. Analogous to the cortex effect within a lichen body[8], high-energy blue light absorbed by the Chls likely has diminished intensity within the symbiotic host ascidians. *A. marina* has exploited this environment with the niche-filling introduction of Chl *d* that absorbs far-red light[9,10]. The discovery of *A. marina* prompted intensive research into the mechanism of this low-energy-driven system. Although the organism also contains Chl *a* at ~5% relative abundance, spectroscopic and pigment analyses revealed that it is Chl *d* that plays a central role in the photoreaction[4,9,11]. With the acquisition of Chl *d*, *A. marina* harnesses far-red light, the energy of which is lower than that of visible light that is utilized in most oxygenic photosynthetic reactions. Chl *d* has a peak wavelength of absorption at ~697 nm in methanol (Supplementary Fig. 3; so-called Qy band; 700–750 nm in vivo, Supplementary Fig. 2), which is longer than the 665.2 nm of Chl *a* (670–700 nm in vivo, Supplementary Figs. 2 and 3) by ~30 nm. This means that the photon energy absorbed is ~80 mV (10%) lower. How PSI and PSII of *A. marina* drive the similar photochemical reactions that occur in the Chl *a*-dependent systems of other oxygenic photosynthetic organisms has been a long-standing puzzle.

PSI generates reducing power for NADPH production by accepting electrons originating from PSII[3]. PSI of *A. marina* has a similar subunit composition to that of the PSIs of other oxygenic photosynthetic organisms[12,13] (Supplementary Table 1). However, the peak wavelength of the light-induced redox difference absorption spectrum of the paired Chls, the so-called special pair, in *A. marina* PSI is longer (740 nm) than that of Chl *a* (700 nm) in the PSI of plants and typical cyanobacteria[14] (Fig. 1a). The special pair P740 have been assumed to be a heterodimer of Chl *d* and *d*′ (refs. [15,16]) as in P700, which is composed of a heterodimer of Chl *a*/Chl *a*′ in all other PSIs (Fig. 1a)[1,17]. Cofactors of the PSI electron transfer chain in *A. marina* (Fig. 1b) and the reduction potential of P740 are also similar to those of P700 in other organisms[12,13,18].

The far-red light-absorbing Chl, Chl *f*, is also utilized in some cyanobacteria to capture far-red light (Supplementary Fig. 2). How Chl *f*, as well as Chl *d*, has succeeded in expanding harvesting the low-energy far-red light region is of great interest[5–7]. Chl *f* is only induced under far-red light conditions, and even then makes up less than ~10% of total Chls in PSI (seven Chl *f* and 83 Chl *a* in *Halomicronema hongdechloris* PSI[19]), and the locations of these Chl *f* molecules is still debated[20,21]. Recent cryo-EM analyses show that Chl *f* is not part of the charge-separating pigments, including and following the special pair[19,22,23]. Thus, Chl *f* in these PSIs is likely entirely responsible for light harvesting. High-resolution[24] and ultrafast[25] spectroscopic studies support this. Still, PSII in some cyanobacteria may

use Chl *f* (and/or Chl *d*) for electron transfer when grown under far-red light conditions[21,26]. In contrast to Chl *f*-carrying PSIs[19,20,22,23,25,27], Chl *d* in *A. marina* PSI is always induced even under natural white light and does, in fact, take the place of the special pair Chls in the photochemical reaction chain. The Chl *d*-driven photoreaction occurs by low-energy far-red light corresponding to the Qy band of Chl *d*. Chl *d* also absorbs high-energy blue light, corresponding to the Soret band, but the resulting excited state relaxes to a lower excited state corresponding to the Qy level for the photochemical reaction. In this sense, the Chl *d*-driven PSI-system is unique among photochemical reaction centers, utilizing as it does far-red light level energy.

To date, several structures of type I reaction centers have been determined from higher plants[28,29], green algae[30,31], red algae[32], diatoms[33], cyanobacteria[1,19,22,23,34,35], and an anoxygenic bacterium[36]. All structures of type I reaction centers including Chl *f*-carrying cyanobacteria[19,22,23] have Chl *a* in the electron transfer chain with the one exception, namely, the one from the anoxygenic bacterium that has BChl *g*′ and $8^1$-OH-Chl *a* (ref. [36]). The PSI of *A. marina* with Chl *d* in its electron transfer chain for utilizing low-energy light directly for the photochemistry is thus unique.

Here, we show the structure of the PSI trimer isolated from *A. marina* revealed by cryo-electron microscopy (cryo-EM) at 2.58 Å resolution. The special pair P740 is a dimer of Chl *d* and its epimer Chl *d*′, and the primary electron acceptor is pheophytin (Pheo) *a* not Chl *a* nor Chl *d*, which is embedded in membrane protein complex composed of 11 subunits. The structure helps gain insight into the mechanism for the utilization of far-red light (Fig. 1b).

## Results and discussion

**Overall structure, protein subunits, and cofactors of *A. marina* PSI.** The PSI trimer complex was isolated from *A. marina*. Detailed methods of sample preparation, data processing, and structural refinement by cryo-EM, and biochemical data, are presented in Supplementary Information (Supplementary Figs. 4–12 and Supplementary Tables 1–4). The model of the PSI trimer was refined to give a correlation coefficient and Q-score, which are indices for validating the correctness of a model, of 0.80 and 0.65, respectively (Supplementary Table 4). These index values can be considered reasonable at 2.58 Å resolution (Supplementary Fig. 11).

The PSI trimer has dimensions of 100 Å depth, 200 Å length, and 200 Å width, including the surrounding detergent micelle (Fig. 2). The overall structure resembles those of PSI trimers reported for other cyanobacteria[1,22,23,35]. The root-mean-square deviations using secondary structure matching between the monomer model (chains A–M) and Protein Data Bank (PDB) structures 1JB0 of *Thermosynechococcus elongatus*[1], 5OY0 of *Synechocystis* sp. PCC 6803[35], and 6KMX of *H. hongdechloris*[19] are 1.09 Å, 1.05 Å, and 1.11 Å, respectively.

The *A. marina* PSI monomer contains 11 subunits (PsaA, PsaB, PsaC, PsaD, PsaE, PsaF, PsaJ, PsaK, PsaL, PsaM, and Psa27), all of which could be assigned to the cryo-EM density map (Fig. 2, Supplementary Figs. 6 and 13, and Supplementary Table 1). PsaX (a peripheral subunit in other cyanobacteria such as *T. elongatus*), PsaG, and PsaH (subunits in higher plants) were missing in the density map, consistent with the absence of their genes from the *A. marina* genome[10]. The name Psa27 was given to a subunit protein in *A. marina* that has low sequence identity (29.4%) with PsaI of *T. elongatus* (Supplementary Table 5)[12]. Here, we found that Psa27 is in the same location as PsaI, a transmembrane alpha helix, in *T. elongatus*. Psa27 also contributes to the structural stabilization of the PSI trimer just

**Fig. 1 Comparison of electron transfer chains in PSI. a** Cyanobacterial and higher plant PSI. **b** *A. marina* PSI identified in this work. In cyanobacterial and higher plant PSI, electrons released from the paired Chls, so-called special pair, of P700, a heterodimer of Chl *a* and *a'*, are transferred to ferredoxin (Fd) to reduce $NADP^+$ via Acc (Chl *a*), $A_0$ (Chl *a*), $A_1$ (PhyQ), $F_X$ (iron–sulfur center), and $F_A/F_B$ (iron–sulfur center). The components are arranged in a pseudo-$C_2$ axis on a heterodimer reaction center protein complex (PsaA/PsaB) and the two routes (A- and B-branch located on PsaA and PsaB, respectively) are thought to be equivalent. Acc accessory chlorophyll, $A_0$ primary electron acceptor, $A_1$ secondary electron acceptor, $F_X$, $F_A$, and $F_B$ iron–sulfur centers, PhyQ phylloquinone. In *A. marina* PSI, the electron transfer chain is the same, but some cofactors differ from those in the PSIs of other organisms; the Chls of special pair P740 are Chl *d/d'*, while Acc could be Chl *d* (see text for detail) and $A_0$ is pheophytin (Pheo) *a*.

as PsaI does in *T. elongatus*. Thus, we concluded that Psa27 and PsaI are in effect the same subunit.

The loop regions of PsaB (residues 290−320 and 465−510) were not identified due to disorder because of the absence of PsaX, suggesting that PsaX stabilizes these regions in other cyanobacteria. Most of subunits PsaE, PsaF, and PsaK were modeled using polyalanine because of poor regional map quality (Supplementary Figs. 13 and 14, and Supplementary Table 1), possibly suggesting that some part of those subunits dissociated during sample preparation.

The cofactors assigned (Supplementary Fig. 15) are 70 Chl *d* (Supplementary Data 1), 1 Chl *d'* (an epimer of Chl *d*), 12 α-carotenes (α-Car) but no β-Car (Supplementary Table 6)[10,14], 2 Pheo (a derivative of Chl; however, unlike Chl, no $Mg^{2+}$ ion is coordinated by the tetrapyrrole ring), 2 phylloquinones (PhyQs), 3 iron–sulfur clusters, 2 phosphatidylglycerols (PGs), 1 mono-galactosyl diacylglycerol (MGDG), and 84 water molecules. Two molecules of Pheo were assigned as Pheo *a* on the basis of pigment analysis (Supplementary Table 6). In addition, a small amount of Chl *a* (approximately one Chl *a* per Chl *d'*) was detected by pigment analysis (Supplementary Table 6). In this study, we assigned the Chls of *A. marina* PSI as Chl *d*. This is because the amount of Chl *a* in the *A. marina* PSI is minimal, and it is not possible to distinguish between Chl *d* and Chl *a* at 2.58 Å resolution, as is described later.

In the *A. marina* PSI, a large number of Chl *d* and α-Car harvest light energy and finally transfer the energy to P740. The arrangements of Chl *d* and α-Car in *A. marina* resemble those of Chl *a* and β-Car, respectively, in *T. elongatus*, as shown in Supplementary Fig. 16. The pigments are numbered according to the nomenclature of Chl *a* in *T. elongatus*[37] (Supplementary Data 1). However, there are differences in some amino acid residues surrounding the cofactors, slight gaps in the arrangement of the cofactors, and absence of some Chls in the *A. marina* PSI

compared with those of *T. elongatus* (Supplementary Table 5 and Supplementary Figs. 16–18). For example, Phe49/J in *A. marina*, the counterpart of His39/J that is an axial ligand of Chl *a* 88 in *T. elongatus* PSI, explains the absence of the corresponding Chl *d* (capitalized letters following the slash (/) indicate the subunit names, such as PsaA or PsaB; Supplementary Fig. 16d). Owing to the absence of PsaX in *A. marina*, the Chl *d* corresponding to Chl *a* 95 (refs. [1,38]) in *T. elongatus* whose axial ligand is Asn23/X is also missing. The absence of Chl *a* 94 is seen in some other reported PSI structures, such as *Synechocystis* sp. PCC 6803 (ref. [35]). The arrangements of pigments adjacent to Psa27 (PsaI) and PsaL, whose sequence identity with other cyanobacterial PsaLs is low, are specific to *A. marina* (Supplementary Fig. 16a, black dashed line region, and Supplementary Figs. 17 and 18). Chl *d* 38, 52, 53, and the ring (α or ε) of Car4007 near Chl *d* 53 are set within the surrounding structure differently from those in *T. elongatus* (Supplementary Fig. 17a). These differences when compared with the structure of *T. elongatus* help explain the specific features of light harvesting in *A. marina*. In addition, the C3-formyl groups of some Chl *d* in the *A. marina* PSI form hydrogen bonds with their surrounding amino acid residues (Supplementary Data 1 and Supplementary Fig. 18). These characteristic structural features will be important in future theoretical studies of the light-harvesting mechanism in *A. marina*.

Although the amount of Chl *d* per Chl *d'* determined by pigment analysis (67.0 ± 0.66, *n* = 5 of independently prepared PSI) and assigned by structural analysis (70 including one or two Chl *a*) is lower in this study than those in previous studies (145 ± 8, ref. [14] and 97.0 ± 11.0, ref. [12]), the semi-stoichiometric amount of Phe *a* at 1.92 ± 0.022 (0.3 ± 0.2 per reaction center in a previous study[12]) is consistent with our structural analysis. Notwithstanding the fact that the local resolution of some parts of the outer region is somewhat lower, the PSI was stable enough to keep its

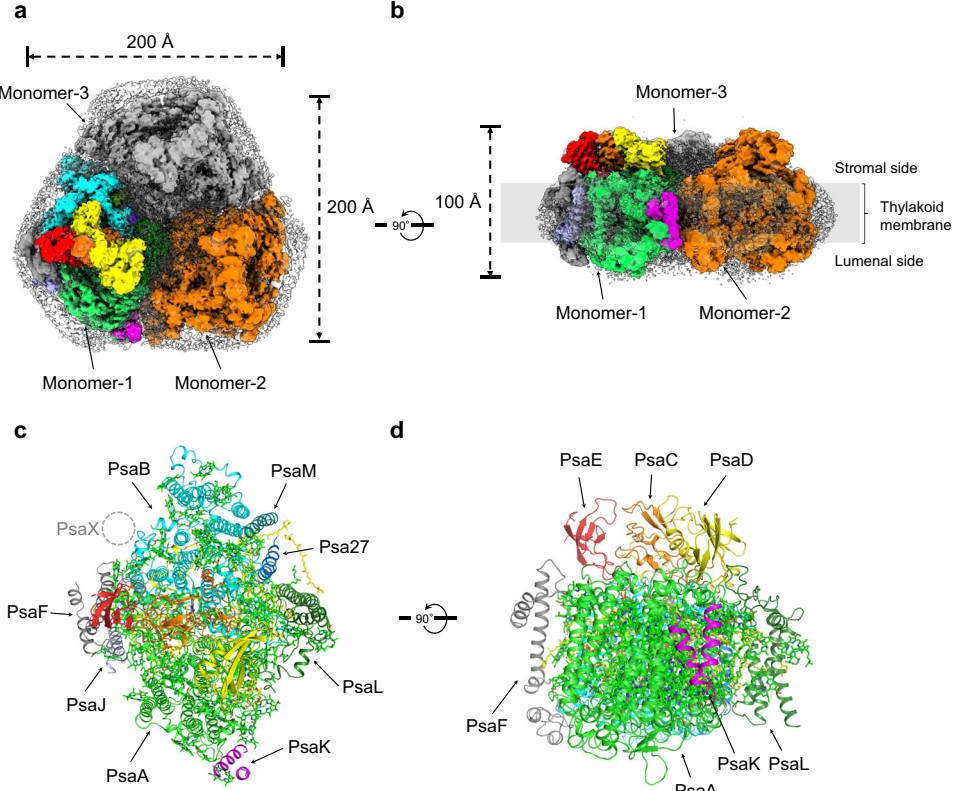

**Fig. 2 Overall structure of the photosystem I (PSI) trimer from *Acaryochloris marina*. a** Cryo-electron microscopy (cryo-EM) density map of PSI trimer viewed from the stromal side perpendicular to the membrane plane. Monomer-1, multicolored; monomer-2, orange; monomer-3, gray; detergent micelle, cloudy gray. **b** Cryo-EM density map of PSI trimer viewed from the side of the membrane plane. **c** Structural model of PSI monomer with view in the same direction as in **a**. PsaA, green; PsaB, cyan; PsaC, orange; PsaD, yellow; PsaE, red; PsaF, gray; PsaJ, light blue; PsaK, magenta; PsaL, dark green; PsaM, teal blue; Psa27, blue. **d** Structural model of PSI monomer with view in the same direction as in **b**.

integrity for 5 days (Supplementary Fig. 5). Chls that were unable to be assigned in *A. marina* PSI compared with those in *T. elongatus* PSI are shown by marking N/D in the column of the ID number in Supplementary Data 1. Such Chls can be recognized by the chlorins in transparent gray in Supplementary Fig. 16. Chls corresponding to Chl-88 and Chl-94 in *T. elongatus* appear to be absent in *A. marina* PSI. The numbers of Chls in the PSI of Chl *f*-carrying *Fischerella thermalis* (89, ref. [22]) and *H. hongdechloris* (90, ref. [19]) are also lower than those of *T. elongatus* PSI (96, ref. [1]). The type I reaction center of *Heliobacterium modesticaldum* carries a much smaller amount of Chl species; 60 molecules per reaction center[36].

Due to the somewhat lower resolutions of the outer regions of the *A. marina* PSI, we mainly focus on the structure and function of the central part of *A. marina* PSI at high local resolution, that is, P740 and the electron transfer components.

**Electron transfer components.** The configuration of the electron transfer chain in *A. marina* PSI is similar to that of the PSI from *T. elongatus* (Fig. 1), although cofactor compositions are different. The important assigned cofactors involved in electron transfer (Fig. 1b) are four Chls, two Pheos, two PhyQs (A₁), and three iron–sulfur clusters ($F_X$, $F_A$, and $F_B$; Fig. 3a). The Chls and PhyQs are arranged in two branches, the A-branch and the B-branch, which are related by a pseudo-$C_2$ axis as in other type I reaction centers, and are stabilized by amino acid residues of subunits PsaA and PsaB. The Chls of special pair P740 are Chl $d'$ ($P_A$) and Chl $d$ ($P_B$), which are coordinated by residues His678/A and His657/B, respectively (Fig. 3b, c). The distance between the ring planes (the π–π interaction distance) of Chl $d'$ and Chl $d$ is 3.5 Å

(Fig. 3d; 3.6 Å in *T. elongatus*). Two Tyr residues, Tyr601/A and Tyr733/A, are positioned within hydrogen bonding distance of $P_A$ as observed in PSI from *T. elongatus*.

In *A. marina* PSI, we identified three water molecules (W1 − W3) around $P_A$. They form hydrogen bonds with surrounding amino acid residues (Tyr601/A, Ser605/A, Asn608/A, Ser741/A, and Try745/A) and one Chl $d$ (Chl $d$ 32; Fig. 3b, c). For the *T. elongatus* $P_A$, only one water molecule forms hydrogen bonds with surrounding amino acid residues (Tyr603/A, Ser607/A, Ile610/A, Thr743/A, and Phe747/A) and Chl $a'$ ($P_A$). There are no water molecules around $P_B$ in *A. marina*, but some water molecules surround $P_B$ without forming hydrogen bonds. The arrangement of these water molecules differs from that of *T. elongatus* PSI, and this difference around $P_B$ may come from the difference in two amino acid residues: Val594/B in *A. marina* vs. Thr597/B in *T. elongatus*, and Asn598/B in *A. marina* vs. His601/B in *T. elongatus*. The hydrogen bonding pattern around $P_A$ probably contributes to the charge distribution ratio ($P_A^{\cdot+}$/$P_B^{\cdot+}$)[1,39], and therefore, is likely different in the two organisms. The midpoint potential values, $E_m$ ($P_A$) and $E_m$ ($P_B$), are influenced by the protein environment, in particular by the presence of charged residues. The $E_m$ difference, $E_m$ ($P_A$) − $E_m$ ($P_B$) ($=\Delta E_m$), is an important factor in determining the $P_A^{\cdot+}$/$P_B^{\cdot+}$ ratio[40].

The orientations of the formyl group in $P_A$ (Chl $d'$) and $P_B$ (Chl $d$) were identified by considering the distribution of the cryo-EM density (Fig. 4a, b). No amino acid residues and pigments capable of forming hydrogen bonds with these formyl groups were found in the vicinity of $P_A$ and $P_B$. This suggests that these formyl groups form hydrogen bonds with the C5 H atom in

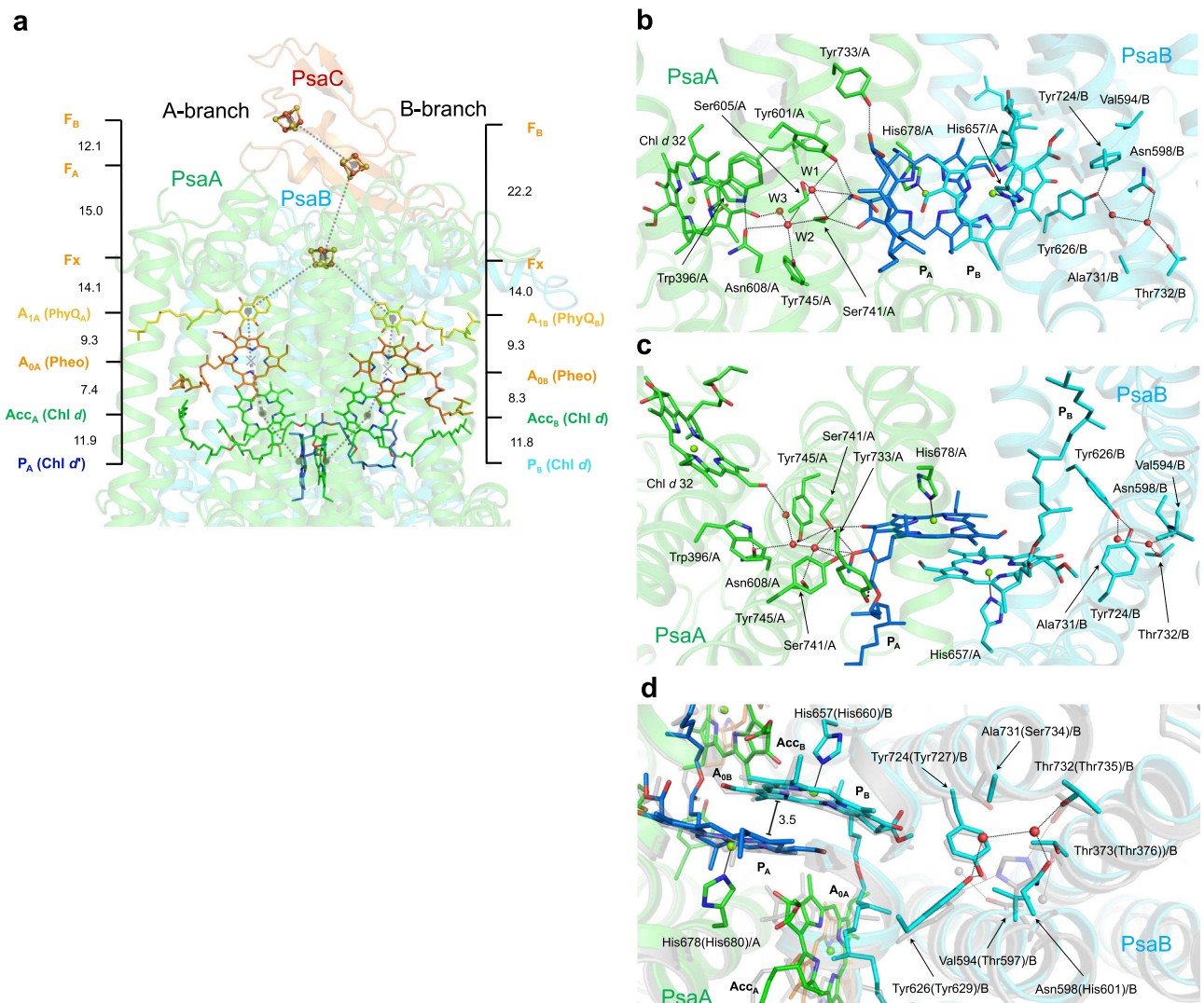

**Fig. 3 Cofactors involved in the electron transfer reaction in *A. marina* PSI and their surrounding amino acid residues. a** Arrangement of cofactors involved in the electron transfer reaction. Values on both sides represent the center–center distances (in Å) between the cofactors. **b** Arrangement of P740 and its surrounding structure. View from the side of the membrane plane. **c** As in **b**, but view from the pseudo-$C_2$ axis. **d** Arrangement of the amino acid residues influencing $E_m$ ($P_A$) and $E_m$ ($P_B$). Superposition of the structural model of PSI from *A. marina* with that from *T. elongatus* (PDB code 1JB0). Transparent green and blue, PSI from *A. marina*; transparent gray, PSI from *T. elongatus*. Chlorophyll (Chl) *d*, green; Chl *d'*, blue; water molecule, red; pheophytin (Pheo) *a*, orange, phylloquinone (PhyQ), yellow.

Chl *d'* and Chl *d*, respectively. Similarly, there were no amino acid residues and pigments capable of forming a hydrogen bond around the formyl group of Acc (Acc$_A$ and Acc$_B$; Fig. 4c, d). However, the orientations of the formyl group of Acc (Acc$_A$ and Acc$_B$) were altered by the hydrophobic environment caused by Trp (Trp585/B and Trp599/A) when compared with P740.

Previous studies have suggested that the primary electron acceptor A$_0$ in *A. marina* PSI (Fig. 1a) is Chl *a* (refs. [13,41]), in line with other species[1,42]. However, we found that there was no Mg$^{2+}$-derived density at the center of the tetrapyrrole rings of A$_{0A}$ and A$_{0B}$, but rather a hole in the cryo-EM density map (Figs. 4e, f and 5a). This indicates, when combined with the result of pigment analysis (Supplementary Table 6 and Supplementary Fig. 19), that A$_0$ is actually Pheo *a*, a derivative of Chl *a*. Furthermore, the position of Leu665/B (Supplementary Fig. 20b)[43] pointing to A$_{0B}$, supports Pheo *a* as A$_{0B}$ in *A. marina*, because Leu cannot serve as a ligand to Mg$^{2+}$ in Chl. Thus, surprisingly, A$_{0A}$ and A$_{0B}$ are assigned as Pheo *a* in *A. marina* PSI (Figs. 1b, 3a, and 5a–d). The present study reveals that the axial amino acid residues at the A$_{0A}$

and A$_{0B}$ sites are Met686/A and Leu665/B, respectively. In comparison, the A$_{0A}$ and A$_{0B}$ sites in *T. elongatus* PSI are both occupied by Chl *a* molecules whose central Mg$^{2+}$ ions are both coordinated by Met residues[1,43]. At present, it is unclear how the *A. marina* PSI selectively binds Pheo *a* at the A$_0$ sites instead of other pigments (Chl *a* or Chl *d*). However, the switch from Chl *a* to Pheo *a* in the A$_{0A}$ and A$_{0B}$ sites of *A. marina* PSI is probably not entirely dependent on the difference in its central ligand.

The amino acid residue nearest to A$_{0A}$ in *A. marina* is Met (Met686/A; Supplementary Fig. 20a)[43] and the identity of close amino acid residues may modify the reduction potentials of the two A$_0$ sites and their absorption peak wavelength(s). Two mechanisms have been proposed for electron transfer in PSI reaction centers, one using both A- and B-branches[44,45], and the other using the A-branch preferentially[46,47]. Only the A-branch may be active in *A. marina* PSI[43]. Two previous studies have reported different mechanisms for the delocalization of the charge distribution in P740 (refs. [48,49]). Future theoretical studies using the *A. marina* PSI structure may throw more light on the details

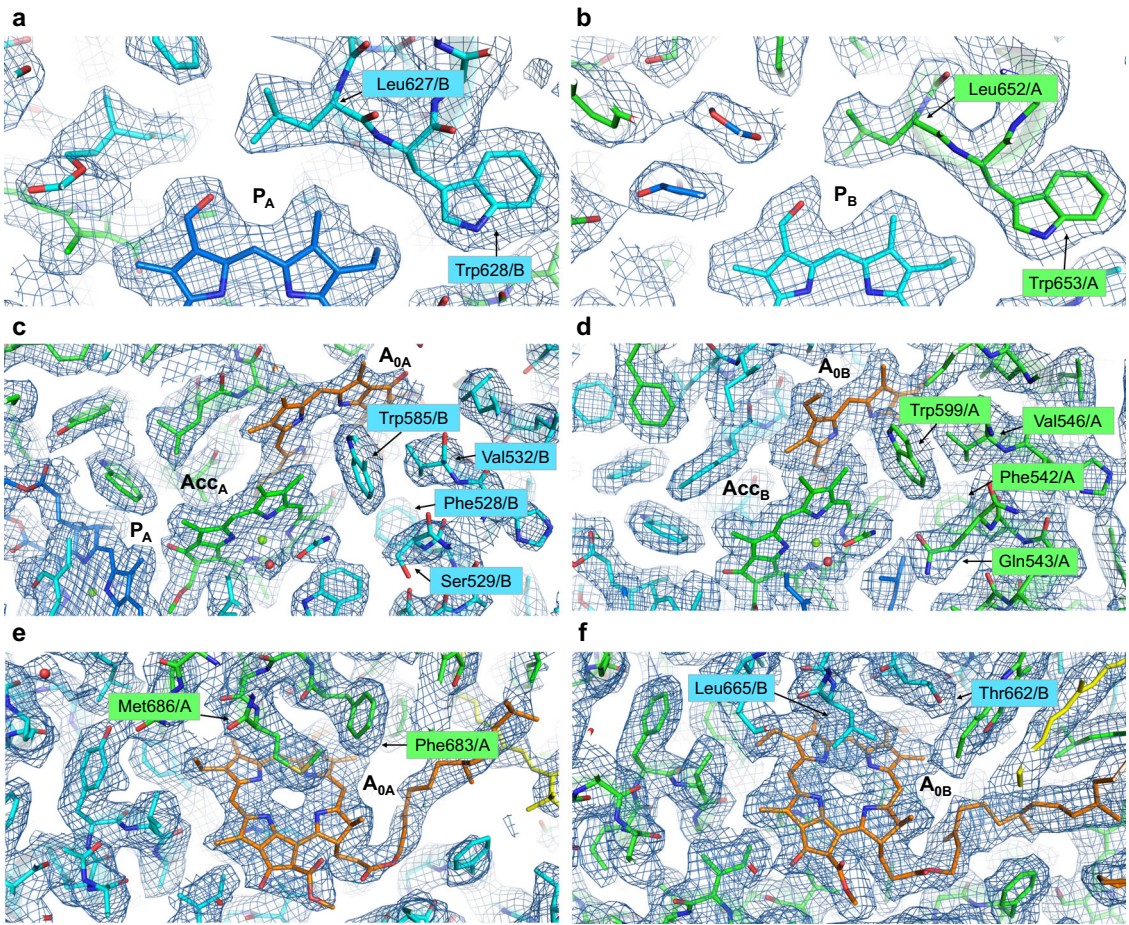

**Fig. 4 Cryo-EM density maps around P740, Acc and $A_0$.** Each map around P740 (**a**, **b**), Acc (**c**, **d**), and $A_0$ (**e**, **f**) is shown in a mesh representation at 3.0 sigma contour level for **a**, **b**, **e**, and **f**, and 5.0 sigma for **c** and **d** for ease of recognition of the water molecule coordinating the $Mg^{2+}$ of chlorin. Colors are the same as in Fig. 3.

of the electron transfer mechanism in relation to the $P_A{}^{\cdot+}/P_B{}^{\cdot+}$ ratio.

Why is Pheo *a* the primary acceptor, $A_0$, in *A. marina* PSI? According to the midpoint potential value ($E_m$), it seems reasonable for the Chl *d*-driven PSI to use Pheo *a* as $A_0$, since the energy gap is sufficient for the primary electron transfer step as is estimated below. The $E_m$ of P740 vs. the standard hydrogen electrode (SHE) is 439 mV (refs. [12,13,18,50]), which is comparable with that of P700 (470 mV, ref. [51]). While the special pair P740, which are Chl *d*/*d′*, use low-energy far-red light of 740 nm (1.68 eV), it generates reducing power almost equivalent to that of the P700 (Chl *a*/*a′*, 1.77 eV) in plants and most cyanobacteria[12]. The produced reducing power of the excited state of P740 (P740*) is weaker by 0.09 eV than that of P700* (i.e., 1.77–1.68 eV). This could result in a slower electron transfer rate to $A_0$ and an increase in reverse reaction without a change in $E_m$ of $A_0$, due to the smaller driving force. It is reported that the rates of electron transfer from P740* to $A_0$, and to PhyQ are actually comparable to those from P700* to $A_0$ and to PhyQ[41]. Then, $E_m$ of $A_0$ has to change for a proper forward reaction. Because most of the amino acid residues around $A_0$ in *A. marina* PSI are similar to those in *T. elongatus*, it is unlikely that the protein structure around $A_0$ influences the $E_m$ value. Therefore, we looked at the $E_m$ value of the cofactor molecule itself, and obtained $E_m$ (vs. SHE) values of purified Chl *a*, Chl *d*, and Pheo *a* in acetonitrile of −1100, −910, and −750 mV, respectively (Supplementary Fig. 21)[51,52]. The $E_m$ of Pheo *a* is the highest. Accordingly, Pheo *a* as $A_0$ should achieve the same electron transfer efficiency as the Chl *a*-type PSI. Then,

reinvestigation of a possible effect on rate of the following $A_0$ to PhyQ (A1) step may be warranted as the charge recombination kinetics between P740$^+$ and $A_1{}^-$ has been reported to be comparable to those between P700$^+$ and $A_1{}^-$ of Chl *a*-type PSI[50].

*A. marina* contains a limited but distinct amount of Chl *a* (refs. [12,14,43]), and we found a small amount in the PSI (1–2 Chl *a* per PSI monomer; Supplementary Table 6). It was once assumed that $A_0$ is Chl *a* (ref. [41]), but we now know this to be incorrect. One possible place for Chl *a* is as the accessory Chls, $Acc_A$, and $Acc_B$ (Figs. 1 and 5). Unfortunately, the functional group containing C3 (ref. [1]) of Acc Chl could not be precisely defined from the cryo-EM density map. While quantum mechanical/molecular mechanical calculations show that the formyl group of Chl *d* adopts two orientations (Supplementary Fig. 1c, d)[53], the oxygen atom of the formyl group on the Chl *d* molecule in a vacuum is more stable when oriented toward the C5 H atom, suggesting the conformer in Supplementary Fig. 1c. In contrast, the vinyl group of Chl *a* can adopt either orientation. Therefore, Acc could not be conclusively identified as Chl *d* or Chl *a* at the present resolution, and we assigned Accs as Chl *d* in this study. The $Mg^{2+}$ in $Acc_A$ and $Acc_B$ are coordinated by water molecules forming hydrogen bonds with Asn588/B and Asn602/A, respectively (Fig. 5b, c). In *T. elongatus* PSI, the methyl ester groups of the Chl *a* in $Acc_A$ and $Acc_B$ affect the charge and spin distributions on P700 (refs. [40,54–56]). In *A. marina* PSI, these distances were estimated to be 6.2 Å ($Acc_A$–$P_A$) and 6.6 Å ($Acc_B$–$P_B$), respectively, in the present structure.

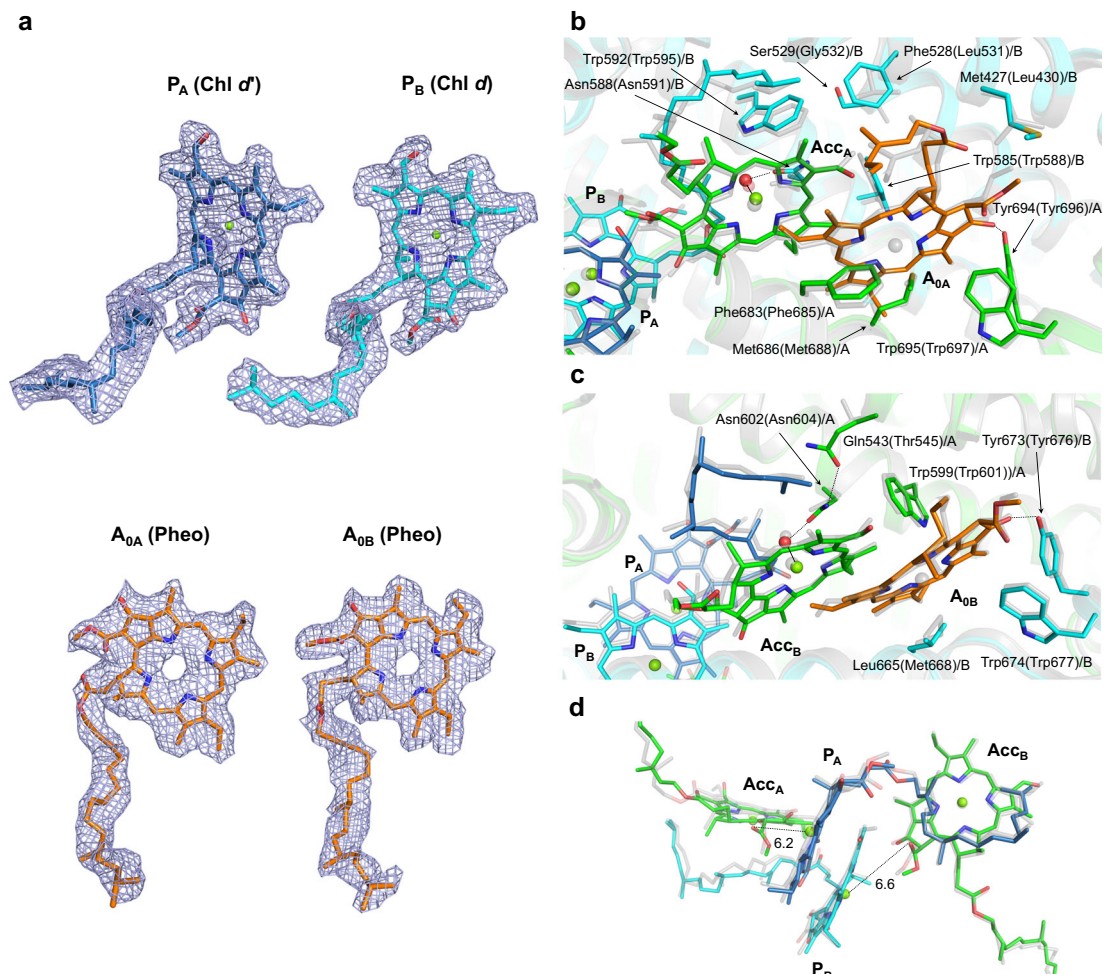

**Fig. 5 Arrangement of A$_0$, Acc, and the surrounding protein environment.** The structural model of PSI from *A. marina* is superposed with that from *T. elongatus* (PDB code 1JB0). **a** Cryo-EM density map of P$_A$, P$_B$, A$_{0A}$, and A$_{0B}$. Each map is shown in a mesh representation at 1.0 sigma contour level. **b** Arrangement of A$_0$, Acc, and the surrounding protein environment. **c** As in **b**, but view from the pseudo-C$_2$ axis. **d** Geometry of P$_A$/P$_B$ and Acc (Acc$_A$ and Acc$_B$). Chl *d*, green; Pheo, orange; water molecule, red. Colors are the same as in Fig. 3.

The amino acid residues surrounding PhyQ (A$_{1A}$ and A$_{1B}$) are switched compared with those in *T. elongatus* PSI—Met720/A and Leu665/B in *A. marina vs.* Leu722/A and Met668/B in *T. elongatus* (Supplementary Fig. 22a, b). The arrangement of the phytol chain of A$_{1B}$ is also different from that in other cyanobacteria (Supplementary Fig. 22b). These structural differences may suggest that the protein environment within *A. marina* PSI containing Chl *d* is modified to support forward electron transfer by suppressing the reverse electron transfer from PhyQ to A$_0$. However, cofactors F$_X$, F$_A$, and F$_B$ and their surrounding structures in *A. marina* PSI are nearly identical to those in other cyanobacteria (Fig. 3a and Supplementary Fig. 22). This indicates that the reduction potentials are likely at the same level, although the three Fe/S clusters, Fx, F$_A$, and F$_B$, transfer electrons from PhyQ (A$_1$) to ferredoxin and the environment does not need to be conserved as long as the overall energy trajectory is downhill.

*A. marina* PSI is a trimer as the other cyanobacterial PSIs, but the amino acid sequence identity of each protein subunit is low, and, not unexpectedly, the divergence is most pronounced around the pigments specifically observed in *A. marina* PSI, for example Chl *d*, α-Car, and Pheo *a*-A$_0$ (Supplementary Table 5). Again, the alterations must reflect optimization to drive efficient photochemistry utilizing low-energy light.

In conclusion, the structure and identification of electron carriers and key light-harvesting pigments provide a basis for understanding how low-energy far-red light is utilized by *A. marina* PSI. However, the full picture must wait for the structure of *A. marina* PSII, the system responsible for oxygen evolution through water-splitting, to be elucidated[3].

## Methods

**Cell culture and thylakoid membrane preparation.** *A. marina* was cultivated in artificial sea water Marine Art SF-1 (Osaka Yakken, Osaka, Japan) enriched with Daigo IMK (Nihon Pharmaceutical, Tokyo, Japan), using 7.5-L culture bottles (Rectangular Polycarbonate Clearboy, Nalgene, Rochester, NY) under continuous white light at 5–15 μmoL photons m$^{-2}$ s$^{-1}$ and 23 °C with continuous air-bubbling for 25–28 days. Cells were harvested by centrifugation at 7800 × *g* for 10 min, suspended in buffer A containing 50 mM 2-(N-morpholino)ethanesulfonic acid (MES)–NaOH (pH 6.5), 5 mM CaCl$_2$, 10 mM MgCl$_2$, 1.0 M betaine, and 5% (w/v) glycerol, and stored frozen until use. The cells were broken at 1.5 mg Chl *d* mL$^{-1}$ with the same volume of glass beads (0.2 mm diameter) at 0 °C by 24 cycles of 10$^{-s}$ breaking, and 2-min cooling in the presence of DNase I (0.5 μg mL$^{-1}$; Sigma, St. Louis, MO) and a protease inhibitor mixture (280 μL per 100 mL cell suspension; Sigma). Thylakoid membranes were collected by centrifuging the broken cells at 31,000 × *g* for 10 min. The resulting pellets of thylakoid membrane were resuspended in buffer A.

**Preparation of PSI trimer from *A. marina* membrane.** Prior to the purification of PSI complex, solubilization of thylakoids was tested using 13 detergents that are frequently used to solubilize membrane protein complexes: *n*-decyl-β-D-maltopyranoside (DM), *n*-undecyl-β-D-maltopyranoside (UDM), *n*-dodecyl-β-D-maltopyranoside, *n*-tetradecyl-β-D-maltopyranoside, *n*-dodecyl-α-D-maltopyranoside, *n*-octyl-β-D-glucopyranoside, 6-cyclohexyl-1-β-D-maltopyranoside, *N*-

lauroylsarcosine, n-decyl-β-D-maltopyratose neopentyl glycol (LMNG), decyl maltose neopentyl glycol, sodium cholate, and digitonin. Because the first three detergents had almost the same solubilization efficiency, UDM was selected. DM was used in sucrose density gradient ultracentrifugation because it was the best of the three detergents for maintaining the stability of the purified PSI trimer (Supplementary Fig. 5).

Thylakoid membranes were solubilized with 1.0% (w/v) UDM at 1 mg Chl $d$ mL$^{-1}$ on ice for 10 min. The extracts were separated from insoluble membranes by centrifugation at $75,600 \times g$ for 10 min, and layered onto a stepwise sucrose gradient (1.4, 1.2, 1.1, 1.05, 1.0, 0.9, 0.8, and 0.7 M sucrose in buffer A supplemented with 0.2% (w/v) DM) for ultracentrifugation at $100,000 \times g$ for 20 h. After centrifugation, fractionation was performed from bottom to top, and the protein content of each fraction was investigated by blue native-polyacrylamide gel electrophoresis using linear gradient gels of 3–12% polyacrylamide (NativePage Bis-Tris; Thermo Fisher Scientific, Waltham, MA; Supplementary Fig. 4). Fractions exhibiting a band of PSI trimer of ~720 kDa, corresponding to sucrose concentrations of 1.05–1.0 M, were collected.

For electron microscopic analysis, the PSI trimer in the above fractions was washed using Amicon centrifugal filters (pore size 100,000 Da) with buffer A supplemented with 0.002% LMNG, and applied to a second stepwise sucrose gradient ultracentrifugation consisting of 1.4, 1.2, 1.1, 1.05, 1.0, 0.9, 0.8, and 0.7 M sucrose in buffer A supplemented with 0.002% LMNG. LMNG was used for electron microscopic analysis because of its low critical micellar concentration.

For biochemical analyses, the fractions obtained after the first sucrose density gradient ultracentrifugation were applied to an anion exchange column (HiLoad 16/10 Q Sepharose HP) and eluted with a linear gradient of 0–600 mM KCl.

**Polypeptide analysis.** Polypeptides denatured with or without dithiothreitol were separated on a gel containing 0.1% (w/v) sodium dodecyl sulfate, 6 M urea, 0.6 M Tris, 0.13 M MES (pH9.0), and 16–22% (w/v) acrylamide[57], and then stained with Coomassie Blue (Supplementary Fig. 6). Separated polypeptides were identified by mass spectrometry analysis after in-gel digestion by trypsin (Supplementary Table 2).

**Spectroscopic analysis.** Chl $a$ and Chl $d$ concentrations were determined after extraction by methanol using methanol as the solvent and a UV-2700 spectrophotometer (Shimadzu; Kyoto, Japan) with a slit-width of 1 nm at 25 °C. The mass extinction coefficients for Chl $a$ and Chl $d$ was 79.95 (at 665.2 nm)[58] and 71.11 (at 697.0 nm)[59] L g$^{-1}$ cm$^{-1}$, respectively, Absorption spectra of the PSI trimers were measured at 25 °C using an UV-2700 spectrophotometer (Supplementary Fig. 6). For cell suspensions, an integrating sphere, model ISR-2600, was used with the UV-2700 spectrophotometer (Supplementary Figs. 2 and 6). Absorption spectra at 77 K were measured using an MPS 2000 spectrophotometer equipped with a low-temperature measurement unit, LTS-2000 (Shimadzu; Supplementary Fig. 7).

**Pigment analysis.** Photosynthetic pigments and quinones were extracted in methanol and quantified using reversed-phase high-performance liquid chromatography carried out, using an LC-20AT with a CBM-20A system controller (Shimadzu), equipped with a Kinetex C18 column (5 μm, 250 × 4.60 mm, 100 Å; Phenomenex, Torrance, CA), equilibrated with 80% methanol containing 20 mM ammonium acetate (solvent A). Pigments were eluted at a flow rate of 0.8 mL min$^{-1}$ over a period of 60 min with a gradient of 0–100% of ethyl acetate/methanol (30/70 v/v, solvent B)[60]; the ratio of solvent B was increased to 30% in 4 min and then to 100% over a further 36 min, and then, keeping the ratio for another 20 min. The elution pattern of the pigments was detected using a Shimadzu photodiode array detector, SPD-M20A, with Shimadzu LabSolutions (ver. 5.82) analysis software at 430 nm for photosynthetic pigments (Supplementary Fig. 19) and 270 nm for PhyQ. Standard curves for Chl $a$ and Chl $d$ were obtained after quantification of Chl $a$ from a cyanobacterium, Synechocystis sp. PCC 6803, and Chl $d$ from A. marina as described above. The standard curve for Chl $d$ was also used for Chl $d'$. A standard curve for Pheo $a$ was made using Pheo $a$ obtained by acidification of quantified Chl $a$. PhyQ (vitamin K$_1$) was obtained from Wako Pure Chemicals (Osaka, Japan). Zeaxanthin and α- and β-carotene were purchased from the VKI Water Quality Institute (Hørsholm, Denmark). PhyQ bound to PSI trimer was quantified after the removal of Chl $a$ contribution at almost the same retention time by acidification[61].

**DNA sequencing.** Genes psaA and psaB were partially sequenced as follows (Supplementary Fig. 20). Pairs of primers (Supplementary Table 7) for psaA (GTACAACTGCATCTCAATTG and CTATCCTAATGCGAGAATTC) and psaB (CCTTGCCTTCTTCTGGATGC and TTAGCCGAGAGGAGCTGTTG) were used to amplify part of the genes by PCR, using genomic DNA as the template. Then, the DNA sequence of the amplified, purified DNA fragment was analyzed using BigDye Terminator ver. 3.1 (Applied Biosystems; Foster City, CA), using the same primers.

**Cryo-EM sample preparation and data collection.** Three microliters of purified PSI (0.4 mg Chl $d$ mL$^{-1}$ in a buffer containing 50 mM MES–NaOH (pH 6.5),

5 mM CaCl$_2$, 10 mM MgCl$_2$, and 0.002% LMNG) were applied to a holey carbon grid (Quantifoil R1.2/1.3, Cu, 200 mesh grids, Microtools GmbH, Berlin, Germany) that had been pretreated by gold-sputtering[62,63] and glow-discharging (JEC-3000FC, JEOL, Japan). The grid was blotted with filter paper for 4 s, then immediately plunge-frozen in condensed ethane using an FEI Vitrobot Mark IV (Thermo Fisher Scientific, Waltham, MA, USA) under 100% humidity at 4 °C. The frozen grids were then introduced into a CRYO ARM 300 electron microscope (JEOL, Tokyo, Japan) equipped with a cold-field emission gun, after which inelastic scattered electrons were removed using an in-column type energy filter with an energy slit of 20 eV. Dose fractionated images were recorded on a K2 summit camera in counting mode with a nominal magnification of 60,000×, which corresponded to a physical pixel size of 0.823 Å. All image data were collected with a JEOL Automatic Data Acquisition System (JADAS)[64] with a dose rate of 8.57 e$^-$ Å$^{-2}$ s$^{-1}$, 10 s exposures, 0.2 s per frame, and a nominal defocus range of −0.8 to −1.8 μm. In total, we collected 3225 (data 1) and 4346 movie stacks (data 2) from two independent sample preparations.

**Cryo-EM image processing.** Cryo-EM movie stacks were grouped into ten separate optics groups based on time of data collection. Drift corrected and dose-weighted summation of movie frames was performed using MotionCor2 (version 1.1.3)[65], and contrast transfer function (CTF) parameters were estimated from the correct images with CTFFIND4 (version 4.1.10)[66]. Based on Thon-ring patterns, 4237 images from data 1 and 2 were selected for particle picking. Good PSI particles were manually selected and subjected to reference-free two-dimensional (2D) classification with RELION-3.1beta. Next, homogeneous 2D class averages were used as templates for reference-based auto-picking, and a total of 774,416 particles were extracted with a pixel size of 1.646 Å and a box size of 220 pixels for further 2D classification. Good 2D class averages contained 242,550 particles, which were passed to three-dimensional (3D) classification with a de novo initial model constructed using cryoSPARC (version 2.12.0), and a well-aligned and symmetrical 3D class was reconstructed from 86,509 particles. The particles in the best 3D class were rescaled and re-extracted with a pixel size of 1.08 Å and a box size of 330 pixels, and 3D refinement with threefold symmetry enforcement and postprocessing yielded a 3.3 Å resolution map based on the gold standard Fourier shell correlation criterion. Particles in the cryo-EM density map were then subjected to Bayesian polishing for correction of particle-based beam-induced motion and CTF refinement of particle-based defoci and optics-group-based high-order astigmatisms. Finally, 86,419 particles were selected by excluding those showing unrealistic defocus values and then subjected to 3D refinement and post-processing again, which yielded a 2.59 Å resolution map with C3 symmetrization. After beam tilt estimation and correction for every micrograph, 3D refinement and post-processing reached 2.58 and 2.97 Å resolution structures with and without C3 symmetrization, respectively. $B$-factor values for map sharpening were estimated as −92.82 and −77.95 Å$^2$, respectively. For further details, see Supplementary Figs. 9–11, and Supplementary Tables 3 and 4.

**Model building and refinement.** An initial model of the PSI monomer from A. marina was built using homology modeling in MODELLER (version 9.23)[67] by referring to the structure of the PSI monomer from T. elongatus (PDB code: 1JB0). The identity of each amino acid in subunit proteins in A. marina PSI was determined using amino acid sequences obtained from UniProt (https://www.uniprot.org/). The resulting homology model of the PSI monomer was fitted to the whole cryo-EM density map (C3 map), using a "fit in map" program in UCSF Chimera (version 1.13)[68], and an initial model of the trimeric form of PSI from A. marina was created. Next, the created PSI trimer model was fitted to the cryo-EM density map using a molecular dynamics simulation program (CryoFit) and simulated annealing in Phenix[69,70]. Each model of the three monomers in the PSI trimer was refined separately, and the refined model was modified using COOT[71] manually to fit the cryo-EM density map. Subsequently, the obtained structure model was finally refined using refmac5 (ref. [72]; Supplementary Fig. 12). Refinement statistics of the refined model were obtained using a comprehensive validation program in Phenix.

The cofactors such as Chl $d$, Chl $d'$, Chl $a$, Pheo $a$, PhyQ, iron–sulfur clusters, α-Car, MGDG, and PG were identified and refined using geometry restraint information, as described in the following section. For A$_0$ (A$_{0A}$ and A$_{0B}$) in A. marina PSI, a cryo-EM density of Mg$^{2+}$ of Chl could not be identified unambiguously, and thus the two A$_0$ were identified as Pheo. Validation between the refined model and the cryo-EM density local resolution map was assessed using Phenix, ModelZ[73], and MapQ[74]. The refinement statistics of the models are summarized in Supplementary Table 1. Structure figures were generated and rendered with PyMOL and Chimera[68,75].

**Identification of cofactors and their geometry restraint information.** The restraints needed for the pigments in PSI from A. marina were generated by Electronic Ligand Bond Builder and Optimization Workbench (eLBOW)[76] because the information was not registered in the restraints library in Phenix. The restraint information for Chl $d$ was created from the model of Chl $d$ identified in a high-resolution structure (PDB code: 2X1Z)[77], and information for Chl $d'$, an epimer of Chl $d$, was created with reference to the structures of Chl $d$ and Chl $a'$, an epimer of

Chl $a$ (PDB codes: 1JBO and 2X1Z). The orientation of the C3-formyl group of Chl $d'$ was determined based on that of Chl $d$ (ref. [77]). Only a small amount of Chl $a$ was present in the sample, and so we assigned the Chls of *A. marina* PSI as Chl $d$ in this study even though it is not possible to distinguish between Chl $d$ and Chl $a$ at 2.58 Å resolution.

The PSI trimer from *A. marina* contained α-Car instead of β-carotene (β-Car). The α-Car molecule possesses two rings—known as β- and ε-rings—at each end, while the β-Car molecule possesses two β-rings. Although the restraint information for α-Car was created using eLBOW, as for the other cofactors, α-Car was registered as an unknown ligand (three-letter code: UNL) in this study because it is hard to distinguish between β- and ε-rings from the present cryo-EM density map. Restraint information for Pheo, iron–sulfur clusters, MGDG (three-letter code: LMG), and PG (three-letter code: LHG) was created with reference to high-resolution structures (PDB codes: 1JB0 and 3WU2).

**Identification of water molecules**. Water molecules were identified manually from the cryo-EM density map at >2.0 root-mean-square level using COOT. Eighty-four water molecules were found in the refined structure model. Most of the water molecules were identified as ligands to the $Mg^{2+}$ in Chl $d$.

**Reporting summary**. Further information on research design is available in the Nature Research Reporting Summary linked to this article.

## Data availability
Atomic coordinates and cryo-EM maps for the reported structure of *Acaryochloris marina* PSI have been deposited in the Protein Data Bank under accession codes 7COY, and in the Electron Microscopy Data Bank under accession codes EMD-30420, respectively. Other data are available from the corresponding authors upon reasonable request. Source data are provided with this paper.

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

## Acknowledgements

K.K. is grateful to Prof. Nobuo Kamiya (Osaka City University) for providing laboratory access. K.I. would like to thank Mr. Yuzo Watanabe (Proteomics Facility, Graduate School of Biostudies, Kyoto University) for his help with mass spectrometric analysis. We are grateful to Ms. Yoko Kunishima and Ms. Kyoko Ishino for their technical assistance, to Prof. Tatsuya Tomo (Faculty of Science Division I, Tokyo University of Science) for providing spectral data of *Halomicronema hongdechloris*. We thank James Allen, DPhil, and Jeremy Kamen, M.Sc. Biology, from the Edanz Group (https://en-author-services.edanzgroup.com/), and Dr. David Mcintosh for editing this manuscript. This work was supported by grants-in-aid from the Japan Society for the Promotion of Science (Grant 18H05175 to Y.K.; 16H04757 to K.Y.; 16H06554 to K.I.; and 20K06684 to S.I., 20K06515 to T.H.; and 20H05109 and 20K06528 to K.K.), the RIKEN Pioneering Project, Dynamic Structural Biology (to T.H. and K.Y.), the Cyclic Innovation for Clinical Empowerment (CiCLE) from the Japan Agency for Medical Research and Development, AMED (to T.H. and K.Y.), Innovations for Light-Energy Conversion (I⁴LEC; to Y.K.), and JST-Mirai Program Grant Number JPMJMI20G5 (to K.Y.).

## Author contributions

K.S.-I., N.I.-K., E.Y., and K.M. optimized purification conditions. K.S.-I. purified the PSI trimer used in cryo-EM experiments. K.S.-I., N.I.-K., S.I., K.I., and Y.K. performed biochemical analyses. T.H. and K.Y. collected and analyzed cryo-EM data. K.K. built and refined the structural model. T.H., K.K., K.S.-I., K.Y., and Y.K. wrote the manuscript with input from all other authors.

## Competing interests

The authors declare no competing interests.
