## [Peer Review File · Nature Communications]

Reviewers' comments:

Reviewer #1 (Remarks to the Author):

The authors are reporting here the structure of photosystem I from the cyanobacterium *A. marina* obtained by cryo-EM at a resolution of 2.48Å. This complex contains mainly Chl d, a pigment absorbing in the far-red region of the solar spectrum and it is thus different from all other PSI analyzed so far. The structure is certainly useful for better understanding the flexibility of PSI complexes and its capacity to use efficiently different light colors. The topic is also hot because of the interest in far-red absorption generated by the discovery of far-red acclimation in cyanobacteria (this is not discussed in the manuscript, but it should). However, the present version of the manuscript needs to be improved. The discussion of the data is not always clear and the authors are making several general statements that are not supported by data neither explained. They have a nice structure, but they should avoid trying to reach conclusions about the spectroscopic properties of the pigments based on the structure, it is not possible, at least not with the tools used in this work.

The writing needs to be improved.

Detailed comments:

The result of the pigment analysis on the preparation does not seem to match with the pigments observed in the structure. This should be explained. Is it possible that some of the pigments are lost during purification? Please discuss it.

In this structure there are far less Chls than in Chl a-containing PSI complexes and even less than in the recently resolved structures of Chl f-containing PSI complexes. It would be interesting to compare all the structures and explain which Chls are missing.

Line 54-55 "With the acquisition of Chl d, *A. marina*, uniquely harnesses far-red light, which is the lowest energy light that can be used in all oxygenic photosynthetic organisms". This is not correct, Chl f-containing cyanobacteria can use light at longer wavelengths.

Line 56-58 (similar sentence also at line 274) "The cyanobacterium can be seen as a missing-link between organisms possessing visible light/Chl a-based oxygenic photosynthesis and 800–900 nm light/BChl-based anoxygenic photosynthesis.". This statement is unclear, please explain what you mean exactly.

Line 119 "The Chls and α -Car pigments are mainly involved in the electron transfer reactions and light-harvesting." It seems that this statement is based on the structure. On what else can the Chls be involved? Please explain.

Line 119 "These configurations are similar"..." how similar? Please be more precise.

Line 163-164. The authors state : "Details of the electron transfer mechanism in relation to the $PA^+ + /PB^+ +$ ratio can now be filled in using the precise structure of *A. marina* PSI." what does it mean? This part seems to require some calculations that are not presented.

Line 165-180 The discussion of the electron transport chain is confusing, the authors should try to rationalize their findings better and present them in a more structured way.

184-185. The small amount of Chl a is not mentioned at the beginning when the authors report the number of pigments present in the complex.

199-203. Are the differences between the distances significant? The authors should discuss them considering the uncertainty in the structure determination.

207-208 "These structural differences may suggest that the protein environment within *A. marina* PSI is modified to optimize electron transfer involving Chl d". This is a very generic statement, the authors should at least formulate some hypotheses.

Line 215- "α-Car harvest low energy far-red light". Carotenoids do not harvest far-red light

Line 220 - "... lack of some Chls, which, we assume, reflects optimization for efficient use of far-red light". How can the lack of some Chls reflect optimization for efficient use of far-red light? Please explain.

From line 222 - The discussion of the possible location of the red forms is not correct. The authors seem to suggest that the location in other PSI complexes is known and that the shift is due to interactions with the protein, this is not correct.

Line 231 "In *T. elongatus* PSI, 9–11 Chl a per PSI monomer are considered to be red Chls". The reference is missing. The numbers seem very large, the authors should carefully check the literature.

Line 234-235 - "In *A. marina*, this latter region seems highly flexible, and Chl d 55, 56, 63–67, and 77–79 could not be assigned as counterparts of Chl a molecules". What does it mean? That there are no Chls in these positions?

Line 237-238 - "the *A. marina* PSI intrinsically does not exhibit features of a long-wavelength Chl d (Extended Data Figs. 2, 6, and 7)". This statement does not seem to be correct or at least needs to be explained. The absorption spectrum peaks around 705 nm and there is a shoulder around 730 nm; these are red-shifted Chls in my opinion.

From 239 - the discussion about the absorption properties of the pigments based on differences in the residues located nearby is not sound and not informative. The complex analyzed here contains Chl d, it has far less Chls than the Chl-a- PSI and we do not even know where each Chl of the Chl-a containing PSI absorb. To say that a specific Chl can absorb differently than in other PSI is noninformative. I suggest to avoid these speculations; they do not add anything to the work.

Line 247 "The *A. marina* PSI appropriately lacks the Chl with most increased site-energy that captures high-energy, short-wavelength light in *T. elongatus*". On what is this conclusion based?

Line 255 "These differences help explain the unique features of light harvesting in *A. marina*." The unique features of *A. marina* is due to the presence of Chl d in most binding sites. The substitution of one specific amino acid is certainly not responsible for it.

Line 268 "Again, these divergences must affect energy transfer efficiencies in the two organisms." Again, this statement is noninformative for the reasons I explained above.

Reviewer #2 (Remarks to the Author):

Acaryochloris marina, a cyanobacterium capable of using the energy of far-red light to drive oxygenic photosynthesis, contains a photosystem I (PSI) with Chl d as the major pigment. Chl d is a special type of chlorophyll molecule absorbing at a wavelength range (700-750 nm) longer than Chl a. The structures of photosystem I with Chl a as the major chlorophyll have been solved (such as the ones from *T. elongatus*, etc), but the structure of PSI from *A. marina* remains to be analyzed at high resolution so that the exact locations of Chl d molecules can be assigned clearly. Hamaguchi T. et al. report the cryo-EM structure of PSI from *A. marina* at 2.5 Å resolution and reveal the detailed arrangement of various cofactor molecules within the complex. The assignment of Chl d molecules in the P740 special pair is consistent with previous spectroscopic studies. Unexpectedly but interestingly, a pair of Pheo a molecules were found to take the positions of primary electron acceptor, a new feature not found for any PSI before. The following are a few questions and suggestions for the author to consider during revision.

Major points:

(1) While the previous spectroscopic study suggested the P740 reaction center of PSI from *A. marina* is likely a heterodimer of Chl d, the slight difference between Chl d and Chl a (O atom vs CH₂ group on the C3₂ position) is too little to be detectable in the 2.5-Å resolution cryo-EM map. Will the authors observe any Fo-Fc (or real space analog of Fo-Fc) difference map densities around the C3(2)-groups if the reaction center is modeled as Chl a? The difference map can also be utilized in searching for potential water molecules around the C3-formyl groups in Chl d and verifying the assignment of Pheo a on the A0 sites.

(2) As an indirect evidence supporting the assignment of Chl d (instead of Chl a) in the cryo-EM map, are there hydrogen-bond donors nearby the C3-formyl groups of the putative Chl d molecules in the electron transfer chains (PA, PB, AccA and AccB)? Alternatively, is the environment around the C3-formyl groups of Chl d more polar than the corresponding one of Chl a in *T. elongatus* PSI?

(3) In lines 184-195 on p8, the authors initially suggested that AccA and AccB may be assigned as Chl a, but then concluded that "The assignment of Chl d to the Accs in this study is actually due to the difference spectrum of P740, ...". Could the authors estimate or discuss the relative efficiency of Acc(Chl a), compared to Acc(Chl d), as a intermediate cofactor mediating electron transfer from P740 to A0(Pheo a)?

(4) Zeaxanthin is a carotenoid with important photoprotective function (Niyogi, K. et al. PNAS, 94: 14162-14167, 1997). The HPLC analysis result indicates that there is one zeaxanthin molecule in the PSI from *A. marina*. Is this a structural component of PSI or contamination by other complexes? If it is a structural component of PSI, where is it located?

Minor points:

(5) It will be great if the cryo-EM densities of AccA and AccB can be included in Fig. 4A.

(6) The statement of the surrounding structures of FX, FA, FB must be invariant for the reduction of

NADP+ (lines 211-212, p9) is too strong. It might be better to use “conserved” or “highly similar” instead of “invariant”.

Reviewer #3 (Remarks to the Author):

The manuscript of Hamaguchi et al. describes the structure of PSI from the unusual cyanobacterium *Acarychloris marina*, which uses mostly Chl *d* to perform oxygenic photosynthesis. The authors claim the structure has an overall resolution of 2.48 Å, although this is very difficult to believe given missing subunits, disordered segments of multiple polypeptides, etc. Perhaps the most interesting claim, if verified, is that the A₀ acceptor is pheophytin *a*, something that surprisingly had not been revealed in previous spectroscopic studies. It is not clear that this manuscript is in the correct format, as it seems to have been written in a style more like that used by other journals. As a result, many points are not very fully described and important details are lacking. Comments for the authors follow.

1. Title. While it is true that this PSI complex absorbs far-red light, that is not what makes it unusual. It is the large Chl *d* content that is unusual, and that should be mentioned in the title.
2. Abstract, line 7. It is fantasy for the authors to claim that this cyanobacterium “uniquely” can use far-red light for photosynthesis. Dozens of cyanobacteria have been identified that can synthesize Chl *f* and Chl *d*, and these organisms use light wavelengths extending to 800 nm to perform oxygenic photosynthesis.
3. Abstract, line 11. It is questionable whether P740 is the primary donor or a secondary donor. In other PSI complexes, the Acc_(A-1) Chl is now thought to be the primary donor and the P700 dimer is thought to reduce the cation produced by transfer of an electron from A₋₁ to A₀.
4. Page 2, lines 20-23. Chls absorb light in the blue and the red, not only in the red. BChls also have absorbance in the blue/ultraviolet region of the spectrum.
5. Page 2, line 18. Chl *d* is not “novel” and is not exclusively found in *A. marina*. It is also found in cyanobacteria that synthesize Chl *f* and perform far-red light photoacclimation.
6. Page 3, lines 6-7. This statement is simply false. Chl *d* is not uniquely found in *A. marina*, as noted above, and it is not the lowest-energy absorbing pigment found in cyanobacteria. That would be Chl *f*, which absorbs productively to 800 nm.
7. Page 4, line 4. The heliobacterial RC contains both BChl *g* and 8¹-OH-chlorophyll *a*. Two cyanobacterial PSI structures have been determined for complexes containing Chl *f*: Kato et al. and Gisriel et al. 2020a, 2020b.
8. Page 4, lines 4-6. No matter how many times the authors repeat this statement, it is not correct that *A. marina* “uniquely” utilizes far-red light for oxygenic photosynthesis. This must be corrected.
9. Page 4, lines 5-7. Three papers describing PSI complexes that can use far-red light have already appeared: Kato et al., 2020; Gisriel et al., 2020a, 2020b. These structures were also determined to understand how PSI can use far-red light. This study is not unique and the other relevant studies should be appropriately cited and contrasted.

10. Page 5, lines 4-7. If Psa27 and PsaI are essentially the same thing—and functionally they are even if the sequence homology is low—why give the protein another name (which in any event would not be valid by Demerec rules?).
11. Page 5, lines 10-18. Given missing subunits, disordered loops, and unassigned amino acids due to resolution issues, it is very difficult to imagine that the stated resolution for the structure could be 2.48 Å, slightly better than that of Jordan et al. for the X-ray structure of PSI from *Thermosynechococcus elongatus*. That structure had more than 200 assigned water molecules, and this one only 84, another indication that the overall

- resolution is unlikely to be so high as the authors are claiming. Furthermore, the authors state that they find 71 Chl *d* molecules, 1 Chl *d'*, 2 pheophytins (should be pheophytin *a*), and only 11 alpha-carotene molecules (the elution profile shown in the supplement indicates the presence of zeaxanthin and probably either echinenone or cryptoxanthin as well). *T. elongatus* PSI was originally assigned 96 Chl *a* (now thought to be only 95) and 22 beta-carotenes, which again suggests that large parts of the structure have rather poor resolution. Given the similarity of the protein sequences for PsaA and PsaB, it is hard to imagine that there should be 23 fewer Chls in the complex. It would be helpful for the authors to produce a clear statement about what is expected—how many Chl *a*, pheophytin *a*, and Chl *d* molecules are expected to be present in the complexes based on biochemical analysis—and how many are found and just as importantly which Chl *a* molecules of *T. elongatus* PSI are NOT found. Same for carotenoids.
12. Page 5, lines 21-23. Considering that the authors do not have sufficient resolution to identify the Chls at the Acc/A₋₁ position and may not have sufficient resolution according to the text in the methods to assuredly specify that the A₀ tetrapyrrole is Pheo *a*, this statement is not correct. They may have assigned four pigments as 3 Chl *d* and 1 Chl *d'*, but the authors don't know that this is correct.
 13. Page 6, line 23. "there was no mass." One does not detect "mass" in an ESP map produced by cryo-EM. In the methods (page 24, lines 24-25) it is stated that Mg could not be "unambiguously" identified, so by default these positions were defined as Pheo *a*. It is not clear whether they could also be Pheo *d*. Considering that the Mg was not unambiguously shown to be missing, it must be that it is also possible that it is present and that these tetrapyrroles are either Chl *d* or Chl *a*. Perhaps the authors are not stating clearly what they mean, but it is not going to be clear to readers at what degree of certainty these pigments have been identified in the structure. Finally, it seems that there should be at least two Chl *a* molecules somewhere in the structure, and authors do not specify where these might be located, although they mention that it could be possible that they are present in the electron transport chain, presumably at the Acc/A₋₁ positions.
 14. Page 7, line 12-13. The authors can speculate about this, but the details of the ETC components do not seem to be fully delineated by this study. The identities of the Acc/A₋₁ pigments and even the A₀ positions are still ambiguous (see #13 above).
 15. Page 7, line 15. Pheophytin *a* is not found in all type-2 reaction centers, because the bacterial reaction centers of proteobacteria and Chloroflexi, which do not synthesize Chl *a*—only BChl *a*. Thus, they contain Bpheo *a*.
 16. Page 8, lines 1-2. Of course the protein could have strong effects on electron transfer by electrostatic effects that do not have to be directly adjacent to the cofactor.
 17. Page 8, lines 6-7. Statement requires a reference.
 18. Page 8, line 2. As noted above, the authors should provide a more detailed analysis of the pigment content of these reaction centers. If there are two Pheo *a*, then it should be possible to quantify Chl *a* better than "1 or 2." Of course, it is still unclear exactly how many Pheo *a* molecules there are. The authors do not state that there are no other Pheo *a* molecules in the complex. The authors make no comments about whether there are H-bonding residues in the vicinity of the C-3 substituent on the Acc/A₋₁ tetrapyrroles. This might help to resolve which pigment is present in that position.
 19. Page 8, lines 17-19. The electrochromic shift argument is a very weak one. In a complex with 90 pigments, there can be small distributed effects or stronger local effects, and

these would be nearly impossible to distinguish. Electronic effects can occur over considerable distances.

20. Page 9, lines 1-2. Perhaps a more relevant question is the distances between Acc/A₋₁ and A₀, given that this pair of Chls appears to be the primary donor in most PSI complexes, not P700. In *A. marina*, if these Chls are Chl *a*, then charge separation would occur between Chl *a* and Pheo *a* and the Chl *d* molecules would mostly serve as a long-wavelength antenna (other than the special pair as possible secondary donor).
21. Page 10, lines 8-10. This is another example where the authors provide ambiguous information. Are the three Chls of the trimer missing in *A. marina* PSI or are they just not assigned as Chl *a*? The assumption is that they are not present, but this is really not clear.
22. Page 11, lines 19-20. Another example. Every pigment in this complex is different from one in *T. elongatus*, so the statement about amino acid changes in proximity to different pigments being different means nothing. Virtually everything should be different. This does not describe how it might be different or if there are patterns of differences—like H-bonding residues for the formyl groups, etc.
23. Page 11, line 21. Considering that some subunits are missing and apparently partially missing from the PSI complexes, making comments about flexibility is meaningless. This could all be due to damage introduced during the isolation. It is unlike to be the case in fully inact PSI complexes for the reason stated—it would be unproductive for energy transfer.
24. Page 11, lines 24-25. It is extremely unlike that this complex, which requires Chl *d* that requires oxygen for its synthesis and evidently arose long after Chl *a*, arose divergently to impact the evolutionary trajectory of the bacterial reaction center (especially homodimeric ones). Any similarities to those other reaction centers almost certainly arose by convergent evolution, which might be informative but not in the way the authors seem to intend. However, because this is just a toss-off statement with no detailed explanation, it is hard to know just what they mean.
25. Page 12, line 3. Sadly, the full picture for the PSI complex of *A. marina* must also await further studies and a thorough rewriting of this manuscript.
26. Page 25, line 25. This statement seems surprising, considering that most Chls in *T. elongatus* PSI are not liganded by water. Do the authors mean that some were H-bonded to the formyl group? How many H-bonded formyl groups were detected? This is extremely important in defining site differences for binding Chl *b* versus Chl *a* in LHC2 and LHC1, and is also important in binding Chl *f*. See Gisriel et al. 2020b.
27. Figure 4 A. The formyl group of P_A appears to fit the density better in the other orientation. With the current resolution how did the authors determine that the Pheo molecules are Pheo *a* and not Pheo *d*? There is virtually no way to distinguish the difference between a vinyl group and a formyl group in an ESP map at this resolution.
28. Extended data Figure 3. Although authors frequently discuss the far-red absorbing properties of Chl *d*, the spectra here show another important aspect: Chl *d* has much enhanced absorbance in the blue compared to Chl *a*. This could also be important in light harvesting.
29. Legend, extended data Figure 5. ...were subjected to analysis by SDS-PAGE electrophoresis...

30. Extended data Figure 14. The orientation of the formyl group in P_A seems to be flipped as noted above. The Acc Chl could be Chl *a* or Chl *d* based on the ESP map. Car4003 appears to be *cis*-carotenoid.
31. Extended data Figure 15. The “unidentified” (not non-determined) carotenoid is likely to be either cryptoxanthin or echinenone. If there is no carotenoid ketolase in the genome, then the most likely carotenoid would be cryptoxanthin, an intermediate in the synthesis of zeaxanthin. There appear to be *cis*-isomers of alpha-carotene present, which is noted in #29 above.

Reviewer #4 (Remarks to the Author):

The manuscript “Structure of the far-red light utilizing system I of *Acaryochloris*” by Hamaguchi et al. resolved the structure of Chl d-PSI isolated from *Acaryochloris* using cryo-EM technology. The highlight of this manuscript is that Pheo *a* is found in PSI and functions as the primary electron acceptor. However, there are detectable Chl *a* from the purified PSI samples, no assigned position/location and possible roles for them and authors gave very speculative explanation without any supporting data (e.g. statement at page 8, lines 184 – 195), which is not acceptable.

The total numbers of assigned chlorophylls (including 2 Pheo *a*) are much lower than the chlorophyll numbers in typical purified PSI from Chl *a*-cyanobacteria, and it is also much lower than the previous report about purified Chl d-PSI from the same organism (Tomo et al, 2008, JBC), in which there were 97 Chl *d*, 2 Chl *a* and 25 alpha-carotene.

There was confused information about the numbers of chlorophylls.

1. Look at the details of numbers of assigned chlorophylls, there are 73 assigned chlorophylls (including Chl *d*' and Pheo *a*) in Extended Data Fig 19, but there are 74 assigned chlorophylls in Extended data table 6.
2. Look at the pigment analysis HPLC data and the summary table (Extended Data Table 7), per PSI sample gave a total of 70 chlorophylls (67 Chl *d* + 1 Chl *a* and 2 Pheo *a*), even more less than the assigned chlorophylls in the structure.
3. Extended data table 6, Chl 5 and 6 should read as pheo *a*, and it said “ assigned Chl *d* are numbered according to the ...” which is not correct statement here because Chl 5 and Chl 6 are pheo *a*, not Chl *d*.
4. In the same table, N/A (not assigned) caused confusion. Authors stated that “Not assigned” is due to the low resolution, which means those chlorophylls might be there, but cannot be detected/assigned here due to the limit of the resolution (?). However, authors confirmed the Chl d-PSI contains less Chl reflecting the optimization for efficient use of far-red light. i.e. these N/A chlorophylls represent “not detected” instead of “not assigned”.

There was confused information about assignment for Chl *d* and Chl *d*'.

Extend data Fig 1 gives explanation the difference between Chl *d* and Chl *d*'. In Chl *d*, the oxygen atom of formyl group is toward C5 side.

However, the 3D model of Chl *d* and Chl *d*' (Fig 4A) demonstrated the same orientation of formyl group. The same 3D maps showed in extended data Fig 14.

Which 3D model map was correct for Chl *d*'?

The inconsistent data:

Extend data table 6 demonstrate N/A number 86 chlorophyll and an assigned the number 88 Chl, but in the Extended data Fig 19, number 88 was shown as “transparent”. Which one is missed from Chl d-PSI?

About presence of Chl *a*, where are they? Are they from uncoupled contamination in the sample?

There were undefined peptides in the region of 25 -45 kDa (in Extend data Fig 5, lane 2 and 3), the contamination of PSII or other chlorophyll-binding protein complexes?

Authors should have the mass data for these undefined peptides to exclude the possible sample contaminations.

Pheo a in the purified PSI seem identified unambiguously. However, detected Chl a, and also much less numbers of assigned Chl d raised question about the quality of isolated PSI? are they intact PSI complexes? Which analysis authors used to confirm that no contamination in the "purified PSI" samples?

Responces to Reviewers

Reviewer #1 (Remarks to the Author):

C1: The authors are reporting here the structure of photosystem I from the cyanobacterium *A. marina* obtained by cryo-EM at a resolution of 2.48Å. This complex contains mainly Chl d, a pigment absorbing in the far-red region of the solar spectrum and it is thus different from all other PSI analyzed so far. The structure is certainly useful for better understanding the flexibility of PSI complexes and its capacity to use efficiently different light colors. The topic is also hot because of the interest in far-red absorption generated by the discovery of far-red acclimation in cyanobacteria (this is not discussed in the manuscript, but it should). However, the present version of the manuscript needs to be improved. The discussion of the data is not always clear and the authors are making several general statements that are not supported by data neither explained. They have a nice structure, but they should avoid trying to reach conclusions about the spectroscopic properties of the pigments based on the structure, it is not possible, at least not with the tools used in this work.

The writing needs to be improved.

A1: Thank you for your evaluation and valuable advice to improve our manuscript. We revised the manuscript to improve the clarity through proof-editing by a professional English editing service. Then, we added description on the acclimation of cyanobacteria to the far-red light in the introduction section (6 lines from last line in page 2, and the second paragraph in page 3). Further, according to the advice by the reviewer, we removed the statement on the spectroscopic properties without referring appropriate references in the section of "Antenna and Red Chls". Here are responses to your comments.

C2: The result of the pigment analysis on the preparation does not seem to match with the pigments observed in the structure. This should be explained. Is it possible that some of the pigments are lost during purification? Please discuss it.

A2: As is pointed out by Reviewer #4 (comment #8), the preparation used for biochemical pigment analysis contains limited amount of PSII, which will slightly increase the amount of pigments per monomer. However, considering the small values of standard errors among five independent preparation, the contribution of contaminating PSII to the amount of pigments should be quite small. Here, cryo-EM structure was obtained through classification by excluding limited amount of contamination such as PSII. Mismatch of pigment values between biochemical and structural analyses is frequently found in the literature. For example, in the article reported the PSI structure of Chl *f*-carrying cyanobacterium, *Fischerella thermalis*, the numbers of Chl species determined by structure and biochemical analyses were 89 and 96 (7.1 ± 0.3 Chl *f* and 89.1 ± 14.1

Responces to Reviewers

Chl *a*) molecules per monomer, respectively (Gisriel et al. *Sci Adv* 2020). We can discuss the mismatch of pigment numbers between structural and biochemical analyses in our study by referring the values in the report by Gisriel et al. (2020), but we think it is not appropriate. Instead, related description was added in the revised manuscript (second paragraph in page 7, see below A3).

C3: In this structure here are far less Chls than in Chl *a*-containing PSI complexes and even less than in the recently resolved structures of Chl *f*-containing PSI complexes. It would be interesting to compare all the structures and explain which Chls are missing.

A3: As is pointed out by Reviewers #3 (comment 11) and #4 (comment 2), there is a possibility that some pigments located in the outer part of PSI could be released during the sample preparation. However, considering the standard errors among independent 5 preparations (Extended Data Table 7) that is smaller than the previous report (Tomo et al. 2008), there is still possibility that the number of pigments could be actually smaller than other type PSI. One paragraph describing such situation and unassigned Chls compared with *T. elongatus* PSI was added as follows in the revised manuscript (second paragraph in page7).

Although the amount of Chl *d* per Chl *d'* determined by pigment analysis (67.0 ± 0.66 , $n=5$ of independently prepared PSI) and assigned by structural analysis (71 including one or two Chl *a*) in this study is lower than those in previous studies (145 ± 8^{14} and 97.0 ± 11.0^{12}), the semi-stoichiometric amount of Phe *a* (1.92 ± 0.022) that was 0.3 ± 0.2 per reaction center in the previous study¹² is consistent with the structural analysis in this study. Although local resolution of some parts of outer region is somewhat lower, our PSI was stable to keep its integrity for five days (Extended Data Fig. 5). Therefore, we deduce here that the number of Chl associated with PSI is smaller than in other oxygenic phototrophs. **Actually, Chls corresponding to Chl-88 and Chl-94 in *T. elongatus* are surely absent in *A. marina* PSI. Chls that were unable to be assigned in *A. marina* PSI compared with those in *T. elongatus* PSI are shown by marking N/D in the column of ID number in Extended Data Table 6, and those Chls can be recognized by chlorins in transparent gray in Extended Data Fig. 17.** The numbers of Chls in PSI of Chl *f*-carrying *Fischerella thermalis* and *H. hongdechloris* are also smaller than that of *T. elongatus* PSI; 89 (*F. thermalis*)³⁰ and 90 (*H. hongdechloris*)¹⁹ Chls vs 96 Chls in *T. elongatus* PSI. Type I reaction center of *Heliobacterium modesticaldum* carries much smaller amount of Chl species; 60 molecules per reaction center²⁹.

Tomo, T. *et al.* Characterization of highly purified photosystem I complexes from the chlorophyll *d*-dominated cyanobacterium *Acaryochloris marina* MBIC 11017. *J. Biol. Chem.* **283**, 18198–18209 (2008).

C4: Line 54-55 “With the acquisition of Chl *d*, *A. marina*, uniquely harnesses far-red light, which is the lowest energy light that can be used in all oxygenic photosynthetic organisms”. This is not correct, Chl *f*-containing cyanobacteria can use light at longer wavelengths.

Responces to Reviewers

A4: The reviewer is correct. Here, we intend to claim that the light energy in the far-red light region is directly used for photochemical reaction. In Chl *f*-carrying cyanobacteria, Chl *f* is accumulated only under far-red light condition up to ~10% of total Chls. The far-red light energy captured by Chl *f* would be used for photochemical reaction after quantum-theoretical uphill transfer with environmental heat to the reaction center. Further, the contribution of far-red light absorption to the total absorption is larger in *A. marina* than Chl *f*-carrying cyanobacteria (see Extended Data Fig. 2, in which absorption spectrum Chl *f*-carrying *Halomicronema hongdechloris* cells was added). Thus, we modified the sentence as follows:

With the acquisition of Chl *d*, *A. marina* harnesses far-red light, which is the lowest energy light that can be used for the oxygenic photosynthetic organisms.

Additionally, description on light utilization in Chl *f*-using photosystem was added in the last sentence in page 2 and the second paragraph in page 4.

<the last sentence in page 2>

Later, several cyanobacteria were found to induce limited amount of Chl *f* for capturing far-red light only under far-red light condition⁵⁻⁷ (Extended Data Fig. 2, and see below).

<the second paragraph in page 4>

The far-red light-absorbing Chl, Chl *f*, is also utilized in some cyanobacteria to capture far-red light (Extended Data Fig. 2). In combination with Chl *d*-carrying cyanobacteria, acclimation strategy to the light quality expanding to far-red light region gathers attention⁵⁻⁷. Chl *f* that is only induced under far-red light condition occupies less than ~10% of total Chls in photosystem I (seven Chl *f* and 83 Chl *a* in *Halomicronema hongdechloris* PSI¹⁹) and plays solely a light-harvesting function⁷ without engaging photochemical reaction nor electron transfer chain, which is contrasting to Chl *d* that takes place of special pair responsible to photochemical reaction. Low light energy of far-red light captured by Chl *f* is transferred to Chl *a* by quantum-theoretical uphill process using environmental heat for photochemical reaction²⁰. Therefore, the photochemical reaction of Chl *f*-carrying PSI is definitely the same as other PSI operated by Chl *a*; the photoreaction occurs using “visible red light’ by P700. However, Chl *d*-driven photoreaction occurs by low-energy far-red light corresponding to Qy band of Chl *d*. Chl *d* also absorbs high-energy blue light as well corresponding to Soret band, but the resulting excited state of Chl relaxes to lower excited state corresponding to Qy level for photochemical reaction. In this sense, Chl *d*-driven system is unique in the photochemical reaction occurring by far-red light level energy.

C5: Line56-58 (similar sentence also at line 274) “The cyanobacterium can be seen as a missing-link between organisms possessing visible light/Chl *a*-based oxygenic photosynthesis and 800–900 nm light/BChl-based anoxygenic photosynthesis.”. This statement is unclear, please explain what you mean exactly.

A5: This statement was intended to describe the evolutionary position of this organisms as is described in Hu et al (PNAS 1998). However, the description is not essential in this manuscript.

Responces to Reviewers

Therefore, the sentence was removed in the revised manuscript.

C6: Line 119 “The Chls and α -Car pigments are mainly involved in the electron transfer reactions and light-harvesting.” It seems that this statement is based on the structure. On what else can the Chls be involved? Please explain.

A6: Chl and α -Car in PSI are involved in electron transfer and light-harvesting. Since the contents in “Pigment environments: antennas and red chlorophylls” in the original version were rewritten in a shorter form and placed at the latter part of “Overall structure” section in the revised manuscript, the sentence would be more appropriate to be placed there and was moved to the top of third paragraph in page 6 with minor change as “In the *A. marina* PSI, a large number of Chl *d* and α -Car harvest light energy and finally transfer the energy to P740”.

C7: Line 119 “These configurations are similar”...” how similar? Please be more precise.

Two Tyr residues, Tyr601/A and Tyr733/A, are positioned within hydrogen bonding distance of P_A . These configurations are similar to those in the PSI from *T. elongatus* (Fig.1), although cofactor compositions are different.

A7: The sentence was meant to state that the electron transfer chain of reaction center is similar between the two organisms as shown in Fig. 1. We apologize for leading misunderstanding. Now, we modified the sentences as follows including the information of hydrogen bonding of two Tyr with P_A in *T. elongatus* PSI: Two Tyr residues, Tyr601/A and Tyr733/A, are positioned within hydrogen bonding distance of P_A as observed in PSI from *T. elongatus*. The configuration of electron transfer chain in *A. marina* PSI is similar to that in the PSI from *T. elongatus* (Fig.1), although cofactor compositions are different.

C8: Line 163-164. The authors state : “Details of the electron transfer mechanism in relation to the $P_A^{\cdot+}/P_B^{\cdot+}$ ratio can now be filled in using the precise structure of *A. marina* PSI.” what does it mean? This part seems to require some calculations that are not presented.

A8: Yes, our structure is very useful to calculate the details. We can find two reports that measured the $P_A^{\cdot+}/P_B^{\cdot+}$ ratio in *A. marina* PSI experimentally. One study reported 8:2 (Mino et al. Chem. Phys.Lett. 2005) and the other reported 6:4 (Hastings and Wang, Photosynth Res 2007) for the ratio in *A. marina*PSI. Our PSI structure would be very useful to determine the ratio theoretically. Therefore, we modified the description as follows: Relating to these mechanisms, two previous studies reported differently on the delocalization of the charge distribution in P740^{41,42}. Details of

Responses to Reviewers

the electron transfer mechanism in relation to the $P_A^{\cdot+}/P_B^{\cdot+}$ ratio can be analyzed in the future, e.g. by theoretical calculations using the precise structure of *A. marina* PSI.

C9: Line 165-180 The discussion of the electron transport chain is confusing, the authors should try to rationalize their findings better and present them in a more structured way.

A9: According to the advice by the reviewer, we rewrote the description on the electron transport chain for clarity as follows in the second paragraph in page 10.

Why is Pheo *a* the A_0 in *A. marina* PSI? According to the midpoint potential value (E_m), it seems reasonable for the Chl *d*-driven PSI to use Pheo *a* as A_0 , since the energy gap is sufficient for the primary electron transfer step as is estimated below. The E_m of P740 vs. the standard hydrogen electrode (SHE) is 439 mV, which is slightly higher than that of P700 (470 mV⁴¹). While the reaction center P740, which is composed of Chl *d/d'*, uses low-energy far-red light of 740 nm (1.68 eV), it generates reducing power almost equivalent to that of the P700 (Chl *a/a'*, 1.77 eV) in plants and most cyanobacteria. The produced reducing power of excited state of P740 (P740*) is weaker by 0.09 eV than that of P700* (i.e., 1.77 – 1.68 eV). Then, the E_m of P740* and P700* could be estimated to be -1,241 mV and -1,300 mV, respectively. This could result in a slower electron transfer rate to A_0 and an increase in reverse reaction without the change in E_m of A_0 , due to the smaller driving force. It is reported that the rates of electron transfer from P740* to A_0 , and to PhyQ are actually comparable to those from P700* to A_0 and to PhyQ³⁴. Then, E_m of A_0 have to change for the proper forward reaction. Because most of the amino acid residues around A_0 in *A. marina* PSI are similar to those in *T. elongatus*, it is unlikely that the protein structure around A_0 influences the E_m value. Therefore, we took notice of the E_m value of cofactor molecule itself, and obtained E_m (vs. SHE) values of purified Chl *a*, Chl *d*, and Pheo *a* in acetonitrile of -1100, -910, and -750 mV, respectively (Extended Data Fig. 21). The E_m of Pheo *a* is the highest of these. Accordingly, Pheo *a* as A_0 should achieve the same electron transfer efficiency as the Chl *a*-type PSI. Then, a possible effect on the rate of the following A_0 to PhyQ step should be investigated as the rate is also known to be comparable to that in Chl *a*-type PSI.

C10: 184-185. The small amount of Chl *a* is not mentioned at the beginning when the authors report the number of pigments present in the complex.

A10: Yes, only cofactors assigned by structural analysis were presented. But now, the result of the pigment analysis for Chl *a* was added in the middle part of page 6 in the revised manuscript as follows.

In addition, a small amount of Chl *a* (approximately one Chl *a* per Chl *d'*) were detected by pigment analysis.

C11: 199-203. Are the differences between the distances significant? The authors should discuss them considering the uncertainty in the structure determination.

Responses to Reviewers

A11: The significance of the distances cannot be discussed in terms of the cryo-EM map accuracy obtained in this study. Therefore, we now only presented the distances estimated in the obtained structure for reference without comparison with those of *T. elongatus* as follows:

In *A. marina* PSI, these distances were estimated to be 6.2 Å (Acc_A-P_A) and 6.6 Å (Acc_B-P_B), respectively (Fig. 4D), in the present structure. (lines 18 - 19 in page 11)

C12: 207-208 “These structural differences may suggest that the protein environment within *A. marina* PSI is modified to optimize electron transfer involving Chl *d*”. This is a very generic statement, the authors should at least formulate some hypotheses.

A12: As is wrote in the text to explain the certainty of Phe *a* as A₀ (second paragraph in page 10), preventing reverse reaction is very important for the effective electron transfer reaction. Therefore, according to the comment by the reviewer, we modified this statement as follows (from line 4 to 1 from the bottom in page 11):

These structural differences may suggest that the protein environment within *A. marina* PSI containing Chl *d* is modified to support forward electron transfer by suppressing the reverse electron transfer from PhyQ to A₀.

C13: Line 215- “Chl *d* and α -Car harvest low energy far-red light”. Carotenoids do not harvest far-red light

A13: The reviewer is correct. The sentence was changed as follows in the revised manuscript (line 15 in page 6).

“, a large number of Chl *d* and α -Car harvest light energy and finally transfer the energy to P740.”

C14: Line 220 – “.. lack of some Chls, which, we assume, reflects optimization for efficient use of far-red light”. How can the lack of some Chls reflect optimization for efficient use of far-red light? Please explain.

A14: We have to wait the theoretical calculation using our structure for the precise understanding. However, far-red light utilizing PSI whose structures were resolved until now carries Chls less than that of *T. elongatus* PSI, which makes us assume that optimization resulting from far-red light utilization leads to less Chls. Less amount of Chl would relate to the low consuming energy for the biosynthesis of Chls. The amount of Chls in far-red light utilizing PSI are shown in the

Responces to Reviewers

revised manuscript without presumption as below. And the sentence Reviewer commented is not included in the revised manuscript since only structural information is described for antenna system to avoid speculation.

<the second paragraph in page 7>

Therefore, we deduce here that the number of Chl associated with PSI is smaller than in other oxygenic phototrophs. Actually, Chls corresponding to Chl-88 and Chl-94 in *T. elongatus* are surely absent in *A. marina* PSI. Unassigned Chls in *A. marina* PSI compared with those in *T. elongatus* PSI are shown by marking N/D in the column of ID number in Extended Data Table 6, and those Chls can be recognized by chlorins in transparent gray in Extended Data Fig. 17. The numbers of Chls in PSI of Chl *f*-carrying *Fischerella thermalis* and *H. hongdechloris* are also smaller than that of *T. elongatus* PSI; 89 (*F. thermalis*)³⁰ and 90 (*H. hongdechloris*)¹⁹ Chls vs 96 Chls in *T. elongatus* PSI. Type I reaction center of *Heliobacterium modesticaldum* carries much smaller amount of Chl species; 60 molecules per reaction center²⁹.

C15: From line 222 – The discussion of the possible location of the red forms is not correct. The authors seem to suggest that the location in other PSI complexes is known and that the shift is due to interactions with the protein, this is not correct.

A15: The location of red Chls in *T. elongatus* PSI is predicted by theoretical calculation (Jordan et al. 2001; Byrdin et al. 2002). And most Chls in PSI shows red-shifted feature, which is usually explained by the interaction with or affection by the internal environment formed by protein (Adolphs et al. 2010) although there could be other factors influencing the spectroscopic property of individual Chl molecule. However, the description on red forms was totally removed in the revised manuscript since it is difficult to determine their position based solely on the structure.

Jordan, P. *et al.* Three-dimensional structure of cyanobacterial photosystem I at 2.5 Å resolution. *Nature* **411**, 909–917 (2001).

Byrdin, M., Patrick Jordan, P., Krauss, N., Fromme, P., Stehlik, D., and Schlodder, E. Light Harvesting in Photosystem I: Modeling Based on the 2.5-Å Structure of Photosystem I from *Synechococcus elongatus*. *Biophys. J.* **83**, 433–457 (2002)

Adolphs, J., Müh, F., Madjet, M. E. A., Am Busch, M. S. & Renger, T. Structure-based calculations of optical spectra of photosystem I suggest an asymmetric light-harvesting process. *J. Am. Chem. Soc.* **132**, 3331–3343 (2010).

C16: Line 231 “In *T. elongatus* PSI, 9–11 Chl a per PSI monomer are considered to be red Chls”. The reference is missing. The numbers seem very large, the authors should carefully check the literature.

A16: In the report by Jordan et al. (*Nature*, 2001) and Pålsson et al. (*Biophys. J.*, 1998), the number of Red Chls are mentioned to be 9-11 for *Synechococcus elongatus* (currently,

Responces to Reviewers

Thermosynechococcus elongatus). While, the number of red Chl is variable depending on the source organisms. For example, the number is two, much smaller in organisms such as *Synechococcus* PCC 7942 (Andrizhiyevskaya et al. BBA, 2002) or *Chlamydomonas reinhardtii* (Gibasiewicz et al. J. Phys. Chem. B, 2005).

In the revised manuscript, we decided to remove the description on red Chls because *A. marina* does not harbor red Chls (Extended Data Fig. 8 in the revised manuscript).

- Jordan, P. *et al.* Three-dimensional structure of cyanobacterial photosystem I at 2.5 Å resolution. *Nature* **411**, 909–917 (2001).
- Palsson, L. O., Dekker, J. P., Schlodder, E., Monshouwer, R. & Van Grondelle, R. Polarized site-selective fluorescence spectroscopy of the long-wavelength emitting chlorophylls in isolated Photosystem I particles of *Synechococcus elongatus*. *Photosynth. Res.* **48**, 239–246 (1996).
- Andrizhiyevskaya, E. G. *et al.* Spectroscopic properties of PSI-IsiA supercomplexes from the cyanobacterium *Synechococcus* PCC 7942. *Biochim. Biophys. Acta - Bioenerg.* **1556**, 265–272 (2002).
- Gibasiewicz, K. *et al.* Characterization of low-energy chlorophylls in the PSI-LHCI supercomplex from *Chlamydomonas reinhardtii*. A site-selective fluorescence study. *J. Phys. Chem. B* **109**, 21180–21186 (2005).

C17: Line 234-235 – “In *A. marina*, this latter region seems highly flexible, and Chl d 55, 56, 63–67, and 77–79 could not be assigned as counterparts of Chl a molecules”. What does it mean? That there are no Chls in these positions?

A17: We thought that the reason these Chls (Chl d 55, 56, 63-67, 77-79) could not be definitely assigned was the regional flexibility of the *A. marina*-PSI structure. We reasonably assumed that the regional flexibility is closely related to the somewhat lower local resolution of some parts of outer region (Extended Data Fig. 11). In the revised manuscript, we removed this statement to avoid excessive speculation which had been written in the Antenna section in the previous version.

C18: Line 237-238 – “the *A. marina* PSI intrinsically does not exhibit features of a long-wavelength Chl d (Extended Data Figs. 2, 6, and 7)”. This statement does not seem to be correct or at least needs to be explained. The absorption spectrum peaks around 705 nm and there is a shoulder around 730 nm; these are red-shifted Chls in my opinion.

A18: Yes, they are red-shifted Chls including most other Chl associated with *A. marina* PSI. However, they are not so-called red Chls. We mentioned red-Chls here and we apologize for leading misunderstanding by using the words “long-wavelength Chl *d*”. Red-Chls are Chls that

Responces to Reviewers

absorbs light whose wavelengths are longer than that of Qy peak of reaction center; 740 nm for *A. marina* PSI.

As is written in A16, in the revised manuscript, we removed the description on red Chls because *A. marina* does not harbor red Chls (Extended Data Fig. 8 in the revised manuscript).

C19: From 239 - the discussion about the absorption properties of the pigments based on differences in the residues located nearby is not sound and not informative. The complex analyzed here contains Chl d, it has far less Chls than the Chla- PSI and we do not even know where each Chl of the Chl-a containing PSI absorb. To say that a specific Chl can absorb differently than in other PSI is noninformative. I suggest to avoid these speculations; they do not add anything to the work.

A19: We agree that it is difficult to determine spectroscopic property of individual Chl molecules experimentally without theoretical calculation based on our current structure. Therefore, according to the reviewer's advice, we deleted the related description.

C20: Line 247 "The *A. marina* PSI appropriately lacks the Chl with most increased site-energy that captures high-energy, short-wavelength light in *T. elongatus*". On what is this conclusion based?

A20: Adolphs et al. (2010) theoretically calculated that the positive charge of Arg24/M induces a significant increase in the site-energy of Chl *a* 94 in *T. elongatus* PSI. In *A. marina* PSI, Chl pigment corresponding to Chl *a* 94 in *T. elongatus* PSI is apparently absent (line 15 in page 7 in the revised manuscript). The description commented by Reviewer was based on the absence of this pigment and the calculation by Adolphs et al (2020) above. However, in the revised manuscript, we removed the description regarding spectroscopic properties which does not associate experimental data.

Adolphs, J., Müh, F., Madjet, M. E. A., Am Busch, M. S. & Renger, T. Structure-based calculations of optical spectra of photosystem I suggest an asymmetric light-harvesting process. *J. Am. Chem. Soc.* **132**, 3331–3343 (2010).

C21: Line 255 "These differences help explain the unique features of light harvesting in *A. marina*."

The unique features of *A. marina* is due to the presence of Chl d in most binding sites. The substitution of one specific aminoacid is certainly not responsible for it.

A21: Yes, of course, the reviewer is correct. However, it is also true that the light-capturing property of individual Chl molecule is affected by its environments including interaction with nearby amino acids. Because of this, the half width at half maximum of Qy band in reaction center

Responses to Reviewers

complexes becomes quite large as we can deconvolve it compared with that of Chl in solvent. To make more clearer, we changed the sentence as follows. (from line 4 in page 7)

These differences help explain the unique features of light harvesting in *A. marina* through theoretical calculations using the present structure or spectroscopic analysis using mutants and so on.

C22: Line 268 “Again, these divergences must affect energy transfer efficiencies in the two organisms.” Again, this statement is noninformative for the reasons I explained above.

A22: As is wrote above (A21), spectroscopic property of individual Chl molecule could be affected by surrounding environments. However, it is also true that such spectroscopic properties and resulting energy transfer between Chl molecules can be obtained through theoretical calculation based on structure. Therefore, according to the comment by the reviewer, we removed these statements from the revised manuscript.

Responces to Reviewers

Reviewer #2 (Remarks to the Author):

Acaryochloris marina, a cyanobacterium capable of using the energy of far-red light to drive oxygenic photosynthesis, contains a photosystem I (PSI) with Chl *d* as the major pigment. Chl *d* is a special type of chlorophyll molecule absorbing at a wavelength range (700-750 nm) longer than Chl *a*. The structures of photosystem I with Chl *a* as the major chlorophyll have been solved (such as the ones from *T. elongatus*, etc), but the structure of PSI from *A. marina* remains to be analyzed at high resolution so that the exact locations of Chl *d* molecules can be assigned clearly. Hamaguchi T. et al. report the cryo-EM structure of PSI from *A. marina* at 2.5 Å resolution and reveal the detailed arrangement of various cofactor molecules within the complex. The assignment of Chl *d* molecules in the P740 special pair is consistent with previous spectroscopic studies. Unexpectedly but interestingly, a pair of Pheo *a* molecules were found to take the positions of primary electron acceptor, a new feature not found for any PSI before. The following are a few questions and suggestions for the author to consider during revision.

A: Thank you for your evaluation based on your deep understanding on structural analysis. Here are responses to your comments.

C1: While the previous spectroscopic study suggested the P740 reaction center of PSI from *A. marina* is likely a heterodimer of Chl *d*, the slight difference between Chl *d* and Chl *a* (O atom vs CH₂ group on the C3_2 position) is too little to be detectable in the 2.5-Å resolution cryo-EM map. Will the authors observe any Fo-Fc (or real space analog of Fo-Fc) difference map densities around the C3(2)-groups if the reaction center is modeled as Chl *a*? The difference map can also be utilized in searching for potential water molecules around the C3-formyl groups in Chl *d* and verifying the assignment of Pheo *a* on the A0 sites.

A1: We tried to evaluate Acc as Chl *a* rather than Chl *d*, but the shape of its map and that of PSI model did not differ. Therefore, we did not pursue further for P740. Although the difference map calculation is a valid cryo-EM method, it is difficult to assess the difference between the oxygen and carbon atoms with the present cryo-EM map distribution (GSFSC0.143 = 2.5 Å).

We appreciate your advice on searching water molecules using the difference map. As is commented by Reviewer #3 (comment #11), we identified fewer water molecules in our structure than in the structure by Jordan et al. (X-ray crystallography). This is because the resolutions of the cryo-EM structure differ from one part to another (this data is shown in Extended Data Fig. 11 (A)), and the resolutions of the outer region of PSI are lower than the central parts of PSI. For this reason, we detected the water molecules with significant cryo-EM map contour level for identification in this study. For the assignment of Phe *a* for A0, we added

Responses to Reviewers

new figure (Fig. 5) to help convincing recognition of the absence of Mg^{2+} at the center of chlorin of A_0 .

Jordan, P. *et al.* Three-dimensional structure of cyanobacterial photosystem I at 2.5 Å resolution. *Nature* **411**, 909–917 (2001).

Fig. 5. Each cryo-EM density maps around P740 (A, B), Acc (C, D) and A_0 (E, F). Each map is shown in a mesh representation at 3.0 sigma contour level for panels A, B, E, and F, and 5.0 sigma for panels C and D contour level for the ease of recognition of water molecule coordinating to Mg^{2+} of chlorin.

C2: As an indirect evidence supporting the assignment of Chl d (instead of Chl a) in the cryo-EM map, are there hydrogen-bond donors nearby the C3-formyl groups of the putative Chld molecules in the electron transfer chains (P_A , P_B , AccA and AccB)? Alternatively, is the environment around the C3-formyl groups of Chl d more polar than the corresponding one of Chl a in *T. elongatus* PSI?

A2: The formyl group of P740 (P_A and P_B) did not form a hydrogen bond with another molecule (e.g. water molecule), and unfortunately, we could not identify P740 as Chl d from the hydrogen bond network. This situation is similar to that of P700 in the surrounding polar environment of *T. elongatus*-PSI and we could not determine P740 to be Chl d from the environment. Therefore, we identified P740 as a heterodimer of Chl d' and Chl d based on previously reported spectral analysis (Itoh *et al.* 2007; Tomo *et al.* 2008).

Acc (AccA and AccB) were identified as Chl d in our study, but the C3-formyl group was oriented in the opposite direction compared to that of P740 (P_A and P_B). If the C3-formyl group

Responses to Reviewers

of Chl *d* interacts with its own hydrogen atom (a hydrogen atom bound to C5), it could take a structure similar to that in Extended Data Fig. 1 (C), but the orientation of the formyl group of Acc would be the opposite. If we evaluate Acc on this basis alone, then Acc could be Chl *a*. In addition, there were no hydrogen-bonding networks in the C3-group of Acc even if the surrounding structure of the C3-group was carefully checked. Eventually, we assigned Acc as Chl *d* because it was hard to assign with structural confidence to assign Acc as Chl *a*.

These situations were now added in the revised manuscript from line 13 in page 9 as follows:

The orientations of the formyl group in P_A (Chl *d'*) and P_B (Chl *d*) were identified by considering the distribution of the cryo-EM density (Fig. 5A and B). No amino acid residues and pigments capable of forming hydrogen bonds with these formyl groups were found in the vicinity of P_A and P_B. This suggests that these formyl groups form hydrogen bonds with the C5 H atom in Chl *d'* and Chl *d*, respectively. Similarly, there were also no amino acid residues and pigments capable of forming a hydrogen bond around the formyl group of Acc (Acc_A and Acc_B) (Fig. 5C and D). However, the orientations of the formyl group of Acc (Acc_A and Acc_B) were altered by the hydrophobic environment caused by Trp (Trp585/B and Trp599/A) when compared with P740.

Itoh, S. *et al.* Function of chlorophyll d in reaction centers of photosystems I and II of the oxygenic photosynthesis of *Acaryochloris marina*. *Biochemistry* **46**, 12473–12481 (2007).

Tomo, T. *et al.* Characterization of highly purified photosystem I complexes from the chlorophyll d-dominated cyanobacterium *Acaryochloris marina* MBIC 11017. *J. Biol. Chem.* **283**, 18198–18209 (2008).

C3: In lines 184-195 on p8, the authors initially suggested that AccA and AccB may be assigned as Chl *a*, but then concluded that “The assignment of Chl *d* to the Accs in this study is actually due to the difference spectrum of P740, ...”. Could the authors estimate or discuss the relative efficiency of Acc (Chl *a*), compared to Acc(Chl *d*), as a intermediate cofactor mediating electron transfer from P740 to A0 (Pheo *a*)?

A3: Such estimation would be good to evaluate the suitability of Chl *a* or Chl *d* for Accs under the current resolution. However, unfortunately, we were unable to make such inference based on structural data alone. We think that more detailed spectral analysis or theoretical calculations are needed to determine Acc to be Chl *a* or Chl *d*.

C4: Zeaxanthin is a carotenoid with important photoprotective function (Niyogi, K. *et al.* PNAS, 94: 14162-14167, 1997). The HPLC analysis result indicates that there is one zeaxanthin

Responses to Reviewers

molecule in the PSI from *A. marina*. Is this a structural component of PSI or contamination by other complexes? If it is a structural component of PSI, where is it located?

A4: In this study, unfortunately, we were unable to identify zeaxanthin in the structure of *A. marina*-PSI because of the limit of resolution in our structure. However, the number of α -Cars identified by structural analysis was 11 molecules, and the number of carotenoids (α -Cars + zeaxanthin) identified by pigment analysis was approximately 11 molecules. Therefore, we think that the zeaxanthin detected by pigment analysis is not a contamination but one of the pigments contained in *A. marina*-PSI.

C5: It will be great if the cryo-EM densities of AccA and AccB can be included in Fig. 4A.

A5: Thank you for your advice. We added a figures of Acc (AccA and AccB) in the revised manuscript as Fig. 5 C and D.

C6: The statement of the surrounding structures of FX, FA, FB must be invariant for the reduction of NADP⁺ (lines 211-212, p9) is too strong. It might be better to use “conserved” or “highly similar” instead of “invariant”.

A6: Thank you for your advice. We changed the word ”invariant“ to “conserved“ in the revised manuscript as follows (lines 2 and 3 in page 12):

We suggest that F_X, F_A, F_B, and their surrounding structures must be conserved for the reduction of NADP⁺.

Responces to Reviewers

Reviewer #3 (Remarks to the Author):

C: The manuscript of Hamaguchi et al. describes the structure of PSI from the unusual cyanobacterium *Acarychloris marina*, which uses mostly Chl d to perform oxygenic photosynthesis. The authors claim the structure has an overall resolution of 2.48 Å, although this is very difficult to believe given missing subunits, disordered segments of multiple polypeptides, etc. Perhaps the most interesting claim, if verified, is that the A₀ acceptor is pheophytin a, something that surprisingly had not been revealed in previous spectroscopic studies. It is not clear that this manuscript is in the correct format, as it seems to have been written in a style more like that used by other journals. As a result, many points are not very fully described and important details are lacking. Comments for the authors follow.

A: We are very grateful to Reviewer #3 for her/his critical reading comments to improve our manuscript and her/his positive evaluation to our novel finding of pheophytin as a primary electron acceptor A₀. Additionally, we have important results on the analysis of P740 and surrounding structures in *A. marina*-PSI.

The cryo-EM map obtained in this study and the validation of the *A. marina*-PSI model refined from the map are explained in Extended Data Fig. 10-12, and Extended Data Table 3 and 4. The definition of resolution of cryo-EM structure is different from that of crystal structure. The resolution of cryo-EM map is determined by the FSC (Fourier Shell Correlation)-based comparison of two half maps by the “gold standard FSC” at 0.143 criterion (Scheres and Chen, Nature Methods, 2012) (Extended Data Fig. 11 (C)). In this study, the refined PSI model was validated by Q-score, measurement of atom resolvability in cryo-EM map (Pintilie et al., Nature Methods, 2020), and its resulting estimated resolution was 2.49 Å (Extended Data Table 4).

As is recognized by reviewer #3, our cryo-EM map has some disordered parts in the outer region. This kind of disordered parts are frequently observed for any cryo-EM map because the cryo-EM map reflects the structure of the solution state, which is fundamentally different from crystal structure map. Because of this, the resolution of the structure obtained by cryo-EM differs from one part to another within the structure. Therefore, we presented Extended Data Fig. 10 and Extended Data Table 4 to show Correlation Coefficient (CC) and Q-score values of each amino acid residue, protein subunit, ligand, and water molecule for the valid evaluation by the readers.

We revised our manuscript according to the reviewers' comments, and here are point by point responses to the comments.

Responces to Reviewers

C1: Title. While it is true that this PSI complex absorbs far-red light, that is not what makes it unusual. It is the large Chl *d* content that is unusual, and that should be mentioned in the title.

A1: Yes, it is certainly true. But we believe that the most important uniqueness is that the photochemical reaction is driven by far-red light. In Chl *f*-carrying cyanobacteria, far-red light is transferred to Chl *a* through quantum-theoretical process with environmental thermal energy, which means that the photochemical reaction is driven by red-light, not far-red light. This point was added in the revised manuscript to make readers clearly understand (second paragraph in page 4) as transcribed below. Then, we think that the initial title is suitable for our manuscript. And, no other reviewers advised to change title.

The far-red light-absorbing Chl, Chl *f*, is also utilized in some cyanobacteria to capture far-red light (Extended Data Fig. 2). In combination with Chl *d* -carrying cyanobacteria, acclimation strategy to the light quality expanding to far-red light region gathers attention⁵⁻⁷. Chl *f* that is only induced under far-red light condition occupies less than ~10% of total Chls in photosystem I (seven Chl *f* and 83 Chl *a* in *Halomicronema hongdechloris* PSI¹⁹) and plays solely a light-harvesting function⁷ without engaging photochemical reaction nor electron transfer chain, which is contrasting to Chl *d* that takes place of special pair responsible to photochemical reaction. Low light energy of far-red light captured by Chl *f* is transferred to Chl *a* by quantum-theoretical uphill process using environmental heat for photochemical reaction²⁰. Therefore, the photochemical reaction of Chl *f*-carrying PSI is definitely the same as other PSI operated by Chl *a*; the photoreaction occurs using “visible red light” by P700. However, Chl *d*-driven photoreaction occurs by low-energy far-red light corresponding to Qy band of Chl *d*. Chl *d* also absorbs high-energy blue light as well corresponding to Soret band, but the resulting excited state of Chl relaxes to lower excited state corresponding to Qy level for photochemical reaction. In this sense, Chl *d*-driven system is unique in the photochemical reaction occurring by far-red light level energy.

C2: Abstract, line 7. It is fantasy for the authors to claim that this cyanobacterium “uniquely” can use far-red light for photosynthesis. Dozens of cyanobacteria have been identified that can synthesize Chl *f* and Chl *d*, and these organisms use light wavelengths extending to 800 nm to perform oxygenic photosynthesis.

A2: As is written in the response to comment #1, the uniqueness is on the use of far-red light without quantum-theoretical uphill process that can be seen in Chl *f*-carrying cyanobacteria. Therefore, to show more clearly this point, we modified the sentence as follows (top of Abstract):

***Acaryochloris marina* is one of the cyanobacteria that can use far-red light to drive photochemical reaction for oxygenic photosynthesis.**

C3: Abstract, line 11. It is questionable whether P740 is the primary donor or a secondary donor. In

Responces to Reviewers

other PSI complexes, the Acc (A-1) Chl is now thought to be the primary donor and the P700 dimer is thought to reduce the cation produced by transfer of an electron from A-1 to A0.

A3: Yes, it is an important recent theory. Accordingly, we will change the word primary donor to special pair or so-called “reaction center“ in this manuscript hereafter. For the first appearance in Abstract, we want to use common word “reaction center”.

The primary electron donor P740 is a dimer
=> The so-called reaction center P740 is a dimer (line 5 in Abstract)

C4: Page 2, lines 20-23. Chls absorb light in the blue and the red, not only in the red. BChls also have absorbance in the blue/ultraviolet region of the spectrum.

A4: Of course, the reviewer is correct. We modified the sentence as follows (from line 21 to 23 in page 2):

Anoxygenic photosynthetic bacteria use bacteriochlorophylls (BChl) that absorb 800–900 nm light in addition to blue/ultraviolet light with either a type I or type II reaction center.

C5: Page 2, line 18. Chl *d* is not “novel” and is not exclusively found in *A. marina*. It is also found in cyanobacteria that synthesize Chl *f* and perform far-red light photoacclimation.

A5: The reviewer is correct. Originally, we are not intending to claim that *A. marina* is the only cyanobacterium that uses Chl *d*. However, to avoid confusion, we modified the sentence as follows (Line 2 from the bottom in page 2).

Acaryochloris marina (*A. marina*) was found as a cyanobacterial species that uses Chl *d* (Extended Data Fig. 1), which absorbs 700–750-nm light *in vivo*⁴ (Extended Data Fig. 2).

Additionally, information on Chl *f* was also added following the above modified sentence (the last line in page 2, second paragraph in page 4). For many readers, the comparison between Chl *d* and *f* would be very useful. Therefore, we made the following figure (Extended Data Fig. 2 in the revised manuscript), in which, the spectra of *Halomicronema hongdechloris* (blue line) cells is a courtesy of Prof. Tatsuya Tomo.

(from the last line in page 2)

Later, several cyanobacteria were found to induce limited amount of Chl *f* for capturing far-red light only under far-red light condition⁵⁻⁷ (Extended Data Fig. 2, and see below).

(second paragraph in page 4)

- Please see transcribed sentences in the response to comment 1 (A1) -

Extended Data Fig. 2: Comparison of cellular absorption spectra of three cyanobacteria. Absorption spectra of *A. marina* (red line) and *Thermosynechococcus vulcanus* (green line) cells equivalent to 8.1 $\mu\text{g Chl } d/\text{mL}$ and 5.0 $\mu\text{g Chl } a/\text{mL}$, respectively, were measured with an integrating sphere. Absorption spectrum of Chl *f*-carrying *Halomicronema hongdechloris* (blue line) cells, which is a courtesy of Prof. Tatsuya Tomo, is also presented. Spectra were normalized by the peak height of the Qy band with offset at 800 nm.

C6: Page 3, lines 6-7. This statement is simply false. Chl *d* is not uniquely found in *A. marina*, as noted above, and it is not the lowest-energy absorbing pigment found in cyanobacteria. That would be Chl *f*, which absorbs productively to 800 nm.

A6: Please see the above response A5.

C7: Page 4, line 4. The heliobacterial RC contains both BChl *g* and 81-OH-chlorophyll *a*. Two cyanobacterial PSI structures have been determined for complexes containing Chl *f*: Kato et al. and Gisriel et al. 2020a, 2020b.

A7: Thank you for your advice. We added the information as follows (second and third paragraphs in page 4).

To date, several structures of type I reaction centers have been determined from higher plants^{21,22}, green algae^{23,24}, red algae²⁵, diatoms²⁶, cyanobacteria^{1,27,28} and an anoxygenic bacterium²⁹. All structures including Chl *f*-carrying cyanobacteria^{19,30} have Chl *a* in the reaction center with the exception of the one from the anoxygenic bacterium which has BChl *g* and 81-OH-chlorophyll *a*.

19. Kato, K. *et al.* Structural basis for the adaptation and function of chlorophyll *f* in photosystem I. *Nat. Commun.* **11**, 238 (2020).

Responses to Reviewers

30. Gisriel, C. *et al.* The structure of Photosystem I acclimated to far-red light illuminates an ecologically important acclimation process in photosynthesis. *Sci. Adv.* **6**, eaay6415 (2020).

C8: Page 4, lines 4-6. No matter how many times the authors repeat this statement, it is not correct that *A. marina* “uniquely” utilizes far-red light for oxygenic photosynthesis. This must be corrected.

A8: According to the comment by the reviewer, we removed the word “uniquely” here.

C9: Page 4, lines 5-7. Three papers describing PSI complexes that can use far-red light have already appeared: Kato *et al.*, 2020; Gisriel *et al.*, 2020a, 2020b. These structures were also determined to understand how PSI can use far-red light. This study is not unique and the other relevant studies should be appropriately cited and contrasted.

A9: Chl *d* in *A. marina* is permanently expressed for photochemical reactions. On the other hand, Chl *f* in other far-red light utilizing organisms is only slightly expressed in the far-red environment (less than 10%). From this perspective, *A. marina*, which uses Chl *d* as the major pigment and drives photochemical reactions with far-red light directly without uphill energy transfer using environmental thermal energy, is unique. The difference of far-red light usage between Chls *d* and *f* was added with relevant papers were added in the revised manuscript (please see transcribed sentences in A1; second paragraph in page 4).

C10: Page 5, lines 4-7. If Psa27 and PsaI are essentially the same thing—and functionally they are even if the sequence homology is low—why give the protein another name (which in any event would not be valid by Demerec rules?).

A10: This subunit has been named Psa27 by Tomo *et al.* (JBC 2008) considering the absence of known homology of sequence. Since the situation is similar to PsbQ' of red algae (same function as PsbQ with less homology), we think no problem for the usage of previously-named Psa27 for the subunit.

Tomo, T. *et al.* Characterization of highly purified photosystem I complexes from the chlorophyll *d*-dominated cyanobacterium *Acaryochloris marina* MBIC 11017. *J. Biol. Chem.* **283**, 18198–18209 (2008).

Responces to Reviewers

C11: Page 5, lines 10-18. Given missing subunits, disordered loops, and unassigned amino acids due to resolution issues, it is very difficult to imagine that the stated resolution for the structure could be 2.48 Å, slightly better than that of Jordan et al. for the X-ray structure of PSI from *Thermosynechococcus elongatus*. That structure had more than 200 assigned water molecules, and this one only 84, another indication that the overall resolution is unlikely to be so high as the authors are claiming.

Furthermore, the authors state that they find 71 Chl *d* molecules, 1 Chl *d'*, 2 pheophytins (should be pheophytin *a*), and only 11 alpha-carotene molecules (the elution profile shown in the supplement indicates the presence of zeaxanthin and probably either echinenone or cryptoxanthin as well). *T. elongatus* PSI was originally assigned 96 Chl *a* (now thought to be only 95) and 22 beta-carotenes, which again suggests that large parts of the structure have rather poor resolution. Given the similarity of the protein sequences for PsaA and PsaB, it is hard to imagine that there should be 23 fewer Chls in the complex. It would be helpful for the authors to produce a clear statement about what is expected—how many Chl *a*, pheophytin *a*, and Chl *d* molecules are expected to be present in the complexes based on biochemical analysis—and how many are found and just as importantly which Chl *a* molecules of *T. elongatus* PSI are NOT found. Same for carotenoids.

A11: Regarding the resolution matter, please refer to the response to your general comment at the top; the validation of cryo-EM map and the refined PSI model are shown in Extended Data Fig. 10-12, and Extended Data Table 3 and 4.

As is recognized by the reviewer, the resolutions of the outer regions of *A. marina*-PSI is low (we have already shown this data in Extended Data Fig. 11 (A)), and some parts in the outer region may have been desorbed. Also, some pigments could be released during sample preparation. However, considering the standard errors among independent 5 preparations (Extended Data Table 7) that is smaller than the previous report (Tomo et al. 2008), there is still possibility that the number of pigments could be actually smaller than other type PSI. One paragraph describing such situation and unassigned Chls compared with *T. elongatus* PSI was added as follows in the revised manuscript (second paragraph in page7).

Thank you for your suggestion on the candidate of carotenoid separated by HPLC. We had tried to identify the carotenoid eluted at ~42 min, but we found that it is neither of echinenone nor cryptoxanthin based on the elution time and the spectra of their standard samples.

(second paragraph in page7)

Although the amount of Chl *d* per Chl *d'* determined by pigment analysis (67.0 ± 0.66 , $n=5$ of independently prepared PSI) and assigned by structural analysis (71 including one or two Chl *a*) in this study is lower than those in previous studies (145 ± 8^{14} and 97.0 ± 11.0^{12}), the semi-stoichiometric amount of Phe *a* (1.92 ± 0.022) that was 0.3 ± 0.2 per reaction center in the

Responces to Reviewers

previous study¹² is consistent with the structural analysis in this study. Although local resolution of some parts of outer region is somewhat lower, our PSI was stable to keep its integrity for five days (Extended Data Fig. 5). Therefore, we deduce here that the number of Chl associated with PSI is smaller than in other oxygenic phototrophs. **Actually, Chls corresponding to Chl-88 and Chl-94 in *T. elongatus* are surely absent in *A. marina* PSI. Chls that were unable to be assigned in *A. marina* PSI compared with those in *T. elongatus* PSI are shown by marking N/D in the column of ID number in Extended Data Table 6, and those Chls can be recognized by chlorins in transparent gray in Extended Data Fig. 17.** The numbers of Chls in PSI of Chl *f*-carrying *Fischerella thermalis* and *H. hongdechloris* are also smaller than that of *T. elongatus* PSI; 89 (*F. thermalis*)³⁰ and 90 (*H. hongdechloris*)¹⁹ Chls vs 96 Chls in *T. elongatus* PSI. Type I reaction center of *Heliobacterium modesticaldum* carries much smaller amount of Chl species; 60 molecules per reaction center²⁹.

Tomo, T. *et al.* Characterization of highly purified photosystem I complexes from the chlorophyll *d*-dominated cyanobacterium *Acaryochloris marina* MBIC 11017. *J. Biol. Chem.* **283**, 18198–18209 (2008).

C12: Page 5, lines 21-23. Considering that the authors do not have sufficient resolution to identify the Chls at the Acc/A–1 position and may not have sufficient resolution according to the text in the methods to assuredly specify that the A0 tetrapyrrole is Pheo *a*, this statement is not correct. They may have assigned four pigments as 3 Chl *d* and 1 Chl *d'*, but the authors don't know that this is correct.

A12: The resolution of the current structure (2.5 Å) is higher than that of Chl *f*-carrying PSI reported by Gisriel (2020) and comparable with that reported by Kato et al. (2020) that were analyzed by cryo-EM; global resolution of 3.19 Å (~3.0 Å of local resolution for important elements) and 2.41 Å, respectively.

We had described the assignment of Acc as below in the revised manuscript (second paragraph in page 11). Since we could not identify Acc as Chl *a* conclusively, we assigned Acc as Chl *d* based on the previous report by Itoh et al. (Biochemistry, 2007). Even if the resolution exceeds 2.0 Å (by either cryo-EM or X-ray crystallography), it is impossible to distinguish between oxygen and carbon atoms based only on the difference of the signal level. Therefore, in order to distinguish between vinyl and formyl groups, it is necessary to confirm the structure surrounding them. The structure surrounding the pigments involved in electron transfer reactions, such as P740 and A₀, in *A. marina*-PSI were almost the same as those of *T. elongatus*-PSI. Unfortunately, Chl *d* (or Chl *a* in *T. elongatus*) could not be identified by referring the surrounding structure. Eventually, we identified Chl *d* in *A. marina*-PSI considering the description of the reference below (Saito et al. 2012; Schulte et al. 2010).

Regarding A₀ (A_{0A} and A_{0B}), no Mg²⁺-derived density maps were detected and it was reasonably enough to identify these molecules (A_{0A} and A_{0B}) to be pheophytin based on the

Responses to Reviewers

cryo-EM map. For the purpose of clear understanding of the absence of Mg^{2+} -derived density maps, we added a new Fig. 5 (see panels E and F) in addition to Fig. 4A. Furthermore, Leu (Leu665/B in *A. marina*-PSI) should not serve as a ligand to Mg^{2+} in Chl, which is described in lines 9-12 on page 9 as transcribed below.

(second paragraph in page 11 in the revised manuscript)

A. marina contains a limited but distinct amount of Chl *a*^{12,14,36}, and we found a small amount in the PSI (1–2 Chl *a* per PSI monomer) (Extended Data Table 7). One possible locality is as the accessory Chls, Acc_A and Acc_B (Figs. 1 and 4). However, unfortunately, Acc could not be conclusively identified as Chl *d* or Chl *a*, because the functional group containing C3¹ of Acc Chl could not be precisely defined from the cryo-EM density map. While quantum mechanical/molecular mechanical calculations show that the formyl group of Chl *d* adopts two orientations (Extended Data Fig. 1C, D)⁴⁶, the oxygen atom of the formyl group on the Chl *d* molecule in a vacuum is more stable when oriented towards the C5 H atom, suggesting the conformer in Extended Data Fig. 1C. In contrast, the vinyl group of Chl *a* can adopt either orientation. The assignment of Chl *d* to the Acc s in this study is actually due to the difference spectrum of P740, which suggests that the large electrochromic shift of Acc at 710 nm arises from Chl *d*¹¹. The Mg^{2+} in Acc_A and Acc_B are coordinated by water molecules forming hydrogen bonds with Asn588/B and Asn602/A, respectively (Fig. 4B and C). In *T. elongatus* PSI, the methyl ester groups of the Chl *a* in Acc_A and Acc_B affect the charge and spin distributions on P700^{33,47–49}. In *A. marina* PSI, these distances were estimated to be 6.2 Å (Acc_A -P_A) and 6.6 Å (Acc_B -P_B), respectively (Fig. 4D), in the present structure.

(lines 9-12 on page 9)

Furthermore, the position of Leu665/B (Extended Data Fig. 20B)³⁶, pointing to A_{0B} and in the same place as conserved Met688/B that ligates A_{0B} in *T. elongatus* PSI^{1,36}, supports Pheo *a* as A_{0B} in *A. marina*, because Leu cannot serve as a ligand to Mg^{2+} in Chl.

Fig. 5. Each cryo-EM density maps around P740 (A, B), Acc (C, D) and A₀ (E, F).

Responces to Reviewers

Each map is shown in a mesh representation at 3.0 sigma contour level for panels A, B, E, and F, and 5.0 sigma for panels C and D contour level for the ease of recognition of water molecule coordinating to Mg²⁺ of chlorin.

- Gisriel, C. *et al.* The structure of Photosystem I acclimated to far-red light illuminates an ecologically important acclimation process in photosynthesis. *Sci. Adv.* **6**, eaay6415 (2020).
- Kato, K. *et al.* Structural basis for the adaptation and function of chlorophyll *f* in photosystem I. *Nat. Commun.* **11**, 238 (2020).
- Itoh, S. *et al.* Function of chlorophyll *d* in reaction centers of photosystems I and II of the oxygenic photosynthesis of *Acaryochloris marina*. *Biochemistry* **46**, 12473–12481 (2007).
- Saito, K., Shen, J. R. & Ishikita, H. Cationic state distribution over the chlorophyll *d*-containing P D1/PD2 pair in photosystem II. *Biochim. Biophys. Acta - Bioenerg.* **1817**, 1191–1195 (2012)
- Schulte, T., Hiller, R. G. & Hofmann, E. X-ray structures of the peridinin-chlorophyll-protein reconstituted with different chlorophylls. *FEBS Lett.* **584**, 973–978 (2010).

C13: Page 6, line 23. “there was no mass.” One does not detect “mass” in an ESP map produced by cryo-EM.

In the methods (page 24, lines 24-25) it is stated that Mg could not be “unambiguously” identified, so by default these positions were defined as Pheo *a*. It is not clear whether they could also be Pheo *d*. Considering that the Mg was not unambiguously shown to be missing, it must be that it is also possible that it is present and that these tetrapyrroles are either Chl *d* or Chl *a*. Perhaps the authors are not stating clearly what they mean, but it is not going to be clear to readers at what degree of certainty these pigments have been identified in the structure.

Finally, it seems that there should be at least two Chl *a* molecules somewhere in the structure, and authors do not specify where these might be located, although they mention that it could be possible that they are present in the electron transport chain, presumably at the Acc/A–1 positions.

A13: We changed the word “mass” to “Mg²⁺-derived density map” in the revised manuscript as follows (lines 5-7 in page 9).

However, we found that there was no Mg²⁺-derived density map at the center of the tetrapyrrole rings of A_{0A} and A_{0B}, but rather a hole in the cryo-EM density map (Fig. 4A; Fig. 5E, F).

As described above (please see response to your comment #12; A12), Mg²⁺-derived density maps were not detected in chlorin-ring of A₀ (A_{0A} and A_{0B}). Therefore, we identified A₀ to be pheophytin.

Regarding the identity of A₀ to be Pheo *a* or Pheo *d*, we reasonably assumed that A₀ is Pheo *a* because Pheo *d* was not detected by the pigment analysis.

As described above (response to your comment #12; A12), even if the resolution exceeds 2.0

Responces to Reviewers

Å (by either cryo-EM or X-ray crystallography), it is impossible to distinguish between oxygen and carbon atoms based only on the difference in the signal level. Therefore, we identified pigments in *A. marina*-PSI considering the description of the reference below (Saito et al. 2012; Schulte et al. 2010). Regarding the position of Chl *a*, we speculated Acc to be Chl *a* based on the orientation of the C3 group, but we were unable to identify Acc to be Chl *a* as is written in the text (second paragraph in page 11). As is recognized by Reviewer, determination of the position and function of Chl *a* is important. For this purpose, further studies including theoretical calculations are necessary.

Saito, K., Shen, J. R. & Ishikita, H. Cationic state distribution over the chlorophyll *d*-containing P D1/PD2 pair in photosystem II. *Biochim. Biophys. Acta - Bioenerg.* **1817**, 1191–1195 (2012)

Schulte, T., Hiller, R. G. & Hofmann, E. X-ray structures of the peridinin-chlorophyll-protein reconstituted with different chlorophylls. *FEBS Lett.* **584**, 973–978 (2010).

C14: Page 7, line12-13. The authors can speculate about this, but the details of the ETC components do not seem to be fully delineated by this study. The identities of the Acc/A–1 pigments and even the A0 positions are still ambiguous (see #13 above).

A14: As is described above (please see response to your comment #13), the position of A0 is not ambiguous. Even if Acc could not be completely identified whether Chl *a* or Chl *d*, we think it is important to propose the electron transfer reaction of *A. marina*-PSI based on our current structural data. This is because the electron transfer reaction of *A. marina*-PSI has been discussed in the absence of any structural data in previous studies.

C15: Page 7, line 15. Pheophytin *a* is not the found in all type-2 reaction centers, because the bacterial reaction centers of proteobacteria and Chloroflexi, which do not synthesize Chl *a* - only BChl *a*. Thus, they contain Bp heo *a*.

A15: Thank you for your advice. According to the advice of Reviewer #1 (comment #9), the paragraph was totally rewritten and the phrase indicated by this Reviewer was removed in the revised manuscript.

(The second paragraph in page 10. For bold-underlined sentence, see response to comment #16 below.)

Why is Pheo *a* the A₀ in *A. marina* PSI? According to the midpoint potential value (E_m), it seems reasonable for the Chl *d*-driven PSI to use Pheo *a* as A₀, since the energy gap is sufficient for the primary electron transfer step as is estimated below. The E_m of P740 vs. the

Responses to Reviewers

standard hydrogen electrode (SHE) is 439 mV^{12,13,18}, which is slightly higher than that of P700 (470 mV⁴³). While the reaction center P740, which is composed of Chl *d/d'*, uses low-energy far-red light of 740 nm (1.68 eV), it generates reducing power almost equivalent to that of the P700 (Chl *a/a'*, 1.77 eV) in plants and most cyanobacteria¹². The produced reducing power of excited state of P740 (P740*) is weaker by 0.09 eV than that of P700* (i.e., 1.77 – 1.68 eV). Then, the E_m of P740* and P700* could be estimated to be -1,241 mV and -1,300 mV, respectively. This could result in a slower electron transfer rate to A_0 and an increase in reverse reaction without the change in E_m of A_0 , due to the smaller driving force. It is reported that the rates of electron transfer from P740* to A_0 , and to PhyQ are actually comparable to those from P700* to A_0 and to PhyQ³⁴. Then, E_m of A_0 have to change for the proper forward reaction. **Because most of the amino acid residues around A_0 in *A. marina* PSI are similar to those in *T. elongatus*, it is unlikely that the protein structure around A_0 influences the E_m value.** Therefore, we took notice of the E_m value of cofactor molecule itself, and obtained E_m (vs. SHE) values of purified Chl *a*, Chl *d*, and Pheo *a* in acetonitrile of -1100, -910, and -750 mV, respectively (Extended Data Fig. 21)^{43,44}. The E_m of Pheo *a* is the highest of these. Accordingly, Pheo *a* as A_0 should achieve the same electron transfer efficiency as the Chl *a*-type PSI. Then, a possible effect on the rate of the following A_0 to PhyQ step should be investigated as the rate is also known to be comparable to that in Chl *a*-type PSI⁴⁵.

C16: Page 8, lines 1-2. Of course the protein could have strong effects on electron transfer by electrostatic effects that do not have to be directly adjacent to the cofactor.

A16: Thank you for your convincing agreement. Although the paragraph that included the sentence was totally re-written, the sentence is remained in the revised manuscript (second paragraph in page 10). Please refer to the bold underlined sentence in the transcribed sentences in the response to your comment #16 above.

C17: Page 8, lines 6-7. Statement requires a reference.

A17: Thank you. The following article was cited, which is the last part in the transcribed sentences in the response to your comment #16 above.

45. Schenderlein, M., Çetin, M., Barber, J., Telfer, A. & Schlodder, E. Spectroscopic studies of the chlorophyll *d* containing photosystem I from the cyanobacterium, *Acaryochloris marina*. *Biochim. Biophys. Acta - Bioenerg.* **1777**, 1400–1408 (2008).

C18: Page 8, line 2. As noted above, the authors should provide a more detailed analysis of the pigment content of these reaction centers. If there are two Pheo *a*, then it should be possible to quantify Chl *a* better than “1 or 2.” Of course, it is still unclear exactly how many Pheo *a* molecules there are.

The authors do not state that there are no other Pheo *a* molecules in the complex. The authors make no comments about whether there are Hbonding residues in the vicinity of the C-3

Responces to Reviewers

substituent on the Acc/A-1 tetrapyrroles. This might help to resolve which pigment is present in that position.

A18: We performed the pigment analysis of *A. marina*-PSI using HPLC and the values are expressed with standard errors among five independent preparations. The amount of Chl *a* per Chl *d*' was 1.10 with standard errors of ± 0.044 using 5 independently prepared PSI sample (Extended Data Table 7).

As is noted by other reviewers (Reviewer #1, comment 3; Reviewer #4, comment 2), some pigments associating outer part of PSI might be lost during sample preparation. However, we believe that we can discuss the structure of the central part of the *A. marina*-PSI based on our cryo-EM structure and pigment analysis. Based on the analysis of cryo-EM and pigment analysis, two molecules of Pheo *a* are contained in PSI and their positions are A_0 (A_{0A} and A_{0B}). It is not possible that there are other molecules of Pheo *a* elsewhere in *A. marina*-PSI.

Regarding the hydrogen bonds around Acc, if the C3-formyl group of Chl *d* interacts with its own hydrogen atom (a hydrogen atom bound to C5), it could take a structure similar to that in Extended Data Fig. 1 (C), but the orientation of the formyl group of Acc would be the opposite. In addition, there were no hydrogen-bonding networks in the C3-group of Acc even if the surrounding structure of the C3-group was carefully checked. These situations were now added in the revised manuscript in lines 17-24 in page 9 as follows with new Fig. 5:

Similarly, there were also no amino acid residues and pigments capable of forming a hydrogen bond around the formyl group of Acc (Acc_A and Acc_B) (Fig. 5C and D). However, the orientations of the formyl group of Acc (Acc_A and Acc_B) were altered by the hydrophobic environment caused by Trp (Trp585/B and Trp599/A) when compared with P740.

C19: Page 8, lines 17-19. The electrochromic shift argument is a very weak one. In a complex with 90 pigments, there can be small distributed effects or stronger local effects, and these would be nearly impossible to distinguish. Electronic effects can occur over considerable distances.

A19: Yes, the reviewer is correct regarding the electronic effect. However, the light-activated difference spectra cover spectral changes in few pigments directly concerning electron transfer chain, not 90 pigments. Such pigments could be those of P740, Acc and A_0 . Our conclusive assignment of Chl *d* as Accs was based on the published data on such difference spectrum of P740, which suggested that the large electrochromic shift of Acc at 710 nm arises from Chl *d*.

Itoh, S. *et al.* Function of chlorophyll *d* in reaction centers of photosystems I and II of the oxygenic photosynthesis of *Acaryochloris marina*. *Biochemistry* **16**, 12473–12481 (2007).

Responses to Reviewers

C20: Page 9, lines 1-2. Perhaps a more relevant question is the distances between Acc/A-1 and A0, given that this pair of Chls appears to be the primary donor in most PSI complexes, not P700.

A20: Yes, the distance should be important if we stand on the interpretation that the pair of Acc and A0 is the primary donor. One can perceive the approximate distance between Acc/A-1 and A0 and presume that the distance is similar to that in *T. elongatus* PSI by viewing Fig. 4B, C. However, in the revised manuscript, we decided not to present such statement and to remove the description indicated in this comment because Reviewer #1 (comment 11) wondered the accuracy of the distance. We remained the distances between P and Acc just for information as follows since relevant studies were present (lines 14-18 in page 11).

The Mg^{2+} in Acc_A and Acc_B are coordinated by water molecules forming hydrogen bonds with Asn588/B and Asn602/A, respectively (Fig. 4B and C). In *T. elongatus* PSI, the methyl ester groups of the Chl *a* in Acc_A and Acc_B affect the charge and spin distributions on P700^{33,47-49}. In *A. marina* PSI, these distances were estimated to be 6.2 Å (Acc_A-P_A) and 6.6 Å (Acc_B-P_B), respectively (Fig. 4D), in the present structure.

C21: Page 10, lines 8-10. This is another example where the authors provide ambiguous information.

Are the three Chls of the trimer missing in *A. marina* PSI or are they just not assigned as Chl *a*?
The assumption is that they are not present, but this is really not clear.

A21: This issue is related to the response to your comment #11. As is written there, we can now only say that the trimer was seemingly absent because of the disordered structure affected by the absence of PsaX. More importantly, *A. marina* intrinsically does not carry so-called red Chls. Therefore, we removed the description relating to red Chl in the revised manuscript since it is not appropriate to discuss the originally absent materials.

C22. Page 11, lines 19-20. Another example. Every pigment in this complex is different from one in *T. elongatus*, so the statement about amino acid changes in proximity to different pigments being different means nothing. Virtually everything should be different. This does not describe how it might be different or if there are patterns of differences—like Hbonding residues for the formyl groups, etc

A22: We agree that every pigment in this complex is different from one in *T. elongatus*. However, the sentence indicated by Reviewer here just described the results without any speculation. Therefore, the sentence is remained in the revised manuscript (lines 4-7 in page 12). Instead, according to the advice of Reviewer, we added descriptions on the hydrogen-bonding network environment around the formyl group of chlorins that we can now identify in Fig. 5 (please see the response to your comment #12) in the revised manuscript as follows (latter half of the second

Responses to Reviewers

paragraph in page 9).

The orientations of the formyl group in P_A (Chl *d'*) and P_B (Chl *d*) were identified by considering the distribution of the cryo-EM density (Fig. 5A and B). No amino acid residues and pigments capable of forming hydrogen bonds with these formyl groups were found in the vicinity of P_A and P_B. This suggests that these formyl groups form hydrogen bonds with the C5 H atom in Chl *d'* and Chl *d*, respectively. Similarly, there were also no amino acid residues and pigments capable of forming a hydrogen bond around the formyl group of Acc (Acc_A and Acc_B) (Fig. 5C and D). However, the orientations of the formyl group of Acc (Acc_A and Acc_B) were altered by the hydrophobic environment caused by Trp (Trp585/B and Trp599/A) when compared with P740.

C23. Page 11, line 21. Considering that some subunits are missing and apparently partially missing from the PSI complexes, making comments about flexibility is meaningless. This could all be due to damage introduced during the isolation. It is unlikely to be the case in fully intact PSI complexes for the reason stated—it would be unproductive for energy transfer.

A23. The sentence does not mention the structure itself of *A. marina*. By our current structure, the flexibility of the photosynthetic machinery was revealed. Therefore, the sentence was modified as follows (lines 7-8 in page 12 in the revised manuscript).

The structure exhibits unexpectedly high flexibility of photosynthetic machinery by comparing other structures of type 1 reaction center.

C24. Page 11, lines 24-25. It is extremely unlikely that this complex, which requires Chl *d* that requires oxygen for its synthesis and evidently arose long after Chl *a*, arose divergently to impact the evolutionary trajectory of the bacterial reaction center (especially homodimeric ones). Any similarities to those other reaction centers almost certainly arose by convergent evolution, which might be informative but not in the way the authors seem to intend. However, because this is just a toss-off statement with no detailed explanation, it is hard to know just what they mean.

A24: Actually, it is hard to rationalize to proposed or discussed this statement based on the result of the structural analysis alone. So, the sentence was removed in the revised manuscript.

C25. Page 12, line 3. Sadly, the full picture for the PSI complex of *A. marina* must also await further studies and a thorough rewriting of this manuscript.

A25: We revised our manuscript thoroughly based on the reviewers' comments. Thank you.

Responces to Reviewers

C26. Page 25, line 25. This statement seems surprising, considering that most Chls in *T. elongatus* PSI are not liganded by water. Do the authors mean that some were H-bonded to the formyl group? How many H-bonded formyl groups were detected? This is extremely important in defining site differences for binding Chl *b* versus Chl *a* in LHC2 and LHC1, and is also important in binding Chl *f*. See Gisriel et al. 2020b.

A26: Most of the water molecules were identified as ligands to “Mg²⁺” in chlorin ring of Chl *d*. The situation seems different from that in Chl *f* (please refer to Gisriel et al. 2020), in which it is speculated that water molecules locate to form the axial coordination between Chls since water molecules were not visible at the resolution (global resolution of 3.19 Å (~3.0 Å of local resolution for important elements)).

Gisriel, C. *et al.* The structure of Photosystem I acclimated to far-red light illuminates an ecologically important acclimation process in photosynthesis. *Sci. Adv.* **6**, eaay6415 (2020).

C27. Figure 4 A. The formyl group of P_A appears to fit the density better in the other orientation. With the current resolution how did the authors determine that the Pheo molecules are Pheo *a* and not Pheo *d*? There is virtually no way to distinguish the difference between a vinyl group and a formyl group in an ESP map at this resolution.

A27: We modified Fig. 4A for more clarity in the revised manuscript. There is no hydrogen-bonding molecule in the vicinity of the formyl group of P_A, and its oxygen atom is considered to be interacting with the hydrogen atom bound to the C5 group.

For the determination of Phe *a* not Phe *d*, please see response to Reviewer’s comment #13.

C28: Extended data Figure 3. Although authors frequently discuss the far-red absorbing properties of Chl *d*, the spectra here show another important aspect: Chl *d* has much enhanced absorbance in the blue compared to Chl *a*. This could also be important in light harvesting.

A28: Of course, the reviewer is correct. However, in this manuscript, we want to focus on the far-red light that would be directly used to drive photochemical reaction. The excited state upon the absorption of blue light will relax to Q_y transition, and then, photochemical reaction will occur. Further, by analogy with the cortex effect within the lichen body, high-energy blue light that would be absorbed by Chls could be diminished within the body of symbiotic host ascidians. We added these informations in our revised manuscript to avoid confusion (lines 2-6 in page 3, and second paragraph in page 4). Additionally, we added spectra of Chl *f*-carrying *Halomiconema hongdechloris* cells (Extended Data Fig. 2; please see response to your comment #5) and its PSI

Responses to Reviewers

(Extended Data Fig. 7) as references for the readers.

(lines 2-6 in page 3 in the revised manuscript)

A. marina was isolated from colonial ascidians, which harbor mainly Chl *a*-type cyanobacteria, resulting in an environment with low visible light and high far-red light. By analogy with the cortex effect within the lichen body⁸, high-energy blue light that would be absorbed by Chls could be diminished within the symbiotic host ascidians.

(second paragraph in page 4)

The far-red light-absorbing Chl, Chl *f*, is also utilized in some cyanobacteria to capture far-red light (Extended Data Fig. 2). In combination with Chl *d*-carrying cyanobacteria, acclimation strategy to the light quality expanding to far-red light region gathers attention⁵⁻⁷. Chl *f* that is only induced under far-red light condition occupies less than ~10% of total Chls in photosystem I (seven Chl *f* and 83 Chl *a* in *Halomicronema hongdechloris* PSI¹⁹) and plays solely a light-harvesting function⁷ without engaging photochemical reaction nor electron transfer chain, which is contrasting to Chl *d* that takes place of special pair responsible to photochemical reaction. Low light energy of far-red light captured by Chl *f* is transferred to Chl *a* by quantum-theoretical uphill process using environmental heat for photochemical reaction²⁰. Therefore, the photochemical reaction of Chl *f*-carrying PSI is definitely the same as other PSI operated by Chl *a*; the photoreaction occurs using ‘‘visible red light’’ by P700. However, Chl *d*-driven photoreaction occurs by low-energy far-red light corresponding to Q_y band of Chl *d*. Chl *d* also absorbs high-energy blue light as well corresponding to Soret band, but the resulting excited state of Chl relaxes to lower excited state corresponding to Q_y level for photochemical reaction. In this sense, Chl *d*-driven system is unique in the photochemical reaction occurring by far-red light level energy.

(Extended Data Fig. 7)

Extended Data Fig. 7: Absorption spectra of *A. marina* PSI trimer and cells. Absorption spectrum of *A. marina* cells (black line) equivalent to 8.1 μg Chl *d* measured with an integrating sphere. Absorption spectrum of PSI trimer (red line) measured at 2.0 μg Chl *d*. Absorption spectrum of PSI from Chl *f*-carrying *Halomicronema hongdechloris* (blue line), which is a courtesy of Prof. Tatsuya Tomo, is also shown for reference. Spectra were normalized by the peak height of the Q_y band.

Responses to Reviewers

C29: Legend, extended data Figure 5. ...were subjected to analysis by SDS-PAGE electrophoresis...

A29: Thank you. We modified the statement accordingly in the revised manuscript.

PSI trimer equivalent to 5 μg Chl *d* was subjected to analysis by SDS-PAGE with (lane 2) or without (lane 3) dithiothreitol.

Responces to Reviewers

Reviewer #4 (Remarks to the Author):

C1: The manuscript “Structure of the far-red light utilizing system I of *Acaryochloris*” by Hamaguchi et al. resolved the structure of Chl d-PSI isolated from *Acaryochloris* using cryo-EM technology. The highlight of this manuscript is that Pheo a is found in PSI and functions as the primary electron acceptor. However, there are detectable Chl a from the purified PSI samples, no assigned position/location and possible roles for them and authors gave very speculative explanation without any supporting data (e.g. statement at page 8, lines 184 – 195), which is not acceptable.

A1: Thank you for your precise evaluation of our manuscript. We agree that we could not determine the binding site of detectable Chl *a* in our *A. marina* PSI structure. We tried to assign Acc as Chl *a* with intensive simulation described in the text (second paragraph in page 11). The simulation was based on the theoretical experiment reported by Saito et al. (2012). Yes, the information on the binding site of Chl *a* and its (their) function is very interesting, we are not able to propose the function without determination of the exact binding site of Chl *a*. Meanwhile, the structure of *A. marina* gives us new insight to the far-red light usage by photochemical reaction.

Responses to your comments follows hereafter.

Saito, K., Shen, J. R. & Ishikita, H. Cationic state distribution over the chlorophyll *d*-containing P D1/PD2 pair in photosystem II. *Biochim. Biophys. Acta - Bioenerg.* **1817**, 1191–1195 (2012).

C2: The total numbers of assigned chlorophylls (including 2 Pheo a) are much lower than the chlorophyll numbers in typical purified PSI from Chl a-cyanobacteria, and it is also much lower than the previous report about purified Chl d-PSI from the same organism (Tomo et al, 2008, JBC), in which there were 97 Chl d, 2 Chl a and 25 alpha-carotene.

A2: As is pointed out by Reviewers #1 (comment #3) and #3 (comment 11), there is a possibility that some pigments located in the outer part of PSI could be released during the sample preparation. However, considering the standard errors among independent 5 preparations (Extended Data Table 7) that is smaller than the previous report (Tomo et al. 2008), there is still possibility that the number of pigments could be actually smaller than other type PSI. One paragraph describing such situation and unassigned Chls compared with *T. elongatus* PSI was added as follows in the revised manuscript (second paragraph in page7).

Although the amount of Chl *d* per Chl *d'* determined by pigment analysis (67.0 ± 0.66 , $n=5$ of independently prepared PSI) and assigned by structural analysis (71 including one or two Chl *a*) in this study is lower than those in previous studies (145 ± 8^{14} and 97.0 ± 11.0^{12}), the semi-stoichiometric amount of Phe *a* (1.92 ± 0.022) that was 0.3 ± 0.2 per reaction center in the previous study ¹² is consistent with the structural analysis in this study. Although local

Responces to Reviewers

resolution of some parts of outer region is somewhat lower, our PSI was stable to keep its integrity for five days (Extended Data Fig. 5). Therefore, we deduce here that the number of Chl associated with PSI is smaller than in other oxygenic phototrophs. Actually, Chls corresponding to Chl-88 and Chl-94 in *T. elongatus* are surely absent in *A. marina* PSI. Unassigned Chls in *A. marina* PSI compared with those in *T. elongatus* PSI are shown by marking N/D in the column of ID number in Extended Data Table 6, and those Chls can be recognized by chlorins in transparent gray in Extended Data Fig. 17. The numbers of Chls in PSI of Chl *f*-carrying *Fischerella thermalis* and *H. hongdechloris* are also smaller than that of *T. elongatus* PSI; 89 (*F. thermalis*)³⁰ and 90 (*H. hongdechloris*)¹⁹ Chls vs 96 Chls in *T. elongatus* PSI. Type I reaction center of *Heliobacterium modesticaldum* carries much smaller amount of Chl species; 60 molecules per reaction center²⁹.

Tomo, T. *et al.* Characterization of highly purified photosystem I complexes from the chlorophyll *d*-dominated cyanobacterium *Acaryochloris marina* MBIC 11017. *J. Biol. Chem.* **283**, 18198–18209 (2008).

There was confused information about the numbers of chlorophylls.

C3-1: Look at the details of numbers of assigned chlorophylls, there are 73 assigned chlorophylls (including Chl d' and Pheo a) in Extended Data Fig 20, but there are 74 assigned chlorophylls in Extended data table 6.

A3-1: The pigment No. 88 in Extended Data Table 6 was incorrect. This pigment was not detected in *A. marina*-PSI structure and removed from the table in the revised manuscript. We apologize for this mistake and thank you for finding this mistake.

C3-2: Look at the pigment analysis HPLC data and the summary table (Extended Data Table 7), per PSI sample gave a total of 70 chlorophylls (67 Chl d + 1 Chl a and 2 Pheo a), even more less than the assigned chlorophylls in the structure.

A3-2: As is pointed out by Reviewer #1 (comment #2), the preparation used for biochemical pigment analysis contains limited amount of PSII, which will slightly increase the amount of pigments per monomer. However, considering the small values of standard errors among five independent preparation, the contribution of contaminating PSII to the amount of pigments should be quite small. Here, cryo-EM structure was obtained through classification by excluding limited amount of contamination such as PSII. Mismatch of pigment values between biochemical and structural analyses is frequently found in the literature. For example, in the article reported the PSI structure of Chl *f*-carrying cyanobacterium, *Fischerella thermalis*, the numbers of Chl species determined by structure and biochemical analyses were 89 and 96 (7.1 ± 0.3 Chl *f* and 89.1 ± 14.1 Chl *a*) molecules per monomer, respectively (Gisriel *et al.* *Sci Adv* 2020). We can discuss the mismatch of pigment numbers between structural and biochemical analyses in our study by referring the values in the report by Gisriel *et al.* (2020), but we think it is not appropriate.

Responses to Reviewers

Instead, related description was added in the revised manuscript as transcribed in A2 above.

C4: Extend data table 6, Chl 5 and 6 should read as pheo a, and it said “ assigned Chl d are numbered according to the ...” which is not correct statement here because Chl 5 and Chl 6 are pheo a, not Chl d.

A4: We modified Extend Data Table 6 in the revised manuscript. “Chl” in the table was changed to “Pigment”.

C5: In the same table, N/A (not assigned) caused confusion. Authors stated that “Not assigned” is due to the low resolution, which means those chlorophylls might be there, but cannot be detected/assigned here due to the limit of the resolution (?). However, authors confirmed the Chl d-PSI contains less Chl reflecting the optimization for efficient use of far-red light. i.e. these N/A chlorophylls represent “not detected” instead of “not assigned”.

A5: We modified Extend Data Table 6 in the revised manuscript. “Not assigned (N/A)” in the table was changed to “Not detected (ND)”. And, according to the advice by Reviewer #1 (comment #2 and #3), we added discussion on the low number of pigments in the current structure of *A. marina* PSI as shown above (A3).

C6: There was confused information about assignment for Chl d and Chl d’.

Extend Data Fig. 1 gives explanation the difference between Chl d and Chl d’. In Chl d, the oxygen atom of formyl group is toward C5 side.

However, the 3D model of Chl d and Chl d’ (Fig 4A) demonstrated the same orientation of formyl group. The same 3D maps showed in extended data Fig 14.

Which 3D model map was correct for Chl d’?

A6: We did not include the figure of Chl d’ in Extend Data Fig. 1 in the previous manuscript. Now, we added a figure of Chl d’ compared with Chl d (Extend Data Fig. 1, B and C). Chl d’ has a different orientation of the methyl ester group compared with Chl d, not formyl group at C3. The orientations of the formyl group in P_A (Chl d’) and P_B (Chl d) were identified by considering the distribution of the cryo-EM density that is shown in new Fig. 5A and B in the revised manuscript. Explanation on this point was also added in the text as follows (from line 13 in page 9).

The orientations of the formyl group in P_A (Chl d’) and P_B (Chl d) were identified by considering the distribution of the cryo-EM density (Fig. 5A and B). No amino acid residues and pigments capable of forming hydrogen bonds with these formyl groups were

Responses to Reviewers

found in the vicinity of P_A and P_B . This suggests that these formyl groups form hydrogen bonds with the C5 H atom in Chl d' and Chl d , respectively. Similarly, there were also no amino acid residues and pigments capable of forming a hydrogen bond around the formyl group of Acc (Acc_A and Acc_B) (Fig. 5C and D). However, the orientations of the formyl group of Acc (Acc_A and Acc_B) were altered by the hydrophobic environment caused by Trp (Trp585/B and Trp599/A) when compared with P740.

Fig. 5. Each cryo-EM density maps around P740 (A, B), Acc (C, D) and A_0 (E, F). Each map is shown in a mesh representation at 3.0 sigma contour level for panels A, B, E, and F, and 5.0 sigma for panels C and D contour level for the ease of recognition of water molecule coordinating to Mg^{2+} of chlorin.

C7: The inconsistent data: Extend data table 6 demonstrate N/A number 86 chlorophyll and an assigned the number 88 Chl, but in the Extended data Fig 19, number 88 was shown as ‘transparent’. Which one is missed from Chl d-PSI?

A7: The number 88 Chl was not detected in the *A. marina*-PSI structure. We modified the table accordingly. We apologize for this mistake and thank you for finding this mistake.

C8: About presence of Chl a, where are they? Are they from uncoupled contamination in the sample? There were undefined peptides in the region of 25 -45 kDa (in Extend data Fig 6, lane 2 and 3), the contamination of PSII or other chlorophyll-binding protein complexes? Authors should have the mass data for these undefined peptides to exclude the possible sample contaminations.

A8: Unfortunately, we do not have mass data for the protein bands noted by Reviewer. Those pale bands suggest a slight contamination of PSII. However, the constant presence of Chl *a* can not be

Responces to Reviewers

explained by the plausible contamination of PSII considering the small standard errors of the amount of pigments among independent five preparations. Based on the orientation of the C3 group of Acc, Acc could be Chl-*a*, but we were not able to definitely determine Acc to be Chl *a* (we identified Acc as Chl-*d*) nor the location of Chl *a* in other sites in this study.

C9: Pheo *a* in the purified PSI seem identified unambiguously. However, detected Chl *a*, and also much less numbers of assigned Chl *d* raised question about the quality of isolated PSI? are they intact PSI complexes? Which analysis authors used to confirm that no contamination in the “purified PSI” samples?

A9: As is written in the response to previous comment (C8), limited contamination of PSII could be possible. However, if PSII is contaminated, then, the numbers of pigments per Chl *d* should become larger. There is a possibility that some pigments were released during the sample preparation. On the other hand, comparing the standard error values of this work (67.0 ± 0.66) and those of previous publication (97.0 ± 11.0) (Tomo et al. JBC 2008) for Chl *d*, there still remains possibility that the number of Chl *d* could be actually smaller in *A. marina* PSI than that of other Chl-*a* based PSIs. One paragraph describing such situation and undetected Chls compared with *T. elongatus* PSI was added in the revised manuscript (second paragraph in page7) as transcribed in the response A2.

Cryo-EM structure was obtained through classification excluding plausible PSII contamination. Therefore, the limited amount of PSII contamination did not disturb to build structural model shown in this work.

Tomo, T. *et al.* Characterization of highly purified photosystem I complexes from the chlorophyll *d*-dominated cyanobacterium *Acaryochloris marina* MBIC 11017. *J. Biol. Chem.* **283**, 18198–18209 (2008).

REVIEWER

COMMENTS

Reviewer #1 (Remarks to the Author):

The authors have revised the manuscript removing most of the incorrect statements that I indicated before. They also added some text regarding Chl f-containing cyanobacteria, but the text is not always clear.

Line 59 – “With the acquisition of Chl d, *A. marina* harnesses far-red light, which is the 60 lowest energy light that can be used for the oxygenic photosynthetic reaction”. This statement is misleading as explained by the other referees and me already. Chl f containing cyanobacteria use light of longer wavelengths.

Line 85 “quantum-theoretical uphill process”. This does not mean anything
 Lines 86-87 “Therefore, the photochemical reaction of Chl f-carrying PSI is definitely the same as other PSI operated by Chl a; the photoreaction occurs using “visible 88 red light’ by P700”. This is a matter of debate; there are different proposals in the literature.
 Lines 154-156 “These differences help explain the unique features of light harvesting in *A. marina* through theoretical calculations using the present structure or spectroscopic analysis using mutants and so on.” This sentence needs to be reformulated

Reviewer #2 (Remarks to the Author):

The authors have addressed my previous questions and the manuscript has been improved during the revision. The quality of cryo-EM map in the AmPSI core region is good, consistent with the claimed resolution of 2.5 Å and the structural model fits well with the map. Overall, the structure of the Chl d-rich and pheophytin a-containing PSI from *A. marina* is novel and will be helpful for further mechanistic studies if published.

Comments/suggestions for further revision:

1. The manuscript writing needs to be polished further to improve the clarity and readability. It will be easier for the readers to understand the authors’ points if the long sentences can be rewritten into shorter ones. Subjective words like “definitely” and “unique” should be avoided or replaced with more objective ones.

2. The authors are encouraged to search in the map and model for hydrogen-bond donors (amino acid residues or water molecule) around the C3-formyl groups of peripheral Chl d molecules in AmPSI (beyond those in the electron transfer chains). Basing on the map and structural model provided by the authors, the following are a list of such cases for the authors to analyze further and discuss the adaptation of protein environment in AmPSI for binding and selecting Chl d molecules (instead of Chl a).
 1) Asn461 (chain A) form a hydrogen bond with C3-formyl group of Chl d1206, whereas the corresponding residue in *T. elongatus* PSI (TePSI, PDB: 1jb0) is a Thr.
 2) His 57 form hydrogen bond with C3-formyl group of Chl d1116. Although this His residue is conserved in TePSI, the side chain orientation of His57 is rotated to an orientation more favorable for hydrogen bonding with Chl d1116 in AmPSI.
 3) Trp360 form a hydrogen bond with C3-formyl group of Chl d1117, while the corresponding residue is Met in TePSI.

4) Ser507 form hydrogen bond with C3-formyl group of Chl d1138, whereas it becomes Ala in T. elongatus PSI.

5) Besides the water molecule (W3) forming a hydrogen bond with C3-formyl group of Chl d1139 (Chl d32 as shown in Fig. 3B and C), there is another water-like density (uninterpreted) forming hydrogen bond with C3-formyl group of Chl d1144 and Chl d1503. This additional water molecule is further linked to Ser65 of Chain aL (corresponding to Leu65 in chain L of TePSI) through hydrogen bond.

6) Trp50 appears to form hydrogen bond with C3-formyl group of Chl d1152, while TePSI has a Trp at similar position but is at much larger distance (4.2 Å) from the C3-vinyl group of the Chl a nearby. It will be great if the author could go through the structure more carefully to find all the potential hydrogen bond donors for the C3-formyl groups of Chl d molecules, and add a new column in the ED Table6 to list them.

3. The cryo-EM map of AmPSI does show clear evidences supporting that A0A and A0B are both occupied by pheophytin (without a central Mg²⁺). Curiously, the amino acid residue at the axial position of A0B is a Leu (Leu665/B), while that of A0A is a Met (Met686/A). In comparison, the A0A and A0B sites in TePSI are both occupied by Chl a molecules whose central Mg²⁺ ions are both coordinated by Met residues (instead of one Leu and one Met). Therefore, the switch from Chl a to Pheo a in the A0A and A0B sites of AmPSI is probably not dependent (at least not solely) on the difference of its central ligand. In p9 (lines210-214), Please describe the difference of axial amino acid residues at the A0A and A0B sites, and it will be great if they could go further to discuss more about how AmPSI selectively bind Pheo a molecules on the A0A and A0B sites instead of Chl a.

Reviewer #3 (Remarks to the Author):

The revised manuscript of Hamaguchi et al., and the extensive document of responses to four reviewers, reveals a substantially improved manuscript, but one that still presents some problems. Although the magnitude of those problems is smaller, there were nonetheless many points that still remain to be clarified in the text. Specific comments for consideration by the authors to some of their statements in the reviewer comments and then to the manuscript follow.

Comments in response to rebuttal comments

1. Page 3, Response A4. The comments by the authors concerning the functions of Chl *f* are no longer exactly correct. In FRL-PSI complexes, Chl *f* only functions as an antenna pigment. However, in FRL-PSII, there is no evidence that either Chl *d* (only found in FRL-PSII and not in FRL-PSI) or Chl *f* plays a role in electron transfer. Either way, it is either the case that Chl *d* has electron transfer capability outside *Acaryochloris marina*, or that Chl *f* has a role in electron transfer. The authors need to cite the newer studies and get these details correct relative to their own work.
2. Page 5, Response A5. The authors can claim to know that there is precisely 1 ± 0.04 Chl *a* per complex, but no one is going to believe that. It is simply not possible—even with many replicates—to determine that amount of Chl *a* in a complex that contains 70 other pigments to that degree of accuracy. Furthermore, based on earlier comments to Reviewer 1, it is clear that there is some contamination from PSII in their PSI preparations. That may not be a problem because of filtering in cryo-EM, but there is no “filter” to remove contaminating Chls from PSI complexes being used for Chl quantitation. To claim the accuracy they

mention, is completely unrealistic. Furthermore, given that there is some PSII contamination, the source of the Chl *a* in their preparations is actually uncertain to a degree that makes other discussion about roles of specific Chls somewhat moot. Those were points that had previously raised (and addressed) by this reviewer, but when it wasn't clear that there was contamination to an extent that could account for this issue.

What this means in the end is the authors should clearly state that there may be one or two Chl *a* in their PSI preparations, or it could arise from PSII contamination. They will probably claim the latter, but they really can't tell. Because they also can't tell reliably Chl *d* from Chl *a*, they should clearly state that they modeling all Chls except the two pheophytin a molecules as chlorophyll d for the time being. If/when a higher resolution structure becomes available, then they can revisit this point. What is most important is that readers should clearly understand what is and is not supported by the structural and compositional evidence.

3. Page 7, Response A15. The authors seem to be unaware that some far-red Chl *a* molecules have been identified in the PSI complexes of *Synechocystis* 6803. This point doesn't matter much now, since the text has been modified, but this is just to let the authors know that there is information on this point should they wish to revisit this point in their next revision. Toporik et al., The structure of a red-shifted photosystem I reveals a red site in the core antenna. *Nature Communications* **11**, 5279 (2020). <https://doi.org/10.1038/s41467-020-18884-w>
4. Page 13, Response A3. The authors seem to miss the point here. If you can't tell whether something is definitely Chl *a* or Chl *d* you should say so--and you should be up-front about

stating that all Chls except for Pheophytin were modeled as Chl d. This absolutely should be stated in the paper, and the uncertain origin of Chl a should also be made absolutely clear. The authors do not know from HPLC or structure what the pigment content of their PSI complexes is.

5. Page 14, Response A6. The three Fe/S clusters have absolutely nothing to do, directly, with the reduction of NADP⁺. They transfer electrons from phylloquinone (A0) to ferredoxin, which then reduces the flavin of FNR, that finally reduces NADP⁺. The environment and even the potentials of these clusters doesn't matter so much so long as the overall energy trajectory is downhill for electron transfer from phylloquinone to ferredoxin.
6. Page 16, Response A1. The title should at least be modified to correctly name the organism by adding "*marina*." There are *Acaryochloris* sp. strains that do not even use Chl *d*, so this distinction is important.
7. Page 16, Response A3. The authors seem to consider a definition of "reaction center" that is not widely held. The term reaction center usually refers to the entire electron transfer chain and the 10 alpha helices (C-terminal 5 helices of PsaA and PsaB). The term special pair refers the pair of Chls that ultimately is where the photochemically generated hole resides—P740, P700, P800, etc. This should be changed throughout the manuscript.
8. Page 18-19 Response A7. The authors should also cite Gisriel et al. (2020) here, which gives a better assignment of Chl *f* sites in the two structures.

Gisriel, C. J., Wang, J., Brudvig, G. W., and Bryant, D. A. 2020. Opportunities and challenges for assigning cofactors in cryo-EM density maps of chlorophyll-containing proteins. *Commun. Biol.* **3**, 408. doi/10.1038/s42003-020-01139-1

9. Page 23, Response A13. It should be "Mg²⁺-derived density," not density map.
10. Page 26, Response A19. This response, at least to this reviewer, is too simplistic and probably not correct in detail. It is widely assumed, without any real direct evidence, that electrochromic shifts originate from nearby neighboring Chls. However, it is obviously the case that electronic effects from a Chl cation can have effects over quite some distance, and because there are many Chls within a reasonable radius for coupling to occur—most of which are spectroscopically indistinguishable—it is not really possible to know what gives rise to such an electrochromic shift. This becomes knowable when some of the Chls are replaced by Chl *f*, for example, which are not immediately adjacent to P700⁺ but can nevertheless be affected by it at some distance.

Comments on the revised manuscript

1. Page 1, line 1. *Acaryochloris marina*, not *Acaryochloris*.
2. Page 2, line 31. reactions, not reaction
3. Page 2, line 24. ...the so-called special pair, P740, is a...
4. Page 2, line 26. Delete "oxygenic"
5. Page 2, lines 36-37. The arrangement of the 11 subunits... This is actually an overstatement, since the arrangement of the 11 subunits is exactly like it is in all other PSI complexes and details are missing because of low resolution.
6. Page 2, line 40. Photosynthesis can be performed with a single type of reaction center. Oxygenic photosynthesis is powered by two types of reaction centers specifically.

Photosynthesis is a term that describes light-driven CO₂ fixation, and photosystems do not directly participate in CO₂ fixation.

7. Page 3, line 59-60. The statement here is still misleading and not really correct. Chl *f* can productively absorb light to 800 nm for oxygenic photosynthesis, much further into the far-red/near-IR region than Chl *d* does. The statement as written implies that Chl *d* uses the most red-shifted light for oxygenic photosynthesis, but that is not the case. This statement is still false as written, in spite of the comments to reviewer 1. Chl *d* for the moment MAY be the lowest energy pigment that supports photochemistry in oxygenic photosynthesis. However, recent studies have shown that PSII in FRL utilizing organisms use a trap that is shifted by as much as P740, so this statement should be changed and that work cited.
8. Page 3, lines 71-73. ...special pair... not reaction center.
9. Page 4, lines 80 to 84. This run-on sentence is very poorly constructed and its meaning is thus not really clear. Further, as just noted above, either Chl *d* or Chl *f* (or both) participate in the electron transfer reactions of FRL-PSII.
10. Page 4, lines 86 to 88. The authors should cite Kurashov et al. (2019) for this statement, who demonstrated this point and measured the relative quantum yield and action spectrum for Chl *f* function in FRL-PSI.
11. Page 4, line 92. The authors seem to be determined to make this something “unique,” but again as noted above, Chl *d* and/or Chl *f* participate in the photochemical reactions of FRL-PSII, so this is not something “unique.”
12. Page 4, lines 94 to 95. The authors should cite Gisriel et al. (2020) Communications Biology (see above) here.
13. Page 4, lines 97-98. Again, using low energy light is not unique. This sentence should be modified.
14. Page 5, line 111. ..., including the surrounding detergent micelle (Fig. 2).
15. Page 5, line 112. Cite Gisriel et al. (2020) here. The Kato et al. structure did not even include all subunits.
16. Page 5, line 119. Is there even a *psaX* gene in the *A. marina* genome? Not all cyanobacteria have this polypeptide in PSI (or in their genomes).
17. Page 6, lines 124-125. Is an alpha helix a “fold?” Psa27/PsaI is pretty much just a transmembrane alpha helix, isn't it?
18. Page 6, lines 132. Considering that 71 pigments would be vastly smaller than the Chl content of PSI complexes containing Chl *a* or Chl *a* and Chl *f*, how many of the ligands for pigments that are “missing” in this structure are actually conserved? That might provide some indication of which are missing because of isolation and detergent extraction versus missing due to adaptation.
19. Page 6, line 134. The authors should specify the Chl type throughout. The pheophytin is pheophytin *a*, not pheophytin *d*, and the former is a derivative of Chl *a*, not Chl *d*. Chemically, this detail matters.
20. Page 6-7, lines 148-149. Several previous papers have discussed that Chl 95 probably was assigned in error in *T. elongatus* by Jordan et al. This Chl does not occur in multiple other cyanobacterial structures, for example *Synechocystis* 6803.
21. Page 8, line 173-174. Delete “bringing slight disorder”. The disorder is not slight. Segments of the backbone cannot be modeled, and entire subunits were modeled only as poly-alanine. That is more than “slight disorder.”

22. Page 8, lines 178-180. There is a discrepancy here. The authors state that the electron transfer cofactors include four Chls (3 Chl *d* and 1 Chl *d'*), but in the discussion that follows it becomes clear that they cannot actually distinguish Chl *a* from Chl *d*. They may assign all for now as Chl *d*, but that is not established, and it should be clear that it is not established.
23. Page 8, line 182. Special pair, not reaction center.
24. Page 9, line 207. Delete "map." There is no density at the position of Mg in the map...
25. Page 9, line 209. ...a derivate of Chl *a*, ...
26. Page 10, line 232. Delete "precise"
27. Page 10, line 233. Should read: "Why is Pheo *a* the primary acceptor, A₀, in ...
28. Page 11, line 251. Could or should but not should
29. Page 11, line 259. See #21 above. Here is where the authors indicate that they can't really distinguish what Chl is present from the structure, but they were assigned as Chl *d*.
30. Page 11, line 265-266. An electrochromic shift may be suggestive but is not proof of anything. Chl *a* can clearly have absorbance beyond 700 nm as it does for all "far-red" Chls in PSI complexes that do not contain Chl *d* at all.
31. Page 12, lines 280-281. The Fe/S clusters of PSI do not have any direct role in the reduction of NADP⁺ as stated. They transfer electrons from phyloquinone to soluble ferredoxin which reduces the flavin of FNR that finally reduces NADP⁺. While the surrounding structure may contribute somewhat to the redox potentials of these clusters, this has only an indirect role in NADP⁺ reduction.
32. Page 12, lines 286-287. How could flexibility optimize photochemistry? The observed flexibility in the structure is mostly due to damage during isolation. If the authors mean the changes in structure compared to other PSI complexes reflect adaptations to allow efficient binding and function with Chl *d*, then this should be stated more clearly.
33. Page 12, line 289. Replace "the architecture" with "a basis"
34. Page 12, line 290. Structures for PSII are known. What the authors mean, I believe, is a PSII complex containing Chl *d* (although it might also come from the structure of FRL-PSII containing Chl *d* and Chl *f*).
35. Figure 1. The Acc Chls were not unambiguously identified from the structure according to the text. That should be made clear in the figure legend and text as noted above.
36. Extended data, Figure 7. It is important to note that the PSI complexes from *H. hongdechloris* were isolated from cells grown in far red light, the only condition under which Chl *f* is synthesized.
37. Extended data Table 1. What role does PsaE play in electron transfer, other than possibly ferredoxin docking? It has not electron transfer cofactors.

Reviewer #4 (Remarks to the Author):

The revised manuscript is improved. The highlight of this article is A₀ acceptor is pheophytin a. The structure data supports this claim.

The weakness of this manuscript is "possible" incomplete PSI structure. With the resolution of 2.48Å, authors should be able to fully define the chlorophylls.

1. there are significant less number of chlorophylls in PSI, which is so different from previous reports on Chl *d*-PSI system isolated from *Acaryochloris*. For example, Hu et al 1998, Tomo et al 2008. Hastings et al 2008... None of previous purified PSI studies indicates the "less number of Chl in PSI".

2. dubious presence of Chl a. The sample may be contaminated by PSII, resulting the inconsistency between HPLC analysis and the structural assignment.

3. if authors propose the H-bonded formyl groups (due to presence of water molecules), spectrum of this chlorophyll may be shifted, maybe, showing the spectral profiles similar to C3-hydroxymethyl-Chl a, or reduced Chl d?. Both of them have the similar HPLC profiles as Chl a if you use reverse-phase HPLC for pigment analysis.

Better quality PSI complex is needed to address the questions and remove any ambiguous structural assignments.

Reviewer #1:

C1: The authors have revised the manuscript removing most of the incorrect statements that I indicated before. They also added some text regarding Chl *f*-containing cyanobacteria, but the text is not always clear.

C2: Line 59 – “With the acquisition of Chl *d*, *A. marina* harnesses far-red light, which is the lowest energy light that can be used for the oxygenic photosynthetic reaction”. This statement is misleading as explained by the other referees and me already. Chl *f* containing cyanobacteria use light of longer wavelengths.

A2: The sentence was changed as follows (line 10-12 in page 3):

With the acquisition of Chl *d*, *A. marina* harnesses far-red light, the energy of which is lower than that of visible light that is utilized in most oxygenic photosynthetic reactions.

C3: Line 85 “quantum-theoretical uphill process”. This does not mean anything

A3: The sentence now reads (lines 13-16 in page 4):

Far-red light captured by Chl *f* in the PSIs does not have enough energy to drive the photochemical reaction with the standard special pair of Chls *a*, and the uphill process is believed to overcome the shortage by utilizing heat energy from its surroundings.

C4: Lines 86-87 “Therefore, the photochemical reaction of Chl *f*-carrying PSI is definitely the same as other PSI operated by Chl *a*; the photoreaction occurs using “visible red light’ by P700”. This is a matter of debate; there are different proposals in the literature.

A4: Yes, there are different proposals that Chl *f* should locate close to or potentially within the charge-separating pigments, however, a recent cryo-EM analysis and other studies showed that Chl *f* is not included in charge-separating pigments including and following the paired Chls (the special pair). We modified the text as follows (lines 5-11 in page 4):

Chl *f* is only induced under far-red light conditions, and even then makes up less than ~10% of total Chls in PSI (seven Chl *f* and 83 Chl *a* in *Halomicronema hongdechloris* PSI¹⁹), and the locations of these Chl *f* molecules is still debated^{20,21}. Recent cryo-EM analyses show that Chl *f* is not part of the charge-separating pigments, including and following the special pair^{19,22,23}. Thus, Chl *f* in these PSIs is likely entirely responsible for light harvesting. High-resolution²⁴ and ultra-fast²⁵ spectroscopic studies support this.

19. Kato, K. *et al.* Structural basis for the adaptation and function of chlorophyll *f* in photosystem I. *Nat. Commun.* **11**, 238 (2020).

20. Kaucikas, M., Nürnberg, D., Dorlhiac, G., Rutherford, A. W. & van Thor, J. J. Femtosecond Visible Transient Absorption Spectroscopy of Chlorophyll *f*-Containing Photosystem I. *Biophys. J.* **112**, 234–249 (2017).
21. Nürnberg, D. J. *et al.* Photochemistry beyond the red limit in chlorophyll *f*-containing photosystems. *Science* **360**, 1210–1213 (2018).
22. Gisriel, C. *et al.* The structure of Photosystem I acclimated to far-red light illuminates an ecologically important acclimation process in photosynthesis. *Sci. Adv.* **6**, eaay6415 (2020).
23. Gisriel, C. J., Wang, J., Brudvig, G. W. & Bryant, D. A. Opportunities and challenges for assigning cofactors in cryo-EM density maps of chlorophyll-containing proteins. *Commun. Biol.* **3**, 408 (2020).
24. Kurashov, V. *et al.* Energy transfer from chlorophyll *f* to the trapping center in naturally occurring and engineered Photosystem I complexes. *Photosynth. Res.* **141**, 151–163 (2019).
25. Cherepanov, D. A. *et al.* Evidence that chlorophyll *f* functions solely as an antenna pigment in far-red-light photosystem I from *Fischerella thermalis* PCC 7521. *Biochim. Biophys. Acta - Bioenerg.* **1861**, 148184 (2020).

C5: Lines 154-156 “These differences help explain the unique features of light harvesting in *A. marina* through theoretical calculations using the present structure or spectroscopic analysis using mutants and so on.” This sentence needs to be reformulated

A5: The sentence was rewritten as follows (line 17-19 in page 7):

These differences **when compared with the structure of *T. elongatus*** help explain the **specific** features of light harvesting in *A. marina*.

Reviewer #2:

C0: The authors have addressed my previous questions and the manuscript has been improved during the revision. The quality of cryo-EM map in the AmPSI core region is good, consistent with the claimed resolution of 2.5 Å and the structural model fits well with the map. Overall, the structure of the Chl d-rich and pheophytin a-containing PSI from *A. marina* is novel and will be helpful for further mechanistic studies if published.

Comments/suggestions for further revision:

C1. The manuscript writing needs to be polished further to improve the clarity and readability. It will be easier for the readers to understand the authors' points if the long sentences can be rewritten into shorter ones. Subjective words like “definitely” and “unique” should be avoided or replaced with more objective ones.

A1: The ms has been revised and edited professionally

C2. The authors are encouraged to search in the map and model for hydrogen-bond donors (amino acid residues or water molecule) around the C3-formyl groups of peripheral Chl d molecules in AmPSI (beyond those in the electron transfer chains). Basing on the map and structural model provided by the authors, the following are a list of such cases for the authors to analyze further and discuss the adaptation of protein environment in AmPSI for binding and selecting Chl d molecules (instead of Chl a).

- 1) Asn461 (chain A) form a hydrogen bond with C3-formyl group of Chl d1206, whereas the corresponding residue in *T. elongatus* PSI (TePSI, PDB: 1jb0) is a Thr.
- 2) His 57 form hydrogen bond with C3-formyl group of Chl d1116. Although this His residue is conserved in TePSI, the side chain orientation of His57 is rotated to an orientation more favorable for hydrogen bonding with Chl d1116 in AmPSI.
- 3) Trp360 form a hydrogen bond with C3-formyl group of Chl d1117, while the corresponding residue is Met in TePSI.
- 4) Ser507 form hydrogen bond with C3-formyl group of Chl d1138, whereas it becomes Ala in *T. elongatus* PSI.
- 5) Besides the water molecule (W3) forming a hydrogen bond with C3-formyl group of Chl d1139 (Chl d32 as shown in Fig. 3B and C), there is another water-like density (uninterpreted) forming hydrogen bond with C3-formyl group of Chl d1144 and Chl

d1503. This additional water molecule is further linked to Ser65 of Chain aL (corresponding to Leu65 in chain L of TePSI) through hydrogen bond.

- 6) Trp50 appears to form hydrogen bond with C3-formyl group of Chl d1152, while TePSI has a Trp at similar position but is at much larger distance (4.2 Å) from the C3-vinyl group of the Chl a nearby.

It will be great if the author could go through the structure more carefully to find all the potential hydrogen bond donors for the C3-formyl groups of Chl d molecules, and add a new column in the ED Table6 to list them.

A2: Thank you for your suggestions and the details. We added information of hydrogen bonds surrounding the C3-formyl group of Chls in Supplementary Table 6 and Supplementary Fig. 19. However, at present, it is unclear how the protein environment around Chls affects the function of the *A. marina* PSI and theoretical calculations will be crucial for such understanding. We added the following sentence in our revised manuscript (line 19-22 in page 7).

Additionally, the C3-formyl groups of some Chl *d* in the *A. marina* PSI form hydrogen bonds with their surrounding amino acid residues (Supplementary Table 6, Supplementary Fig. 19). These unique structural features will be important in future theoretical studies of the light harvesting mechanism in *A. marina*.

C3. The cryo-EM map of AmPSI does show clear evidences supporting that A0A and A0B are both occupied by pheophytin (without a central Mg²⁺). Curiously, the amino acid residue at the axial position of A0B is a Leu (Leu665/B), while that of A0A is a Met (Met686/A). In comparison, the A0A and A0B sites in TePSI are both occupied by Chl a molecules whose central Mg²⁺ ions are both coordinated by Met residues (instead of one Leu and one Met). Therefore, the switch from Chl a to Pheo a in the A0A and A0B sites of AmPSI is probably not dependent (at least not solely) on the difference of its central ligand. In p9 (lines210-214), Please describe the difference of axial amino acid residues at the A0A and A0B sites, and it will be great if they could go further to discuss more about how AmPSI selectively bind Pheo a molecules on the A0A and A0B sites instead of Chl a.

A3: Yes, the selection of Pheo *a* at the A0 sites (A0A and A0B) is very interesting. However, at present, we think it is too speculative to discuss the basis for selection. We modified the pertinent part as follows, incorporating sentences from your comment (line 9-15 in page 10).

The present study reveals that the axial amino acid residues at the A_{0A} and A_{0B} sites are Met686/A and Leu665/B, respectively. In comparison, the A_{0A} and A_{0B} sites in *T. elongatus* PSI are both occupied by Chl *a* molecules whose central Mg²⁺ ions are both coordinated by Met residues^{1,42}. At present, it is unclear how the *A. marina* PSI selectively binds Pheo *a* at the A₀ sites instead of other pigments (Chl *a* or Chl *d*). However, the switch from Chl *a* to Pheo *a* in the A_{0A} and A_{0B} sites of *A. marina* PSI is probably not entirely dependent on the difference in its central ligand.

1. Jordan, P. *et al.* Three-dimensional structure of cyanobacterial photosystem I at 2.5 Å resolution. *Nature* **411**, 909–917 (2001).
42. Itoh, S. *et al.* Unidirectional Electron Transfer in Chlorophyll *d*-Containing Photosystem I Reaction Center Complex of *Acaryochloris marina*. in *Photosynthesis. Energy from the Sun* 93–96 (2008). doi:10.1007/978-1-4020-6709-9_21.

Reviewer #3:

C0: The revised manuscript of Hamaguchi et al., and the extensive document of responses to four reviewers, reveals a substantially improved manuscript, but one that still presents some problems. Although the magnitude of those problems is smaller, there were nonetheless many points that still remain to be clarified in the text. Specific comments for consideration by the authors to some of their statements in the reviewer comments and then to the manuscript follow.

Comments in response to rebuttal comments

C1: Page 3, Response A4. The comments by the authors concerning the functions of Chl *f* are no longer exactly correct. In FRL-PSI complexes, Chl *f* only functions as an antenna pigment. However, in FRL-PSII, there is no evidence that either Chl *d* (only found in FRL-PSII and not in FRL-PSI) or Chl *f* plays a role in electron transfer. Either way, it is either the case that Chl *d* has electron transfer capability outside *Acaryochloris marina*, or that Chl *f* has a role in electron transfer. The authors need to cite the newer studies and get these details correct relative to their own work.

A1: Thank you. we rewrote the pertinent part as follows (second paragraph in page 4).

The far-red light-absorbing Chl, Chl *f*, is also utilized in some cyanobacteria to capture far-red light (Supplementary Fig. 2). **How Chl *f*, as well as Chl *d*, has succeeded in expanding harvesting the low energy far-red light region is of great interest**⁵⁻⁷. Chl *f* is only induced under far-red light conditions, **and even then makes up** less than ~10% of total Chls in PSI (seven Chl *f* and 83 Chl *a* in *Halomicronema hongdechloris* PSI¹⁹), and **the** locations of these Chl *f* molecules **is still debated**^{20,21}. **Recent cryo-EM analyses show that Chl *f* is not part of the charge-separating pigments, including and following the special pair**^{19,22,23}. Thus, Chl *f* in these PSIs is likely **entirely** responsible for light harvesting. High-resolution²⁴ and ultra-fast²⁵ spectroscopic studies support this. **Still, PSII in some cyanobacteria may use Chl *f* (and/or Chl *d*) for electron transfer when grown under far-red light conditions**^{21,26}. Far-red light captured by Chl *f* in the PSIs does not have enough energy to drive the photochemical reaction with the standard special pair of Chls *a*, and the uphill process is believed to overcome the shortage by utilizing heat energy from its surroundings. In contrast, Chl *d* in *A. marina* PSI is always **induced even under natural white light and does, in fact, take the place of the special pair Chls in the** photochemical reaction **chain**. The Chl *d*-driven photoreaction occurs by low-energy far-red light corresponding to **the** Qy band of Chl *d*. Chl *d* also absorbs high-energy blue light, corresponding to **the** Soret band, but the resulting excited state relaxes to **a** lower excited state corresponding to **the** Qy level for **the** photochemical reaction. In this sense, **the** Chl *d*-driven PSI-system is **unique among** photochemical reaction **centers, utilizing as it does** far-red light level energy.

5. Chen, M. *et al.* A red-shifted chlorophyll. *Science* **329**, 1318–1319 (2010).

6. Gan, F. *et al.* Extensive remodeling of a cyanobacterial photosynthetic apparatus in far-red light. *Science* **345**, 1312–1317 (2014).
7. Miyashita, H. Discovery of Chlorophyll *d* in *Acaryochloris marina* and Chlorophyll *f* in a Unicellular Cyanobacterium, Strain KC1, Isolated from Lake Biwa. *J. Phys. Chem. Biophys.* **4**, 149 (2014).
19. Kato, K. *et al.* Structural basis for the adaptation and function of chlorophyll *f* in photosystem I. *Nat. Commun.* **11**, 238 (2020).
20. Kaucikas, M., Nürnberg, D., Dorlhiac, G., Rutherford, A. W. & van Thor, J. J. Femtosecond Visible Transient Absorption Spectroscopy of Chlorophyll *f*-Containing Photosystem I. *Biophys. J.* **112**, 234–249 (2017).
21. Nürnberg, D. J. *et al.* Photochemistry beyond the red limit in chlorophyll *f*-containing photosystems. *Science* **360**, 1210–1213 (2018).
22. Gisriel, C. *et al.* The structure of Photosystem I acclimated to far-red light illuminates an ecologically important acclimation process in photosynthesis. *Sci. Adv.* **6**, eaay6415 (2020).
23. Gisriel, C. J., Wang, J., Brudvig, G. W. & Bryant, D. A. Opportunities and challenges for assigning cofactors in cryo-EM density maps of chlorophyll-containing proteins. *Commun. Biol.* **3**, 408 (2020).
24. Kurashov, V. *et al.* Energy transfer from chlorophyll *f* to the trapping center in naturally occurring and engineered Photosystem I complexes. *Photosynth. Res.* **141**, 151–163 (2019).
25. Cherepanov, D. A. *et al.* Evidence that chlorophyll *f* functions solely as an antenna pigment in far-red-light photosystem I from *Fischerella thermalis* PCC 7521. *Biochim. Biophys. Acta - Bioenerg.* **1861**, 148184 (2020).
26. Judd, M. *et al.* The primary donor of far-red photosystem II: Chl_{D1} or P_{D2}? *Biochim. Biophys. Acta - Bioenerg.* **1861**, 148248 (2020).

C2: Page 5, Response A5. The authors can claim to know that there is precisely 1 ± 0.04 Chl *a* per complex, but no one is going to believe that. It is simply not possible—even with many replicates—to determine that amount of Chl *a* in a complex that contains 70 other pigments to that degree of accuracy. Furthermore, based on earlier comments to Reviewer 1, it is clear that there is some contamination from PSII in their PSI preparations. That may not be a problem because of filtering in cryo-EM, but there is no “filter” to remove contaminating Chls from PSI complexes being used for Chl quantitation. To claim the accuracy they mention, is completely unrealistic. Furthermore, given that there is some PSII contamination, the source of the Chl *a* in their preparations is actually uncertain to a degree that makes other discussion about roles of specific Chls somewhat moot. Those were points that had previously raised (and addressed) by this reviewer, but when it wasn’t clear that there was contamination to an extent that could account for this issue.

What this means in the end is the authors should clearly state that there may be one or two Chl *a* in their PSI preparations, or it could arise from PSII contamination. They will probably claim the latter, but they really can’t tell. Because they also can’t tell reliably

Chl *d* from Chl *a*, they should clearly state that they modeling all Chls except the two pheophytin a molecules as chlorophyll d for the time being. If/when a higher resolution structure becomes available, then they can revisit this point. What is most important is that readers should clearly understand what is and is not supported by the structural and compositional evidence.

A2: Thank you. We certainly expect that future study will improve the resolution.

Regarding the cofactors and pigments, what is supported by the structural and compositional evidence is described in the third paragraph in page 6 as follows.

The cofactors assigned (Supplementary Fig. 15) are 71 Chl *d* (Supplementary Table 6), one Chl *d'* (an epimer of Chl *d*), 11 α -carotenes (α -Car) but no β -Car (Supplementary Table 7) ^{10,14}, two pheophytins (Pheo) (a derivative of Chl; however, unlike Chl, no Mg²⁺ ion is coordinated by the tetrapyrrole ring), two phylloquinones (PhyQs), three iron–sulfur clusters, two phosphatidylglycerols, one monogalactosyl diacylglycerol, and 84 water molecules. **Two molecules of Pheo were assigned as Pheo *a* on the basis of pigment analysis (Supplementary Table 7).**

10. Swingley, W. D. *et al.* Niche adaptation and genome expansion in the chlorophyll d-producing cyanobacterium *Acaryochloris marina*. *Proc. Natl. Acad. Sci. U. S. A.* **105**, 2005–2010 (2008).

14. Hu, Q. *et al.* A photosystem I reaction center driven by chlorophyll d in oxygenic photosynthesis. *Proc. Natl. Acad. Sci.* **95**, 13319–13323 (1998).

It is generally accepted that one molecule of Chl *a'* can be estimated to be present within PSI containing ~90 Chls (e.g.; Gisriel *et al.* 2020 and Kato *et al.* 2020). Then, please visit Supplementary Fig. 16 that shows a typical chromatogram separating pigments from the current *A. marina* PSI sample. Chl *a* was separated from other pigments to be able to quantify. Further, although it is hard to control the contamination level, the standard errors are not large. Additionally, BN-PAGE analysis showed a stable single band (Supplementary Fig. 5).

It is reported that purified PSII in *A. marina* has 55 ± 7 Chl *d*, 3.0 ± 0.4 Chl *a*, and 17 ± 3 β -carotene ($n = 4$) per two molecules of Pheo *a* (i. e. per PSII) even with the bound light harvesting pigment binding protein PcbC (Tomo *et al.* 2007). Without PcbC, the number of pigments could be ~ 29.6 Chl *d* and 1.9 Chl *a* molecules per two molecules of Pheo *a* (i. e. per PSII) (Allakhverdiev *et al.* 2010, Kurashov *et al.* 2019). Then, considering the electrophoretic profile of BN-PAGE showing single band (Supplementary Fig. 5) and the number of Pheo *a* as two (1.92 ± 0.022) per Chl *d'* in our *A. marina* PSI, it is reasonable to assume that the contribution of Chl *a* in contaminated PSII, if any, to the number of Chl *a* estimated by HPCL analysis was very small.

It is also widely accepted that only a limited amount of Chl *a* is associated with *A. marina* PSI (e.g., Tomo *et al.* 2008). Kumazaki *et al.* (2002) proposed that A₀ could be Chl *a*, and Tomo *et al.* (2008) discussed in detail the location and role of Chl *a* in *A. marina* PSI. Accordingly, we think it is important to try to assign Chl *a* in our *A. marina* PSI structure, and present the results, which could be helpful to readers. The outcome that we could not confidently assign Chl *a* at this stage is scientifically important information. Additionally, we could not definitely determine whether the small amount of Chl *a* in the sample is of PSII or PSI. The Chls in the present structure are modeled assuming all are Chl *d*.

In view of the comments (comment #4 specifically), we have modified the following sentences:

(lines 20-24 in page 6)

In addition, a small amount of Chl *a* (approximately one Chl *a* per Chl *d*') was detected by pigment analysis (Supplementary Table 7). In this study, we assigned the Chls of *A. marina*-PSI as Chl *d*. This is because the amount of Chl *a* in the *A. marina* PSI is minimal, and it is not possible to distinguish between Chl *d* and Chl *a* at 2.5 Å resolution as is described later.

(lines 20-22 in page 19)

Only a small amount of Chl *a* was present in the sample, and so we assigned the Chls of *A. marina* PSI as Chl *d* in this study even though it is not possible to distinguish between Chl *d* and Chl *a* at 2.5 Å resolution.

Gisriel, C. *et al.* The structure of Photosystem I acclimated to far-red light illuminates an ecologically important acclimation process in photosynthesis. *Sci. Adv.* **6**, eaay6415 (2020).

Kato, K. *et al.* Structural basis for the adaptation and function of chlorophyll *f* in photosystem I. *Nat. Commun.* **11**, 238 (2020).

Tomo *et al.* Identification of the special pair of photosystem II in a chlorophyll *d*-dominated cyanobacterium, *Proc. Natl. Acad. Sci. U. S. A.* **104**, 7283–7288 (2007)

Allakhverdiev *et al.* Redox potential of pheophytin *a* in photosystem II of two cyanobacteria having the different special pair chlorophylls *Proc. Natl. Acad. Sci. U. S. A.* **107**, 3924–3929 (2010)

Kurashov, V. *et al.* Energy transfer from chlorophyll *f* to the trapping center in naturally occurring and engineered Photosystem I complexes. *Photosynth. Res.* **141**, 151–163 (2019).

Tomo, T. *et al.* Characterization of highly purified photosystem I complexes from the chlorophyll *d*-dominated cyanobacterium *Acaryochloris marina* MBIC 11017. *J. Biol. Chem.* **283**, 18198–18209 (2008).

Kumazaki, S., *et al.* Energy equilibration and primary charge separation in chlorophyll *d*-based photosystem I reaction center isolated from *Acaryochloris marina*. *FEBS Lett.* **530**, 153–157 (2002).

C3: Page 7, Response A15. The authors seem to be unaware that some far-red Chl a molecules have been identified in the PSI complexes of *Synechocystis* 6803. This point doesn't matter much now, since the text has been modified, but this is just to let the authors know that there is information on this point should they wish to revisit this point in their next revision. Toporik et al., The structure of a red-shifted photosystem I reveals a red site in the core antenna. *Nature Communications* **11**, 5279 (2020). <https://doi.org/10.1038/s41467-020-18884-w>

A3: Thank you. We are well aware of this work and had discussed among ourselves its relevance to our current structural model. However, in the light of the comments by Reviewers, we decided to rather remove the part regarding red-Chls and not discuss it.

C4: Page 13, Response A3. The authors seem to miss the point here. If you can't tell whether something is definitely Chl a or Chl d you should say so--and you should be up-front about stating that all Chls except for Pheophytin were modeled as Chl d. This absolutely should be stated in the paper, and the uncertain origin of Chl a should also be made absolutely clear. The authors do not know from HPLC or structure what the pigment content of their PSI complexes is.

A4: Thank you. The following sentences were added: (lines 21-24 in page 6).

In this study, we assigned the Chls of *A. marina*-PSI as Chl *d*. This is because the amount of Chl *a* in the *A. marina* PSI is minimal, and it is not possible to distinguish between Chl *d* and Chl *a* at 2.5 Å resolution as is described later.

C5: Page 14, Response A6. The three Fe/S clusters have absolutely nothing to do, directly, with the reduction of NADP⁺. They transfer electrons from phylloquinone (A0) to ferredoxin, which then reduces the flavin of FNR, that finally reduces NADP⁺. The environment and even the potentials of these clusters doesn't matter so much so long as the overall energy trajectory is downhill for electron transfer from phylloquinone to ferredoxin.

A5: The text is now modified as follows (lines 19-23 in page 12):

However, cofactors F_X, F_A, and F_B and their surrounding structures in *A. marina* PSI are nearly identical to those in other cyanobacteria (Fig. 3A and Supplementary Fig. 22). This indicates that the reduction potentials are likely at the same level, although the three Fe/S clusters, F_X, F_A, F_B, transfer electrons from PhyQ (A₁) to ferredoxin and the environment does not need to be conserved as long as the overall energy trajectory is downhill.

C6: Page 16, Response A1. The title should at least be modified to correctly name the organism by adding “*marina*.” There are *Acarychloris* sp. strains that do not even use Chl *d*, so this distinction is important.

A6: The specific name has been added throughout.

C7: Page 16, Response A3. The authors seem to consider a definition of “reaction center” that is not widely held. The term reaction center usually refers to the entire electron transfer chain and the 10 alpha helices (C-terminal 5 helices of PsaA and PsaB). The term special pair refers the pair of Chls that ultimately is where the photochemically generated hole resides—P740, P700, P800, etc. This should be changed throughout the manuscript.

A7: Thank you. We have changed “reaction center” to “paired Chls of P740” where appropriate for general readers and “special pair”.

C8: Page 18-19 Response A7. The authors should also cite Gisriel et al. (2020) here, which gives a better assignment of Chl *f* sites in the two structures.

Gisriel, C. J., Wang, J., Brudvig, G. W., and Bryant, D. A. 2020. Opportunities and challenges for assigning cofactors in cryo-EM density maps of chlorophyll-containing proteins. *Commun. Biol.* **3**, 408. doi/10.1038/s42003-020-01139-1

A8: The paper is now cited. (lines 25 in page 4)

C9: Page 23, Response A13. It should be “Mg²⁺-derived density,” not density map.

A9: Corrected. (line 3 in page 10, line 7 in page 19)

C10: Page 26, Response A19. This response, at least to this reviewer, is too simplistic and probably not correct in detail. It is widely assumed, without any real direct evidence, that electrochromic shifts originate from nearby neighboring Chls. However, it is obviously the case that electronic effects from a Chl cation can have effects over quite some distance, and because there are many Chls within a reasonable radius for coupling to occur—most of which are spectroscopically indistinguishable—it is not really possible to know what gives rise to such an electrochromic shift. This becomes knowable when some of the Chls are replaced by Chl *f*, for example, which are not immediately adjacent to P700⁺ but can nevertheless be affected by it at some distance.

A10: Thank you. As replied in A19 in the previous review, our assignment of Chl *d* as Accs was based on published data (Itoh *et al.* 2007). We do take your point that a detailed discussion on precise electrochromic shift is not straightforward, and is certainly beyond the scope of this manuscript. A higher resolution structure will clarify Acc in the future.

Itoh, S. *et al.* Function of chlorophyll *d* in reaction centers of photosystems I and II of the oxygenic photosynthesis of *Acaryochloris marina*. *Biochemistry* **16**, 12473–12481 (2007).

We assigned Acc as Chl *d* because it was not possibly to distinguish Chl *a* and Chl *d*, for now, and this part is modified as follows (lines 6 and 7 in page 12) (which is related to Ac29 below):

Therefore, Acc could not be conclusively identified as Chl *d* or Chl *a* at the present resolution, and we assigned Accs as Chl *d* in this study.

Comments on the revised manuscript

Cc1: Page 1, line 1. *Acaryochloris marina*, not *Acaryochloris*.

Ac1: Corrected (see A6 above).

Cc2: Page 2, line 31. reactions, not reaction

Ac2: Corrected (line 6 in page 2 in the revised manuscript)

Cc3: Page 2, line 24. ...the so-called special pair, P740, is a...

Ac3: Changed to “The paired chlorophyll, so-called special pair, of P740, is a ..” (lines 9-10 in page 2 in the revised manuscript; A7 above):

Cc4: Page 2, line 26. Delete “oxygenic”

Ac4: Deleted (line 9 in page 2 in the revised manuscript)

Cc5: Page 2, lines 36-37. The arrangement of the 11 subunits... This is actually an overstatement, since the arrangement of the 11 subunits is exactly like it is in all other PSI complexes and details are missing because of low resolution.

Ac5: We modified the sentence as follows (lines 12-13 in page 2 in the revised manuscript):

Here we show the architecture of the photosystem I reaction center is composed of 11 subunits and identify key components that help

Cc6: Page 2, line 40. Photosynthesis can be performed with a single type of reaction center. Oxygenic photosynthesis is powered by two types of reaction centers specifically. Photosynthesis is a term that describes light-driven CO₂ fixation, and photosystems do not directly participate in CO₂ fixation.

Ac6: The sentence is now modified as follows (line 17 in page 2 in the revised manuscript):

Photosynthesis is **driven** by two types of photoreaction systems,....

Cc7: Page 3, line 59-60. The statement here is still misleading and not really correct. Chl *f* can productively absorb light to 800 nm for oxygenic photosynthesis, much further into the far- red/near-IR region than Chl *d* does. The statement as written implies that Chl *d* uses the most red-shifted light for oxygenic photosynthesis, but that is not the case. This statement is still false as written, in spite of the comments to reviewer 1. Chl *d* for the moment MAY be the lowest energy pigment that supports photochemistry in oxygenic photosynthesis. However, recent studies have shown that PSII in FRL utilizing organisms use a trap that is shifted by as much as P740, so this statement should be changed and that work cited.

Ac7: The sentence was modified as follows (lines 10-12 in page 3 in the revised manuscript):

With the acquisition of Chl *d*, *A. marina* harnesses far-red light, **the energy of which is lower than that of visible light** that is **utilized in most** oxygenic photosynthetic reactions.

Further, in relation to C1 above, the possible involvement of Chl *d*/Chl *f* in the electron transfer reaction of FRL-PSII is now included by citing relevant papers (lines 6-18 in page 4 in the revised manuscript).

Chl *f* is only induced under far-red light conditions, **and even then makes up** less than ~10% of total Chls in **PSI** (seven Chl *f* and 83 Chl *a* in *Halomicronema hongdechloris* PSI¹⁹), and **the locations of these Chl *f* molecules is still debated**^{20,21}. **Recent cryo-EM analyses show that Chl *f* is not part of the charge-separating pigments, including and following the special pair**^{19,22,23}. **Thus, Chl *f* in these PSIs is likely entirely responsible for light harvesting. High-resolution**²⁴ and ultra-fast²⁵ spectroscopic studies support this. **Still, PSII in some cyanobacteria may use Chl *f* (and/or Chl *d*) for electron transfer when grown under far-red light conditions**^{21,26}. Far-red light captured by Chl *f* in the PSIs does not have enough energy to drive the photochemical reaction with the standard special pair of Chls *a*, and the uphill process is believed to overcome the shortage by utilizing heat energy from its surroundings. In contrast, Chl *d* in *A. marina* PSI is always induced even under natural white light and does, in fact, take the place of the special pair Chls in the photochemical reaction chain.

19. Kato, K. *et al.* Structural basis for the adaptation and function of chlorophyll f in photosystem I. *Nat. Commun.* **11**, 238 (2020).

20. Kaucikas, M., Nürnberg, D., Dorlhac, G., Rutherford, A. W. & van Thor, J. J. Femtosecond Visible Transient Absorption Spectroscopy of Chlorophyll f-Containing Photosystem I. *Biophys. J.* **112**, 234–249 (2017).
21. Nürnberg, D. J. *et al.* Photochemistry beyond the red limit in chlorophyll f-containing photosystems. *Science (80-.)*. **360**, 1210–1213 (2018).
22. Gisriel, C. *et al.* The structure of Photosystem I acclimated to far-red light illuminates an ecologically important acclimation process in photosynthesis. *Sci. Adv.* **6**, eaay6415 (2020).
23. Gisriel, C. J., Wang, J., Brudvig, G. W. & Bryant, D. A. Opportunities and challenges for assigning cofactors in cryo-EM density maps of chlorophyll-containing proteins. *Commun. Biol.* **3**, 408 (2020).
24. Kurashov, V. *et al.* Energy transfer from chlorophyll f to the trapping center in naturally occurring and engineered Photosystem I complexes. *Photosynth. Res.* **141**, 151–163 (2019).
25. Cherepanov, D. A. *et al.* Evidence that chlorophyll f functions solely as an antenna pigment in far-red-light photosystem I from *Fischerella thermalis* PCC 7521. *Biochim. Biophys. Acta - Bioenerg.* **1861**, 148184 (2020).
26. Judd, M. *et al.* The primary donor of far-red photosystem II: Chl_{D1} or P_{D2}? *Biochim. Biophys. Acta - Bioenerg.* **1861**, 148248 (2020).

Cc8: Page 3, lines 71-73. ...special pair... not reaction center.

Ac8: The sentences were modified as follows (lines 20-25 in page 3 in the revised manuscript):

However, the peak wavelength of the light-induced redox difference absorption spectrum of the paired Chls, the so-called special pair, in *A. marina* PSI is longer (740 nm) than that of Chl *a* (700 nm) in the PSI of plants and typical cyanobacteria¹⁴ (Fig. 1A). The special pair P740 have been assumed to be a heterodimer of Chl *d* and *d'*^{15,16} as in P700, which is composed of a heterodimer of Chl *a*/Chl *a'* in all other PSIs (Fig. 1A)^{1,17}

14. Hu, Q. *et al.* A photosystem I reaction center driven by chlorophyll *d* in oxygenic photosynthesis. *Proc. Natl. Acad. Sci.* **95**, 13319–13323 (1998).
15. Akiyama, M. *et al.* Stoichiometries of chlorophyll *d'*/PSI and chlorophyll *a*/PSII in a chlorophyll *d*-dominated cyanobacterium *Acaryochloris marina*, *Jpn. J. Phycol.* **52**, 67–72 (2004).
16. Hastings, G. Time-resolved step-scan Fourier transform infrared and visible absorption difference spectroscopy for the study of photosystem I. *Appl. Spectrosc.* **55**, 894–900 (2001).
17. Kobayashi, M. *et al.* Chlorophyll *a'*/P-700 and pheophytin *a*/P-680 stoichiometries in higher plants and cyanobacteria determined by HPLC analysis. *Biochim. Biophys. Acta - Bioenerg.* **936**, 81–89 (1988).

Cc9: Page 4, lines 80 to 84. This run-on sentence is very poorly constructed and its meaning is thus not really clear. Further, as just noted above, either Chl *d* or Chl *f* (or both) participate in the electron transfer reactions of FRL-PSII.

Ac9: We rewrote this part as is described in A1. The rewritten part is copied here from A1 (second paragraph in page 4 in the revised manuscript):

The far-red light-absorbing Chl, Chl *f*, is also utilized in some cyanobacteria to capture far-red light (Supplementary Fig. 2). How Chl *f*, as well as Chl *d*, has succeeded in expanding harvesting the low energy far-red light region is of great interest⁵⁻⁷. Chl *f* is only induced under far-red light conditions, and even then makes up less than ~10% of total Chls in PSI (seven Chl *f* and 83 Chl *a* in *Halomicronema hongdechloris* PSI¹⁹), and the locations of these Chl *f* molecules is still debated^{20,21}. Recent cryo-EM analyses show that Chl *f* is not part of the charge-separating pigments, including and following the special pair^{19,22,23}. Thus, Chl *f* in these PSIs is likely entirely responsible for light harvesting. High-resolution²⁴ and ultra-fast²⁵ spectroscopic studies support this. Still, PSII in some cyanobacteria may use Chl *f* (and/or Chl *d*) for electron transfer when grown under far-red light conditions^{21,26}. Far-red light captured by Chl *f* in the PSIs does not have enough energy to drive the photochemical reaction with the standard special pair of Chls *a*, and the uphill process is believed to overcome the shortage by utilizing heat energy from its surroundings. In contrast, Chl *d* in *A. marina* PSI is always induced even under natural white light and does, in fact, take the place of the special pair Chls in the photochemical reaction chain. The Chl *d*-driven photoreaction occurs by low-energy far-red light corresponding to the Qy band of Chl *d*. Chl *d* also absorbs high-energy blue light, corresponding to the Soret band, but the resulting excited state relaxes to a lower excited state corresponding to the Qy level for the photochemical reaction. In this sense, the Chl *d*-driven PSI-system is unique among photochemical reaction centers, utilizing as it does far-red light level energy.

5. Chen, M. *et al.* A red-shifted chlorophyll. *Science* **329**, 1318–1319 (2010).
6. Gan, F. *et al.* Extensive remodeling of a cyanobacterial photosynthetic apparatus in far-red light. *Science* **345**, 1312–1317 (2014).
7. Miyashita, H. Discovery of Chlorophyll d in *Acaryochloris marina* and Chlorophyll f in a Unicellular Cyanobacterium, Strain KC1, Isolated from Lake Biwa. *J. Phys. Chem. Biophys.* **4**, 149 (2014).
19. Kato, K. *et al.* Structural basis for the adaptation and function of chlorophyll f in photosystem I. *Nat. Commun.* **11**, 238 (2020).
20. Kaucikas, M., Nürnberg, D., Dorlhiac, G., Rutherford, A. W. & van Thor, J. J. Femtosecond Visible Transient Absorption Spectroscopy of Chlorophyll *f*-Containing Photosystem I. *Biophys. J.* **112**, 234–249 (2017).
21. Nürnberg, D. J. *et al.* Photochemistry beyond the red limit in chlorophyll *f*-containing photosystems. *Science* **360**, 1210–1213 (2018).
22. Gisriel, C. *et al.* The structure of Photosystem I acclimated to far-red light illuminates an ecologically important acclimation process in photosynthesis. *Sci. Adv.* **6**, eaay6415 (2020).
23. Gisriel, C. J., Wang, J., Brudvig, G. W. & Bryant, D. A. Opportunities and challenges for assigning cofactors in cryo-EM density maps of chlorophyll-containing proteins. *Commun. Biol.* **3**, 408 (2020).
24. Kurashov, V. *et al.* Energy transfer from chlorophyll *f* to the trapping center in naturally occurring and engineered Photosystem I complexes. *Photosynth. Res.* **141**, 151–163 (2019).
25. Cherepanov, D. A. *et al.* Evidence that chlorophyll *f* functions solely as an antenna pigment in far-red-light photosystem I from *Fischerella thermalis* PCC 7521. *Biochim. Biophys. Acta - Bioenerg.* **1861**, 148184 (2020).
26. Judd, M. *et al.* The primary donor of far-red photosystem II: Chl_{D1} or P_{D2}? *Biochim. Biophys. Acta - Bioenerg.* **1861**, 148248 (2020).

Cc10: Page 4, lines 86 to 88. The authors should cite Kurashov et al. (2019) for this statement, who demonstrated this point and measured the relative quantum yield and action spectrum for Chl *f* function in FRL-PSI.

Ac10: The work is now cited (lines 11-12 in page 4, reference #24). Thank you.

High-resolution²⁴ and ultra-fast²⁵ spectroscopic studies support this.

24. Kurashov, V. *et al.* Energy transfer from chlorophyll *f* to the trapping center in naturally occurring and engineered Photosystem I complexes. *Photosynth. Res.* **141**, 151–163 (2019).

Cc11: Page 4, line 92. The authors seem to be determined to make this something “unique,” but again as noted above, Chl *d* and/or Chl *f* participate in the photochemical reactions of FRL-PSII, so this is not something “unique.”

Ac11: As replied in A1, we now refer to FRL-PSII with relevant citations (#1 below; lines 12-13 in page 4 in the text) and modified the sentence that used the word “unique” and limit “unique” to only PSI (#2 below; lines 21-23 in page 4 in the text).

#1; Still, PSII in some cyanobacteria may use Chl *f* (and/or Chl *d*) for electron transfer when grown under far-red light conditions^{21,26}.

21. Nürnberg, D. J. *et al.* Photochemistry beyond the red limit in chlorophyll *f*-containing photosystems. *Science* **360**, 1210–1213 (2018).

26. Judd, M. *et al.* The primary donor of far-red photosystem II: Chl_{D1} or P_{D2}? *Biochim. Biophys. Acta - Bioenerg.* **1861**, 148248 (2020).

#2; In this sense, the Chl *d*-driven PSI-system is unique among photochemical reaction centers, utilizing as it does far-red light level energy.

Cc12: Page 4, lines 94 to 95. The authors should cite Gisriel et al. (2020) Communications Biology (see above) here.

Ac12: Cited (see A8 above). Thank you.

Cc13: Page 4, lines 97-98. Again, using low energy light is not unique. This sentence should be modified.

Ac13: The sentence was modified as follows limiting the meaning for only PSI (lines 3-5 in page 5):

The PSI of *A. marina* with Chl *d* in its electron transfer chain for utilizing low energy light directly for the photochemistry is thus unique.

Cc14: Page 5, line 111. ..., including the surrounding detergent micelle (Fig. 2).

Ac15: Changed (lines 18-19 in page 5). Thank you.

The PSI trimer has dimensions of 100 Å depth, 200 Å length, and 200 Å width, including the surrounding detergent micelle (Fig. 2).

Cc15: Page 5, line 112. Cite Gisriel et al. (2020) here. The Kato et al. structure did not even include all subunits.

Ac15: Citation was changed (line 20 in page 5).

1. Jordan, P. *et al.* Three-dimensional structure of cyanobacterial photosystem I at 2.5 Å resolution. *Nature* **411**, 909–917 (2001).
22. Malavath, T., Caspy, I., Netzer-El, S. Y., Klaiman, D. & Nelson, N. Structure and function of wild-type and subunit-depleted photosystem I in *Synechocystis*. *Biochim. Biophys. Acta - Bioenerg.* **1859**, 645–654 (2018).
23. Gisriel, C. J., Wang, J., Brudvig, G. W. & Bryant, D. A. Opportunities and challenges for assigning cofactors in cryo-EM density maps of chlorophyll-containing proteins. *Commun. Biol.* **3**, 408 (2020).
34. Gisriel, C. *et al.* The structure of Photosystem I acclimated to far-red light illuminates an ecologically important acclimation process in photosynthesis. *Sci. Adv.* **6**, eaay6415 (2020).

Cc16: Page 5, line 119. Is there even a *psaX* gene in the *A. marina* genome? Not all cyanobacteria have this polypeptide in PSI (or in their genomes).

Ac16: In the genome of *A. marina*, *psaX* gene is absent as described in the text (lines 1-4 in page 6).

PsaX (a peripheral subunit in other cyanobacteria such as *T. elongatus*), PsaG and PsaH (subunits in higher plants) were missing in the density map, consistent with the absence of their genes from the *A. marina* genome¹⁰.

10. Swingley, W. D. *et al.* Niche adaptation and genome expansion in the chlorophyll *d*-producing cyanobacterium *Acaryochloris marina*. *Proc. Natl. Acad. Sci. U. S. A.* **105**, 2005–2010 (2008).

Cc17: Page 6, lines 124-125. Is an alpha helix a “fold?” Psa27/PsaI is pretty much just a transmembrane alpha helix, isn’t it?

Ac17: The sentence was modified to show that PsaI is a transmembrane alpha helix (lines 5-6 in page 6). Thank you.

Here, we found that Psa27 is in the same location as PsaI, a transmembrane alpha helix, in *T. elongatus*.

Cc18: Page 6, lines 132. Considering that 71 pigments would be vastly smaller than the Chl content of PSI complexes containing Chl *a* or Chl *a* and Chl *f*, how many of the ligands for pigments that are “missing” in this structure are actually conserved? That might

provide some indication of which are missing because of isolation and detergent extraction versus missing due to adaptation.

Ac18: In Supplementary Table 6, such information is presented. The ligands do not match completely even for assigned Chls with corresponding Chl *a* in *T. elongatus*. An example is Phe49/J in *A. marina*, the counterpart of His39/J that is an axial ligand of Chl *a* 88 in *T. elongatus* PSI (Lines 8 - 10 in page 7). As is pointed out by Reviewer, the ligand information can be predicted for the conserved Chls. However, we think such proposals would be too speculative with respect to the pigments that were not detected in the cryo-EM density map. In addition, we decided to mainly focus on the cofactors of electron transfer chain in this manuscript. A future study with higher resolution of the peripheral region will identify the more pigments.

Cc19: Page 6, line 134. The authors should specify the Chl type throughout. The pheophytin is pheophytin *a*, not pheophytin *d*, and the former is a derivative of Chl *a*, not Chl *d*. Chemically, this detail matters.

Ac19: It is not appropriate to specify the type of pheophytin by the density map. At the resolution of 2.5Å, Chl *a* and Chl *d* cannot be distinguished. Similarly, it is not possible to distinguish pheophytin *a* and pheophytin *d*. We just know from the cryo-EM density map that the A₀-chlorins are pheophytins. By HPLC analysis, we determined 2 molecules of pheophytin *a* per Chl *d* with no trace of pheophytin *d*. Then, it becomes possible to assign A₀ as pheophytin *a*. A sentence was added as follows (lines 18-20 in page 6), and this matter is explained in the text (lines 1-6 in page 10).

(lines 18-20 in page 6)

Two molecules of Pheo were assigned as Pheo *a* on the basis of pigment analysis (Supplementary Table 7).

(lines 1-6 in page 10)

Previous studies have suggested that the primary electron acceptor A₀ in *A. marina* PSI (Fig. 1A) is Chl *a*^{13,40}, in line with other species^{1,41}. However, we found that there was no Mg²⁺-derived density at the center of the tetrapyrrole rings of A_{0A} and A_{0B}, but rather a hole in the cryo-EM density map (Fig. 4A; Fig. 5E, F). This indicates, when combined with the result of pigment analysis (Supplementary Table 7 and Supplementary Fig. 16), that A₀ is actually Pheo *a*, a derivative of Chl *a*.

1. Jordan, P. *et al.* Three-dimensional structure of cyanobacterial photosystem I at 2.5 Å resolution. *Nature* **411**, 909–917 (2001).
13. Itoh, S. *et al.* Function of chlorophyll *d* in reaction centers of photosystems I and II of the oxygenic photosynthesis of *Acaryochloris marina*. *Biochemistry* **46**, 12473–12481 (2007).

40. Kumazaki, S., Abiko, K., Ikegami, I., Iwaki, M. & Itoh, S. Energy equilibration and primary charge separation in chlorophyll *d*-based photosystem I reaction center isolated from *Acaryochloris marina*. *FEBS Lett.* **530**, 153–157 (2002).
41. Akiyama, M. *et al.* Quest for minor but key chlorophyll molecules in photosynthetic reaction centers - Unusual pigment composition in the reaction centers of the chlorophyll *d*-dominated cyanobacterium *Acaryochloris marina*. *Photosynth. Res.* **74**, 97–107 (2002).

Cc20: Page 6-7, lines 148-149. Several previous papers have discussed that Chl 95 probably was assigned in error in *T. elongatus* by Jordan *et al.* This Chl does not occur in multiple other cyanobacterial structures, for example *Synechocystis* 6803.

Ac20: We think that you are meaning Chl 94, not Chl 95. Chl 94 and 95 in *T. elongatus* are coordinated to a water molecule and Asn23/PsaX, respectively. In the PSI model of *Synechocystis* sp. PCC6803 (PDB code: 5OY0), Chl that is not assigned is the one corresponding to Chl 94 in *T. elongatus* PSI. On the other hand, Chl1302 (coordinated to Asp54/PsaF) is assigned in *Synechocystis* PSI locating the position close to Chl 95 in *T. elongatus*.

As is pointed out by Reviewer, Chl 94 could be assigned in error in *T. elongatus* PSI in the first publication by Jordan *et al.* (2001). However, it is assigned in *T. elongatus* PSI in the work by Kölsch, Zouni *et al.* (2020). Therefore, in this manuscript, one sentence with two citations (Kölsch *et al.* (2020) and Malavath *et al.* (2018)) were just added (lines 10-13 in page 7).

Owing to the absence of PsaX in *A. marina*, the Chl *d* corresponding to Chl *a* 95^{1,37} in *T. elongatus* whose axial ligand is Asn23/X is also missing. **The absence of Chl *a* 94 is seen in some other reported PSI structures such as *Synechocystis* sp. PCC 6803³⁴.**

1. Jordan, P. *et al.* Three-dimensional structure of cyanobacterial photosystem I at 2.5 Å resolution. *Nature* **411**, 909–917 (2001).
34. Malavath, T., Caspy, I., Netzer-El, S. Y., Klaiman, D. & Nelson, N. Structure and function of wild-type and subunit-depleted photosystem I in *Synechocystis*. *Biochim. Biophys. Acta - Bioenerg.* **1859**, 645–654 (2018).
37. Kölsch, A. *et al.* Current limits of structural biology: The transient interaction between cytochrome *c* and photosystem I. *Curr. Res. Struct. Biol.* **2**, 171–179 (2020).

Cc21: Page 8, line 173-174. Delete “bringing slight disorder”. The disorder is not slight. Segments of the backbone cannot be modeled, and entire subunits were modeled only as poly-alanine. That is more than “slight disorder.”

Ac21: Deleted as follows (lines 12-14 in page 8).

Due to the somewhat lower resolutions of the outer regions of the *A. marina* PSI, we mainly focus on the structure and function of the central part of *A. marina* PSI at high local resolution, that is, P740 and the electron transfer components.

Cc22: Page 8, lines 178-180. There is a discrepancy here. The authors state that the electron transfer cofactors include four Chls (3 Chl *d* and 1 Chl *d'*), but in the discussion that follows it becomes clear that they cannot actually distinguish Chl *a* from Chl *d*. They may assign all for now as Chl *d*, but that is not established, and it should be clear that it is not established.

Ac22: Actually, the possibility of Chl *a* involvement into the four Chls is discussed in a later part. Therefore, we modified the sentence as follows omitting the words in parenthesis (lines 18-20 in page 8):

The important assigned cofactors involved in electron transfer (Fig. 1B) are four Chls, two Pheos, two PhyQs (A_1),

Cc23: Page 8, line 182. Special pair, not reaction center.

Ac23: Changed (line 23 in page 8). Thank you.

The Chls of special pair P740 are Chl *d'* (P_A) and Chl *d* (P_B) (Fig. 4A), which....

Cc24: Page 9, line 207. Delete “map.” There is no density at the position of Mg in the map...

Ac24: Deleted (lines 2-3 in page 10). Thank you.

However, we found that there was no Mg^{2+} -derived density at the center of

Cc25: Page 9, line 209. ...a derivate of Chl *a*, ...

Ac25: This part is modified as follows in relation to the reply to Cc19 (lines 4-6 in page 10):

This indicates, when combined with the result of pigment analysis (Supplementary Table 7 and Supplementary Fig. 16), that A_0 is actually Pheo *a*, a derivative of Chl *a*.

Cc26: Page 10, line 232. Delete “precise”

Ac26: Deleted (lines 22-24 in page 10).

Future theoretical studies using the *A. marina* PSI structure may throw more light on the details of the electron transfer mechanism in relation to the $P_A^{\cdot+}/P_B^{\cdot+}$ ratio.

Cc27: Page 10, line 233. Should read: “Why is Pheo *a* the primary acceptor, A_0 , in ...

Ac27: Changed (line 1 in page 11). Thank you.

Why is Pheo *a* the primary acceptor, A_0 , in *A. marina* PSI?

Cc28: Page 11, line 251. Could or should but not should

Ac28: Corrected (lines 17-18 in page 11). Thank you.

Accordingly, Pheo *a* as A₀ **should** achieve the same electron transfer efficiency as the

Cc29: Page 11, line 259. See #21 above. Here is where the authors indicate that they can't really distinguish what Chl is present from the structure, but they were assigned as Chl *d*.

Ac29: Resolution of the central part where electron transfer chain cofactors reside is high. However, it was difficult to distinguish Chl *a* and Chl *d* even with the resolution as high as 2.5Å and the information obtained from quantum mechanical/molecular mechanical calculations. This is described in the text as follows (lines 23 in page 11 and 6 in page 12).

It was once assumed that A₀ is Chl *a*⁴⁰, but we now know this to be incorrect. One possible **place for Chl *a*** is as the accessory Chls, Acc_A and Acc_B (Figs. 1 and 4). **Unfortunately**, the functional group containing C3¹ of Acc Chl could not be precisely defined from the cryo-EM density map. While quantum mechanical/molecular mechanical calculations show that the formyl group of Chl *d* adopts two orientations (**Supplementary Fig. 1C, D**)⁵², the oxygen atom of the formyl group on the Chl *d* molecule in a vacuum is more stable when oriented towards the C5 H atom, suggesting the conformer in **Supplementary Fig. 1C**. In contrast, the vinyl group of Chl *a* can adopt either orientation.

40. Kumazaki, S., Abiko, K., Ikegami, I., Iwaki, M. & Itoh, S. Energy equilibration and primary charge separation in chlorophyll *d*-based photosystem I reaction center isolated from *Acaryochloris marina*. *FEBS Lett.* **530**, 153–157 (2002).
52. Saito, K., Shen, J. R. & Ishikita, H. Cationic state distribution over the chlorophyll *d*-containing P_{D1}/P_{D2} pair in photosystem II. *Biochim. Biophys. Acta - Bioenerg.* **1817**, 1191–1195 (2012).

We assigned Acc as Chl *d* because it was unable to distinguish Chl *a* and Chl *d*, regarding the published data of difference spectrum of P740 (Itoh *et al.* 2007). Now, we rewrote this part as follows (lines 6 and 7 in page 12).

Therefore, Acc could not be conclusively identified as Chl *d* or Chl *a* at the present resolution, and we assigned Accs as Chl *d* in this study.

Itoh, S. *et al.* Function of chlorophyll *d* in reaction centers of photosystems I and II of the oxygenic photosynthesis of *Acaryochloris marina*. *Biochemistry* **16**, 12473–12481 (2007).

Cc30: Page 11, line 265-266. An electrochromic shift may be suggestive but is not proof of anything. Chl *a* can clearly have absorbance beyond 700 nm as it does for all "far-red" Chls in PSI complexes that do not contain Chl *d* at all.

Ac30: As is pointed out by Reviewer, Chl *a* can have absorbance beyond 700 nm especially when oligomerized as shown by Toporik *et al.* (2020). However, the cited data is the

difference spectrum of P740, not the absorption spectra. The shift-type absorption change at 707 nm was assigned as Acc Chl *d* by Itoh *et al.* (2007).

This part was modified as described above (Ac29).

Toporik, H. *et al.*, The structure of a red-shifted photosystem I reveals a red site in the core antenna, *Nat. Commun.* **11**, 5279 (2020).

Itoh, S. *et al.* Function of chlorophyll *d* in reaction centers of photosystems I and II of the oxygenic photosynthesis of *Acaryochloris marina*. *Biochemistry* **16**, 12473–12481 (2007).

Cc31: Page 12, lines 280-281. The Fe/S clusters of PSI do not have any direct role in the reduction of NADP⁺ as stated. They transfer electrons from phylloquinone to soluble ferredoxin which reduces the flavin of FNR that finally reduces NADP⁺. While the surrounding structure may contribute somewhat to the redox potentials of these clusters, this has only an indirect role in NADP⁺ reduction.

Ac31: In relation to A5, the relevant part was modified as follows in the revised manuscript (lines 21-23 in page 12):

This indicates that the reduction potentials are likely at the same level, although the three Fe/S clusters, F_x, F_A, F_B, transfer electrons from PhyQ (A₁) to ferredoxin and the environment does not need to be conserved as long as the overall energy trajectory is downhill.

Cc32: Page 12, lines 286-287. How could flexibility optimize photochemistry? The observed flexibility in the structure is mostly due to damage during isolation. If the authors mean the changes in structure compared to other PSI complexes reflect adaptations to allow efficient binding and function with Chl *d*, then this should be stated more clearly.

Ac32: To avoid over-speculation, the part was deleted, just leaving the last sentence as follows (lines 25 in page 12 - 3 in page 13):

the divergence is most pronounced around the pigments specifically observed in *A. marina* PSI, for example Chl *d*, α -Car, and Pheo *a*-A₀ (Supplementary Table 5). Again, the alterations must reflect optimization to drive efficient photochemistry utilizing low-energy light.

Cc33: Page 12, line 289. Replace “the architecture” with “a basis”

Ac33: Changed (line 5 in page 13).

pigments provide a basis for understanding how low-energy far-red light is utilized by

Cc34: Page 12, line 290. Structures for PSII are known. What the authors mean, I believe, is a PSII complex containing Chl *d* (although it might also come from the structure of FRL-PSII containing Chl *d* and Chl *f*).

Ac34: Yes, the structure of FRL-PSII containing Chl *d* and Chl *f* will also disclose one type mechanism to utilize low energy light for photochemistry in PSII. Here, we meant the PSII in *A. marina*. Therefore, the part was modified as follows (line 6 in page 13):

However, the full picture must wait for the structure of *A. marina* PSII, **the system**

Cc35: Figure 1. The Acc Chls were not unambiguously identified from the structure according to the text. That should be made clear in the figure legend and text as noted above.

Ac35: It is described that Acc could be Chl *d* in the legend to Fig. 1. To make clearer, the part was modified as follows (bottom of the legend) with addition of type of Pheo:

while Acc could be Chl *d* (**see text for detail**) and A₀ is Pheo *a*.

Cc36: Extended data, Figure 7. It is important to note that the PSI complexes from *H. hongdechloris* were isolated from cells grown in far red light, the only condition under which Chl *f* is synthesized.

Ac36: The part was modified as follows (legend to Supplementary Fig. 7)

Absorption spectrum of PSI **isolated** from **cells of** Chl *f*-carrying *Halomicronema hongdechloris* **grown in far red light** (blue line), which

Cc37: Extended data Table 1. What role does PsaE play in electron transfer, other than possibly ferredoxin docking? It has not electron transfer cofactors.

Ac37: Changed to “Possibly ferredoxin docking” for PsaE in Supplementary Table 1 considering the following papers.

Kubota-Kawai, H. *et al.*, X-ray structure of an asymmetrical trimeric ferredoxin-photosystem I complex, *Nat. Plants* **4**: 218-224 (2018)

Barth, P., Guillouard, I., Sétif, P. and Lagoutte, B., Essential role of a single arginine of photosystem I in stabilizing the electron transfer complex with ferredoxin, *J. Biol. Chem.* **275**: 7030-7036 (2000)

Reviewer #4:

C0: The revised manuscript is improved. The highlight of this article is A0 acceptor is pheophytin a. The structure data supports this claim. The weakness of this manuscript is "possible" incomplete PSI structure. With the resolution of 2.48Å, authors should be able to fully define the chlorophylls.

A0: Thank you for your evaluation again to improve our manuscript. The resolutions of the cryo-EM structure differ from one part to another, and the resolutions of the outer region of PSI are lower than the central parts of PSI (this data is shown in Supplementary Fig. 11 (A)). The situation is similar to other structural models resolved by cryo-EM whose peripheral subunits/pigments were dissociated. This kind of dissociation is sometimes caused, especially in huge protein complexes, by the inevitable technical process that remove stabilizing reagents from media upon applying the sample on the grid, which is different from crystallographic analysis. Examples are PsaK subunit dissociated from the Chl *f*-carrying PSI complex (Kato et al. 2020), peripheral LHCs in PSII-LHC and PSI-LHC complexes (Su et al. 2017; Pi et al. 2018), and peripheral PsbY subunit in *T. vulcanus*-PSII (Umena et al., 2011).

Kato, K. *et al.* Structural basis for the adaptation and function of chlorophyll *f* in photosystem I. *Nat. Commun.* **11**, 238 (2020).

Su, X., *et al.* Structure and assembly mechanism of plant C2S2M2-type PSII-LHCII supercomplex, *Science* **357**: 815-820 (2017).

Pi, X. *et al.* Unique organization of photosystem I–light-harvesting supercomplex revealed by cryo-EM from a red alga. *Proc. Natl. Acad. Sci. U. S. A.* **115**, 4423–4428 (2018).

Umena, Y., Kawakami, K., Shen, J. R. & Kamiya, N. Crystal structure of oxygen-evolving photosystem II at a resolution of 1.9Å. *Nature* **473**, 55–60 (2011).

The matter of resolution was explained in the reply to Reviewer #3 in the previous review, which is reprinted here again.

The cryo-EM map obtained in this study and the validation of the *A. marina*-PSI model refined from the map are explained in Extended Data Fig. 10-12, and Extended Data Table 3 and 4. The definition of resolution of cryo-EM structure is different from that of crystal structure. The resolution of cryo-EM map is determined by the FSC (Fourier Shell Correlation)-based comparison of two half maps by the “gold standard FSC” at 0.143 criterion (Scheres and Chen, Nature Methods, 2012) (Extended Data Fig. 11 (C)). In this study, the refined PSI model was validated by Q-score, measurement of atom resolvability in cryo-EM map (Pintilie et al., Nature Methods, 2020), and its resulting estimated resolution was 2.49 Å (Extended Data Table 4).

As is recognized by reviewer #3, our cryo-EM map has some disordered parts in the outer region. This kind of disordered parts are frequently observed for any cryo-EM map because

the cryo-EM map reflects the structure of the solution state, which is fundamentally different from crystal structure map. Because of this, the resolution of the structure obtained by cryo-EM differs from one part to another within the structure. Therefore, we presented Extended Data Fig. 10 and Extended Data Table 4 to show Correlation Coefficient (CC) and Q-score values of each amino acid residue, protein subunit, ligand, and water molecule for the valid evaluation by the readers.

For this reason, it is difficult to determine all Chls located in the outer region of PSI complex, if any. This matter was described from lines 2 - 8 in page 8.

Notwithstanding the fact that the local resolution of some parts of the outer region is somewhat lower, the PSI was stable enough to keep its integrity for five days (Supplementary Fig. 5). Chls that were unable to be assigned in *A. marina* PSI compared with those in *T. elongatus* PSI are shown by marking N/D in the column of the ID number in Supplementary Table 6. Such Chls can be recognized by the chlorins in transparent gray in Supplementary Fig. 17. Chls corresponding to Chl-88 and Chl-94 in *T. elongatus* appear to be absent in *A. marina* PSI.

C1: there are significant less number of chlorophylls in PSI, which is so different from previous reports on Chl d-PSI system isolated from Acaryochloris. For example, Hu et al 1998, Tomo et al 2008. Hastings et al 2008... None of previous purified PSI studies indicates the "less number of Chl in PSI".

A1: This matter was also brought up by Reviewer #3. The reply to Reviewer #3 (A11 in the previous reply) is reprinted here.

As is recognized by the reviewer, the resolutions of the outer regions of *A. marina*-PSI is low (we have already shown this data in Extended Data Fig. 11 (A)), and some parts in the outer region may have been desorbed. Also, some pigments could be released during sample preparation. However, considering the standard errors among independent 5 preparations (Extended Data Table 7) that is smaller than the previous report (Tomo et al. 2008), there is still possibility that the number of pigments could be actually smaller than other type PSI. One paragraph describing such situation and unassigned Chls compared with *T. elongatus* PSI was added as follows in the revised manuscript (second paragraph in page7).

(second paragraph in page7)

Although the amount of Chl *d* per Chl *d'* determined by pigment analysis (67.0 ± 0.66 , $n=5$ of independently prepared PSI) and assigned by structural analysis (71 including one or two Chl *a*) in this study is lower than those in previous studies (145 ± 8^{14} and 97.0 ± 11.0^{12}), the semi-stoichiometric amount of Phe *a* (1.92 ± 0.022) that was 0.3 ± 0.2 per reaction center in the previous study¹² is consistent with the structural analysis in this study. Although local resolution of some parts of outer region is somewhat lower, our PSI was stable to keep its

integrity for five days (Extended Data Fig. 5). Therefore, we deduce here that the number of Chl associated with PSI is smaller than in other oxygenic phototrophs. Actually, Chls corresponding to Chl-88 and Chl-94 in *T. elongatus* are surely absent in *A. marina* PSI. Chls that were unable to be assigned in *A. marina* PSI compared with those in *T. elongatus* PSI are shown by marking N/D in the column of ID number in Extended Data Table 6, and those Chls can be recognized by chlorins in transparent gray in Extended Data Fig. 17. The numbers of Chls in PSI of Chl *f*-carrying *Fischerella thermalis* and *H. hongdechloris* are also smaller than that of *T. elongatus* PSI; 89 (*F. thermalis*)³⁰ and 90 (*H. hongdechloris*)¹⁹ Chls vs 96 Chls in *T. elongatus* PSI. Type I reaction center of *Heliobacterium modesticaldum* carries much smaller amount of Chl species; 60 molecules per reaction center²⁹.

Tomo, T. *et al.* Characterization of highly purified photosystem I complexes from the chlorophyll *d*-dominated cyanobacterium *Acaryochloris marina* MBIC 11017. *J. Biol. Chem.* **283**, 18198–18209 (2008).

None of the previous studies of purified PSI indicated the stoichiometric presence of Pheo *a*. However, Reviewer #3 (Comment #2) shed doubt as to the accuracy of the HPLC analysis itself to determine small amounts of pigments such as Chl *a* in the presence of Chl *d*. As is described in the above reprint, local resolution of some parts of the outer region in the structure is somewhat lower which makes it difficult to definitely assign pigments. Accordingly, the possibility that the number of pigments in *A. marina* PSI is larger than that reported here cannot be excluded although Chls corresponding to Chl-88 and Chl-94 (and Chl-95) in *T. elongatus* are surely absent in *A. marina* PSI. Therefore, we deleted the following sentence from the reprint above.

Therefore, we deduce here that the number of Chl associated with PSI is smaller than in other oxygenic phototrophs.

C2: dubious presence of Chl *a*. The sample may be contaminated by PSII, resulting the inconsistency between HPLC analysis and the structural assignment.

A2: Yes, the sample may be contaminated by PSII. However, if the contamination level is large, then, the number of Chl should be largely different in the structural model and HPLC analysis. Further, the standard error is small among five independently prepared samples. Additionally, BN-PAGE analysis showed a stable single band (Supplementary Fig. 5). Therefore, we estimated that the contribution of pigments containing contaminated PSII was minimal. Published reports (Hu *et al.* 1998; Tomo *et al.* 2008.) indicated by this Reviewer in the comment above (C1) also described the presence of Chl *a* in the purified PSI of *A. marina*. The number of pigments is not usually consistent

between the structural model and HPLC analysis, which is exemplified by the report of PSII crystal from *T. elongatus* (Kern et al., BBA 2005).

Hu, Q. *et al.* A photosystem I reaction center driven by chlorophyll d in oxygenic photosynthesis. *Proc. Natl. Acad. Sci.* **95**, 13319–13323 (1998).

Tomo, T. *et al.* Characterization of highly purified photosystem I complexes from the chlorophyll d-dominated cyanobacterium *Acaryochloris marina* MBIC 11017. *J. Biol. Chem.* **283**, 18198–18209 (2008).

Kern, J. *et al.* Purification, characterisation and crystallisation of photosystem II from *Thermosynechococcus elongatus* cultivated in a new type of photobioreactor, *Biochim Biophys Acta* 1706 (1-2): 147-57 (2005)

C3: if authors propose the H-bonded formyl groups (due to presence of water molecules), spectrum of this chlorophyll may be shifted, maybe, showing the spectral profiles similar to C3-hydroxymethyl-Chl a, or reduced Chl d?. Both of them have the similar HPLC profiles as Chl a if you use reverse-phase HPLC for pigment analysis.

A3: Hydrogen bond of formyl groups with water molecules should be lost during the pigment extraction process. As replied above (C2), the presence of Chl a in purified *A. marina* PSI is previously reported (Hu *et al.* 1998; Tomo *et al.* 2008).

Hu, Q. *et al.* A photosystem I reaction center driven by chlorophyll d in oxygenic photosynthesis. *Proc. Natl. Acad. Sci.* **95**, 13319–13323 (1998).

Tomo, T. *et al.* Characterization of highly purified photosystem I complexes from the chlorophyll d-dominated cyanobacterium *Acaryochloris marina* MBIC 11017. *J. Biol. Chem.* **283**, 18198–18209 (2008).

C4: Better quality PSI complex is needed to address the questions and remove any ambiguous structural assignments.

A4: Yes, ideally, it is the best. However, science is always the accumulation of knowledge. Please remember the first report of PSII crystal from *T. elongatus* in 2001. The resolution was as low as 3.8Å. However, the information presented by the structural model was exceptionally important. After that report, the resolution was improved by several groups resulting in 1.9Å resolution for *T. vulcanus* PSII taking 10 years.

We know that the resolution is lower in the marginally region of *A. marina* PSI studied in this work. Therefore, in this manuscript, we focused on the cofactors involved in electron transfer reaction locating in the central part of PSI complex where the resolution is high. We want to cite the comment by Reviewer #2 here. “The quality of cryo-EM map in the AmPSI core region is good, consistent with the claimed resolution of

2.5 Å and the structural model fits well with the map. Overall, the structure of the Chl *d*-rich and pheophytin *a*-containing PSI from *A. marina* is novel and will be helpful for further mechanistic studies if published.” We are expecting that future study will improve the resolution of the marginally region and provide information on the antenna system removing any ambiguous structural assignments.

REVIEWERS' COMMENTS

Reviewer #1 (Remarks to the Author):

"Far-red light captured by Chl f in the PSIs does not have enough energy to drive the photochemical reaction with the standard special pair of Chls a, and the uphill process is believed to overcome the shortage by utilizing heat energy from its surroundings." The authors should take out this sentence. The topic is far more complex than that as discussed in the literature. The authors do not need to discuss this point in the manuscript, but they must avoid oversimplifications.

Reviewer #2 (Remarks to the Author):

The authors have constructively addressed all my previous questions satisfyingly. The manuscript has been improved significantly. I have no further questions.

Reviewer #4 (Remarks to the Author):

Thank you for the nice revision.

Only a minor comment:

The statement about "uphill" on page 4, line 88-89, should be referred to "Schmitt et al (2019) Photosynthesis Res. 139:185-201".

Reviewer #1:

C: “Far-red light captured by Chl *f* in the PSIs does not have enough energy to drive the photochemical reaction with the standard special pair of Chls *a*, and the uphill process is believed to overcome the shortage by utilizing heat energy from its surroundings.” The authors should take out this sentence. The topic is far more complex than that as discussed in the literature. The authors do not need to discuss this point in the manuscript, but they must avoid oversimplifications.

A: The sentence was removed.

Reviewer #2:

C: The authors have constructively addressed all my previous questions satisfyingly. The manuscript has been improved significantly. I have no further questions.

A: Thank you for your positive evaluation.

Reviewer #4:

C: Thank you for the nice revision.

Only a minor comment:

The statement about "uphill" on page 4, line 88-89, should be referred to "Schmitt et al (2019) Photosynthesis Res. 139:185-201".

A: According to the comment by Reviewer #1, the sentence referring “uphill” was removed. The paper is cited (#27) at the following modified part as follows.

Still, PSII in some cyanobacteria may use Chl *f* (and/or Chl *d*) for electron transfer when grown under far-red light conditions^{21,26}. In contrast to Chl *f*-carrying PSIs^{19,20,22,23,25,27}, Chl *d* in *A. marina* PSI is always induced even under natural white light and does, in fact, take the place of the special pair Chls in the photochemical reaction chain.

19. Kato, K. *et al.* Structural basis for the adaptation and function of chlorophyll *f* in photosystem I. *Nat. Commun.* **11**, 238 (2020).
20. Kaucikas, M., Nürnberg, D., Dorlhiac, G., Rutherford, A. W. & van Thor, J. J. Femtosecond Visible Transient Absorption Spectroscopy of Chlorophyll *f*-Containing Photosystem I. *Biophys. J.* **112**, 234–249 (2017).
22. Gisriel, C. *et al.* The structure of Photosystem I acclimated to far-red light illuminates an ecologically important acclimation process in photosynthesis. *Sci. Adv.* **6**, eaay6415 (2020).
23. Gisriel, C. J., Wang, J., Brudvig, G. W. & Bryant, D. A. Opportunities and challenges for assigning cofactors in cryo-EM density maps of chlorophyll-containing proteins. *Commun. Biol.* **3**, 408 (2020).
25. Cherepanov, D. A. *et al.* Evidence that chlorophyll *f* functions solely as an antenna pigment in far-red-light photosystem I from *Fischerella thermalis* PCC 7521. *Biochim. Biophys. Acta - Bioenerg.* **1861**, 148184 (2020).
27. Schmitt, F. J. *et al.* Photosynthesis supported by a chlorophyll *f*-dependent, entropy-driven uphill energy transfer in *Halomicronema hongdechloris* cells adapted to far-red light. *Photosynth. Res.* **139**, 185–201 (2019).